# SPG: Sandwiched Policy Gradient for Masked Diffusion Language Models

**Chenyu Wang**[1,2,†], **Paria Rashidinejad**[1,3,†], **DiJia Su**[1], **Song Jiang**[1],
**Sid Wang**[1], **Siyan Zhao**[1,4,†], **Cai Zhou**[2], **Shannon Zejiang Shen**[1,2,†],
**Feiyu Chen**[1], **Tommi Jaakkola**[2], **Yuandong Tian**[1], **Bo Liu**[1]
[1]Meta Superintelligence Labs  [2]MIT  [3]USC  [4]UCLA

⌂ https://github.com/facebookresearch/SPG    🌐 Project Page

## Abstract

Diffusion large language models (dLLMs) are emerging as an efficient alternative to autoregressive models due to their ability to decode multiple tokens in parallel. However, aligning dLLMs with human preferences or task-specific rewards via reinforcement learning (RL) is challenging because their intractable log-likelihood precludes the direct application of standard policy gradient methods. While prior work uses surrogates like the evidence lower bound (ELBO), these one-sided approximations can introduce significant policy gradient bias. To address this, we propose the Sandwiched Policy Gradient (SPG) that leverages both an upper and a lower bound of the true log-likelihood. Experiments show that SPG significantly outperforms baselines based on ELBO or one-step estimation. Specifically, SPG improves the accuracy over state-of-the-art RL methods for dLLMs by 3.6% in GSM8K, 2.6% in MATH500, 18.4% in Countdown and 27.0% in Sudoku.

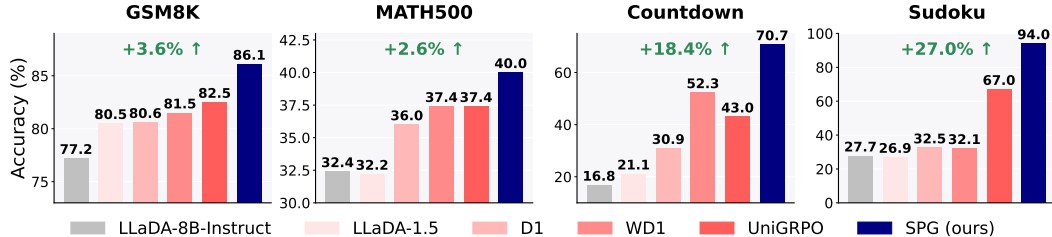

Figure 1: Test accuracy of SPG and baseline methods on four mathematical and logical reasoning benchmarks. All methods are evaluated with a generation length of 256 in 128 denoising steps. Full results are provided in Table 1.

## 1 Introduction

Diffusion models, originally pioneered for high-fidelity image generation (Song et al., 2020; Ho et al., 2020), have recently emerged as a powerful and efficient paradigm for text generation (Austin et al., 2021a; Campbell et al., 2022; Sun et al., 2022; Lou et al., 2023; Sahoo et al., 2024; Shi et al., 2024). These models operate in a discrete space but share architectural similarities with their continuous counterparts (Peebles & Xie, 2023). They employ a fixed noising process that progressively corrupts text data, while a neural network is trained to learn the reverse, denoising process. For instance, Masked Diffusion Language Model (MDLM) (Sahoo et al., 2024) uses random masking as its forward noising process and optimizes an Evidence Lower Bound (ELBO) of the log-likelihood. This ELBO-based objective has been widely adopted by subsequent large-scale diffusion language models (dLLMs), including LLaDA (Nie et al., 2025) and DREAM (Gong et al., 2024).

A key advantage of dLLMs over their autoregressive (AR) counterparts is their ability to decode multiple tokens in parallel. This parallelism can significantly reduce inference latency, making it an attractive alternative for scalable language modeling (Wang et al., 2025a; Labs et al., 2025).

---

[†]Work done at Meta.

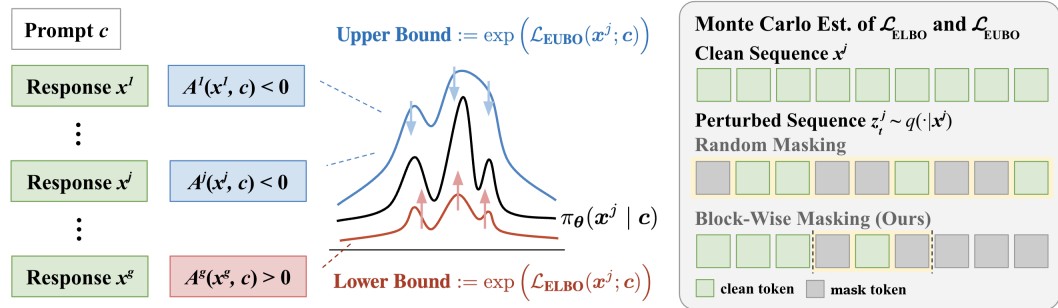

Figure 2: The training process of SPG for MDLM. *Left:* From a prompt $c$, we generate responses $\{x^j\}_{j=1}^g$. We then maximize a lower bound on the likelihood $\pi_\theta(x^j \mid c)$ for high-reward responses while minimizing an upper bound for low-reward ones. *Right:* The upper/lower bound of likelihood is estimated via Monte Carlo using a block-wise masking strategy, where a random block is selected for masking, with earlier blocks kept clean and later blocks fully masked. The example shows a sequence of length 9 with a block size of 3, where the current generation block is highlighted in yellow.

Aligning large language models with human preferences (Ouyang et al., 2022) or task-specific rewards (e.g., inducing reasoning behavior) (Shao et al., 2024; Guo et al., 2025) typically requires a post-training stage of reinforcement learning (RL). However, applying RL to dLLMs remains underexplored. A principal challenge is the computationally intractable log-likelihood of dLLMs, which is essential for accurate policy gradient estimation. To circumvent this, recent works (Zhao et al., 2025; Yang et al., 2025; Zhu et al., 2025; Tang et al., 2025) adapt standard RL and preference optimization algorithms, such as GRPO (Shao et al., 2024) and DPO (Rafailov et al., 2023), by using the ELBO or a one-step estimation as a surrogate for the true likelihood. While straightforward, this approximation leads to misaligned policy gradients, and potential suboptimal performance. Huang et al. (2025) employs a trajectory-level policy update algorithm together with a decoding order prediction module, jointly optimizing both model predictions and the order of unmasking. However, accumulating gradients across the entire diffusion trajectory incurs significant computational costs.

To address these limitations, we propose Sandwiched Policy Gradient (SPG), a novel reinforcement learning algorithm for diffusion language models that computes a more robust and less biased policy gradient. As illustrated in Figure 2, our core idea is to "sandwich" the intractable log-likelihood of a generated sequence: we maximize a tractable lower bound for positive-reward sequences while minimizing an upper bound for negative-reward ones. To ensure a stable estimation of these bounds, we also propose a block-wise masking strategy that better aligns data distributions during policy rollout and optimization. SPG achieves state-of-the-art performance on four mathematical and logical reasoning benchmarks, improving accuracy by up to 3.6% on GSM8K, 2.6% on MATH500, 18.4% on Countdown, and 27.0% on Sudoku compared to the state-of-the-art RL algorithms for diffusion language models. We additionally extend SPG to coding tasks, achieving improved accuracy by 1.9% on HumanEval and 4.7% on MBPP.

In summary, our main contributions are:

- A new policy gradient algorithm, SPG, which reduces bias by optimizing sandwiched variational bounds based on reward.

- A block-wise masking technique that improves the stability of the training objective's estimation.

- State-of-the-art results among RL algorithms for diffusion language models on four reasoning benchmarks and two coding benchmarks, demonstrating the effectiveness of our approach.

## 2 BACKGROUND

In this section, we provide a brief overview of the masked diffusion language model (MDLM) and reinforcement learning for text diffusion models.

**Notation.** We denote scalars by lowercase letters $(x)$, vectors by bold lowercase $(\boldsymbol{x})$, and sequences by $\boldsymbol{x}_{1:n}$. $[k]$ represents $\{1, \dots, k\}$. $\mathrm{Cat}(\boldsymbol{x} \mid \boldsymbol{p})$ is the categorical distribution over $\boldsymbol{x}$ with probabilities $\boldsymbol{p}$, and $\mathcal{U}[a, b]$ denotes the uniform distribution in $[a, b]$. Throughout the paper, we use $i \in [n]$ for position of the token, $j \in [g]$ for a sequence in a group of rollouts, and $t$ for the diffusion timestep. For discrete time processes, $t \in [T]$, while for continuous-time Markov chains, $t \in [0, 1]$.

## 2.1 MASKED DIFFUSION LANGUAGE MODELS

Diffusion models for language learn to generate text by reversing a gradual noising process. Specifically, Masked Diffusion Language Models (MDLMs) (Sahoo et al., 2024) start with clean text $\boldsymbol{x}_{1:n}$ and corrupt it into $\boldsymbol{z}_t \equiv \boldsymbol{z}_{t,1:n}$ over a continuous timestep $t \in [0, 1]$ by progressively replacing tokens with a special [mask] token. At $t = 0$, the data is original $(\boldsymbol{z}_0 = \boldsymbol{x})$, while at $t = 1$, the sequence is fully masked $(\boldsymbol{z}_1$ is all [mask] tokens). Each token is corrupted independently according to the forward transition kernel:

$$q_{t|0}(\boldsymbol{z}_{t,i} \mid \boldsymbol{x}_i) = \mathrm{Cat}\big(\boldsymbol{z}_{t,i} \mid \alpha_t \boldsymbol{x}_i + (1 - \alpha_t)\boldsymbol{m}\big), \tag{1}$$

where $\boldsymbol{m}$ is the one-hot representation of the [mask] token. The noise schedule, $\alpha_t \in [0, 1]$, is a strictly decreasing function, such as the linear schedule $\alpha_t = 1 - t$, with $\alpha_0 = 1$ and $\alpha_1 = 0$.

In the reverse process, a neural network, which we denote as the policy $\pi_{\boldsymbol{\theta}}$, is then trained to perform the reverse process: predicting the original tokens $\boldsymbol{x}$ from a corrupted version $\boldsymbol{z}_t$. The transition from $\boldsymbol{z}_t$ to $\boldsymbol{z}_s$ $(s < t)$ is parameterized with $\pi_{\boldsymbol{\theta}}$ as follows:

$$p_{\boldsymbol{\theta}}(\boldsymbol{z}_s \mid \boldsymbol{z}_t) = q(\boldsymbol{z}_s \mid \boldsymbol{z}_t, \boldsymbol{x} = \pi_{\boldsymbol{\theta}}(\cdot \mid \boldsymbol{z}_t)) = \begin{cases} \mathrm{Cat}(\mathbf{z}_s; \mathbf{z}_t), & \mathbf{z}_t \neq \mathbf{m}, \\ \mathrm{Cat}\left(\mathbf{z}_s; \frac{(1-\alpha_s)\mathbf{m} + (\alpha_s - \alpha_t)\pi_{\boldsymbol{\theta}}(\cdot|\boldsymbol{z}_t)}{1 - \alpha_t}\right), & \mathbf{z}_t = \mathbf{m}. \end{cases}$$

The policy is achieved by maximizing the Evidence Lower Bound (ELBO) of the log-likelihood of each clean sequence $\boldsymbol{x} \sim p_{\mathrm{data}}$, which simplifies to the following objective:

$$\mathcal{L}_{\mathrm{ELBO}}(\boldsymbol{x}; \boldsymbol{\theta}) = \mathbb{E}_{t,\boldsymbol{z}_t}\left[\sum_{i=1}^{n} w(t) \cdot \mathbb{1}(\boldsymbol{z}_{t,i} = \boldsymbol{m}) \cdot \log \pi_{\boldsymbol{\theta}}(\boldsymbol{x}_i \mid \boldsymbol{z}_t)\right], \tag{2}$$

where $w(t) = \alpha_t'/(\alpha_t - 1)$ is a time-dependent loss weight, and the expectation is over a random timestep $t \sim \mathcal{U}[0, 1]$ and the corrupted sequence $\boldsymbol{z}_t \sim q_{t|0}(\cdot \mid \boldsymbol{x})$. In essence, this objective trains the model to "fill in the blanks" by predicting the original tokens at masked positions. For a more comprehensive overview of MDLM, please refer to Appendix B and Sahoo et al. (2024).

## 2.2 REINFORCEMENT LEARNING FOR DIFFUSION LANGUAGE MODELS

Reinforcement Learning (RL) aligns a language model with desired objectives by treating it as a policy $\pi_{\boldsymbol{\theta}}$ that generates a response $\boldsymbol{x}$ to a prompt $\boldsymbol{c}$. This can be defined as a Markov decision process (MDP) (Black et al., 2023), where at each timestep $t$, the model observes the prompt $\boldsymbol{c}$ and a partially masked sequence $\boldsymbol{z}_t$ as the current state, takes an action based on the policy $\pi_{\boldsymbol{\theta}}$, and transitions to a new state $\boldsymbol{z}_{t-1}$ based on the action and inference strategy. A reward function $R(\boldsymbol{c}, \boldsymbol{x})$ provides a scalar score for the complete response, and the training goal is to update $\boldsymbol{\theta}$ to maximize the expected reward: $\mathcal{J}(\boldsymbol{\theta}) := \mathbb{E}_{\boldsymbol{x} \sim \pi_{\boldsymbol{\theta}}(\cdot|\boldsymbol{c})}[R(\boldsymbol{c}, \boldsymbol{x})]$. This objective is commonly optimized using policy gradient methods, which rely on the following gradient estimator.

$$\nabla_{\boldsymbol{\theta}}\mathcal{J}(\boldsymbol{\theta}) = \mathbb{E}_{\boldsymbol{x} \sim \pi_{\boldsymbol{\theta}}(\cdot|\boldsymbol{c})}\left[R(\boldsymbol{c}, \boldsymbol{x})\nabla_{\boldsymbol{\theta}}\log \pi_{\boldsymbol{\theta}}(\boldsymbol{x} \mid \boldsymbol{c})\right]. \tag{3}$$

**The Intractability Challenge.** A central challenge in applying RL to diffusion models is that the policy's log-likelihood, $\log \pi_{\boldsymbol{\theta}}(\boldsymbol{x} \mid \boldsymbol{c})$, is *intractable* and cannot be computed directly. To overcome this, prior work (Zhu et al., 2025; Yang et al., 2025) approximates this term using its ELBO, effectively replacing $\log \pi_{\boldsymbol{\theta}}(\boldsymbol{x} \mid \boldsymbol{c})$ with a score derived from the pre-training objective in Equation (2).

However, this popular workaround introduces a critical flaw. The ELBO is only a *lower bound* on the true log-likelihood (ELBO $\leq \log \pi_{\boldsymbol{\theta}}$). Consequently, the RL objective is only a valid lower bound on the true expected reward if all rewards $R(\boldsymbol{c}, \boldsymbol{x})$ are *non-negative*. This constraint prevents the model from effectively learning from negative feedback (i.e., penalizing bad outputs) and is incompatible with advanced RL algorithms that use relative or negative rewards (Shao et al., 2024), biasing the final policy. Our work aims to resolve this limitation.

---

**Algorithm 1** SPG: Sandwiched Policy Gradient for Masked dLLMs

---

**Require:** prompt distribution $\mathcal{D}$, number of completions per prompt $g$, number of inner updates $\mu$, forward process $q$, number of Monte Carlo samples $m$, initial policy $\pi_0$, learning rate $\epsilon$.

1: Initialize $\pi_\theta \leftarrow \pi_0$
2: **while** not converged **do**
3:     Sample a prompt $c \sim \mathcal{D}$, then $g$ completions $\{x^j \sim \pi_\theta(\cdot \mid c)\}_{j=1}^g$
4:     $\forall j \in [g]$, compute reward $R(c, x^j)$ and advantage $A^j(x^j, c)$
5:     **for** gradient update iterations $\{1, \ldots, \mu\}$ **do**
6:         $\forall j \in [g]$, generate $m$ perturbed samples $\{z_{t_\tau}^j\}_{\tau=1}^m \sim q(\cdot \mid x^j)$ via block-wise masking (Section 3.3).
7:         Compute the sandwiched policy gradient $\nabla \mathcal{J}_{\text{SPG}}(\theta)$ where:

$$\mathcal{J}_{\text{SPG}}(\theta) = \mathbb{E}\left[\frac{1}{g}\sum_{j=1}^g \left(\mathbb{1}_{A^j \geq 0} \cdot A^j \mathcal{L}_{\text{ELBO}}(x^j \mid c; \theta) + \mathbb{1}_{A^j < 0} \cdot A^j \tilde{\mathcal{L}}_{\text{EUBO}}(x^j \mid c; \theta)\right)\right],$$

8:         and $\mathcal{L}_{\text{ELBO}}, \tilde{\mathcal{L}}_{\text{EUBO}}$ are estimated from $\{z_{t_\tau}^j\}_{\tau=1}^m$, using Equation 2 and 7.
9:         Perform gradient update: $\theta \leftarrow \theta + \epsilon \nabla \mathcal{J}_{\text{SPG}}(\theta)$
10: **return** $\pi_\theta$

---

# 3    Sandwiched Policy Gradient with Evidence Bounds

We introduce SPG, a novel policy gradient algorithm designed for masked diffusion language models (Algorithm 1). Our method aims to address a critical issue in applying reinforcement learning to dLLMs by creating a valid optimization objective based on tractable bounds of the model's evidence.

## 3.1    A Lower Bound Objective for Policy Optimization

Our approach is based on group relative policy optimization (Shao et al., 2024; Liu et al., 2025b). For a given prompt $c$, we generate a group of $g$ responses $\{x^j\}_{j=1}^g$ from the policy $\pi_\theta$. We then compute the advantage $A^j(c, x^j) := R(c, x^j) - \frac{1}{g}\sum_{j=1}^g R(c, x^j)$. Moreover, we transform the conventional policy optimization objective as an advantage-weighted log-likelihood objective, for reasons that will be clear later:

$$\mathcal{J}^{\text{group}}(\theta) = \mathbb{E}_{c, \{x^j\} \sim \pi_{\text{sg}[\theta]}}\left[\frac{1}{g}\sum_{j=1}^g A^j(x^j, c) \log \pi_\theta(x^j \mid c)\right], \tag{4}$$

where $\text{sg}[\theta]$ indicates that gradients are not computed for the policy that generates the samples. This objective encourages generations with positive advantages ($A^j > 0$) and discourages those with negative advantages ($A^j < 0$).

For dLLMs, the log-likelihood $\log \pi_\theta$ is intractable. A common surrogate is the evidence lower bound (ELBO). While maximizing the ELBO is a valid way to increase the true log-likelihood, *minimizing* the ELBO for negatively-rewarded samples does not guarantee a reduction in the true log-likelihood. To address this, we propose a *sandwiched* objective. For samples with positive advantages, we maximize the ELBO. For samples with negative advantages, we instead minimize a tractable evidence *upper* bound (EUBO), $\mathcal{L}_{\text{EUBO}}$. This creates a true lower bound for the original objective:

$$\mathcal{J}_{\text{SPG}}(\theta) = \mathbb{E}\left[\frac{1}{g}\sum_{j=1}^g \left(\mathbb{1}_{A^j \geq 0} \cdot A^j \mathcal{L}_{\text{ELBO}}(x^j \mid c; \theta) + \mathbb{1}_{A^j < 0} \cdot A^j \mathcal{L}_{\text{EUBO}}(x^j \mid c; \theta)\right)\right], \tag{5}$$

where the expectation is take with respect to $c, \{x^j\} \sim \pi_{\text{sg}[\theta]}$. Since $\mathcal{L}_{\text{ELBO}} \leq \log \pi_\theta \leq \mathcal{L}_{\text{EUBO}}$, it follows that $\mathcal{J}_{\text{SPG}}(\theta) \leq \mathcal{J}^{\text{group}}(\theta)$. Maximizing this tractable bound therefore serves as a valid proxy for optimizing the true objective.

### 3.2 A Tractable Evidence Upper Bound

To effectively penalize negatively-rewarded samples by minimizing their log-likelihood, we require a tractable EUBO, which we derive in the following theorem based on the Rényi variational bound.

**Theorem 1** (Evidence Upper Bound for Masked Diffusion). *Assume the forward denoising process has $T$ steps with a monotonic schedule $\alpha_t$. For any $\beta \geq 1$ and a sequence $\boldsymbol{x}_{1:n}$, we have:*

$$\mathcal{L}_{EUBO}(\boldsymbol{x}_{1:n}; \boldsymbol{\theta}) = \frac{1}{\beta} \sum_{i=1}^{n} \log \sum_{t=1}^{T-1} \mathbb{E}_{\boldsymbol{z}_{t+1}} \left[ \frac{\alpha_t - \alpha_{t+1}}{1 - \alpha_{t+1}} \cdot \mathbb{1}(\boldsymbol{z}_{t+1,i} = \boldsymbol{m}) \cdot \pi_{\boldsymbol{\theta}}^{\beta}(\boldsymbol{x}_i \mid \boldsymbol{z}_{t+1}) \right] + C(T), \quad (6)$$

*where $C(T) := \mathbb{1}(\beta < n) \cdot \frac{1}{\beta} \log \mathbb{E}_{\boldsymbol{z}_{1:T} \sim q(\cdot|\boldsymbol{x})} \left[ q(\boldsymbol{z}_{1:T} \mid \boldsymbol{x})^{-n} \right]$ is a constant independent of $\boldsymbol{\theta}$.*

Here, $\beta \geq 1$ is a hyperparameter that controls the tightness of the bound, with values closer to 1 yielding a tighter bound. The expectation is taken over the timestep $t \sim \mathcal{U}[0, 1]$ and the noised latent $\boldsymbol{z}_t \sim q_{t|0}(\cdot \mid \boldsymbol{x})$.

**Corollary 1.** *Taking the limit of $T \to \infty$, we have:*

$$\nabla_{\boldsymbol{\theta}} \mathcal{L}_{EUBO}(\boldsymbol{x}_{1:n}; \boldsymbol{\theta}) = \nabla_{\boldsymbol{\theta}} \left( \tilde{\mathcal{L}}_{EUBO}(\boldsymbol{x}_{1:n}; \boldsymbol{\theta}) + C(T) \right) = \nabla_{\boldsymbol{\theta}} \tilde{\mathcal{L}}_{EUBO}(\boldsymbol{x}_{1:n}; \boldsymbol{\theta}), \quad \text{where}$$

$$\tilde{\mathcal{L}}_{EUBO}(\boldsymbol{x}_{1:n}; \boldsymbol{\theta}) = \frac{1}{\beta} \sum_{i=1}^{n} \log \mathbb{E}_{t, \boldsymbol{z}_t} \left[ w(t) \cdot \mathbb{1}(\boldsymbol{z}_{t,i} = \boldsymbol{m}) \cdot \pi_{\boldsymbol{\theta}}^{\beta}(\boldsymbol{x}_i \mid \boldsymbol{z}_t) \right]. \quad (7)$$

In practice, we estimate $\tilde{\mathcal{L}}_{\text{EUBO}}$ using Monte Carlo sampling and plug it in Equation 5 in place of $\mathcal{L}_{\text{EUBO}}$. The proof and theoretical analysis are provided in Appendix D.

**Remark.** A key structural difference from $\mathcal{L}_{\text{ELBO}}$ is that the logarithm in $\mathcal{L}_{\text{EUBO}}$ (Equation (6)) appears outside the expectation. Therefore, in practice, due to Jensen's inequality, applying the concave logarithm to a Monte Carlo estimate of the expectation's argument yields a biased estimate of the true EUBO. While it is possible to derive a looser but unbiased bound using inequalities like $\log(x) \leq x - 1$, we found this approach empirically worse by widening the gap to the true log-likelihood, as shown in Table 14. We therefore retain the tighter, albeit slightly biased, formulation.

### 3.3 Practical Considerations

**Block-Wise Masking Strategy for Monte Carlo Estimation.** In practice, we approximate $\mathcal{L}_{\text{ELBO}}$ and $\tilde{\mathcal{L}}_{\text{EUBO}}$ in Equation (5) via Monte Carlo sampling: for each $\boldsymbol{x}^j$, we randomly sample $m$ timesteps $\{t_\tau\}_{\tau=1}^m$ and generate the corresponding partially masked samples $\{\boldsymbol{z}_{t_\tau}^j\}_{\tau=1}^m \sim q(\cdot \mid \boldsymbol{x}^j)$. One straightforward approach as used in Yang et al. (2025) would be to apply random masking to clean sequences. However, recent dLLMs like LLaDA (Nie et al., 2025) employ a block-wise semi-autoregressive unmasking strategy during generation and achieve state-of-the-art performance over random unmasking. As a result, the policy rollout process actually encounters a much narrower and more structured set of partially masked sequences than with fully random masking.

To better align data distributions during policy rollout and optimization, we adopt a block-wise masking strategy rather than random masking. As depicted in Figure 2, the sequence is divided into several blocks, and a random block is selected, with all preceding blocks left clean and all following blocks fully masked. Within the chosen block, tokens are randomly masked. Additionally, following D1 (Zhao et al., 2025), we lightly perturb the prompt and clean blocks by randomly masking tokens with a small probability $p_{\text{mask}} = 0.15$ to enhance stability and generalization.

Altogether, our block-wise masking strategy improves the stability of the objective's estimation and the efficiency of policy optimization. While similar block-wise masking approaches have been explored in concurrent work for supervised fine-tuning or block diffusion models (Sun et al., 2025; Wang et al., 2025b), our focus is on RL for full-attention masked dLLMs. As shown in Figure 6, our models trained with block-wise masking generalize well to various inference strategies.

**Mixture of Upper and Lower Bound for Negative Advantage Traces.** Monte Carlo estimation of Equation (6) leads to a biased estimation to $\tilde{\mathcal{L}}_{\text{EUBO}}$ and potentially requires a substantial number of

samples to get reliable approximations, resulting in high computational costs and instability during training. To address these challenges, we use a mixture of $\tilde{\mathcal{L}}_{\text{EUBO}}$ and $\mathcal{L}_{\text{ELBO}}$ as a more practical log-likelihood approximation for negative advantage traces:

$$\tilde{\mathcal{L}}_{\text{Mix}}(\boldsymbol{x} \mid \boldsymbol{c}; \boldsymbol{\theta}) := \omega \cdot \tilde{\mathcal{L}}_{\text{EUBO}}(\boldsymbol{x} \mid \boldsymbol{c}; \boldsymbol{\theta}) + (1 - \omega) \cdot \mathcal{L}_{\text{ELBO}}(\boldsymbol{x} \mid \boldsymbol{c}; \boldsymbol{\theta}) \tag{8}$$

where $0 \leq \omega \leq 1$ is a blend coefficient. Intuitively, the upper bound $\tilde{\mathcal{L}}_{\text{EUBO}}$ sharpens the model decisions by applying a $\beta$-power adjustment to the original model output, acting as a strong correction signal for negative advantage traces. In contrast, the lower bound $\mathcal{L}_{\text{ELBO}}$ is easier and more stable to estimate with a small number of Monte Carlo samples, but it tends to introduce larger, systematic bias relative to the true log-likelihood. In particular, as a conservative approximation, $\mathcal{L}_{\text{ELBO}}$ alone is insufficient for effectively penalizing negative advantage traces, thus limiting its efficacy. Therefore, combining them allows us to harness the strengths of each, resulting in a more effective log-likelihood estimation in practice. In the following proposition, we formalize the advantages of using the mixture by deriving the gradient of the mixture loss and analyzing the variance of the gradient.

**Proposition 1** (Optimal Mixture Strictly Reduces Variance). *Fix a coordinate $k$ and let $\rho_\beta :=$ $w(t, \boldsymbol{z}_t)\pi_{\boldsymbol{\theta}}^\beta(\boldsymbol{x}_i \mid \boldsymbol{z}_t, \boldsymbol{c})/\mathbb{E}\left[w(t, \boldsymbol{z}_t)\pi_{\boldsymbol{\theta}}^\beta(\boldsymbol{x}_i \mid \boldsymbol{z}_t, \boldsymbol{c})\right]$, where $w(t, \boldsymbol{z}_t) := w(t)\mathbb{1}(z_t = \boldsymbol{m})$. Then, the gradient of mixture objective (8) is given by*

$$g_{\omega,k} = ((1 - \omega)w(t, \boldsymbol{z}_t) + \omega\rho_\beta)\, \partial_{\boldsymbol{\theta}_k} \log \pi_{\boldsymbol{\theta}}(\boldsymbol{x} \mid \boldsymbol{z}_t, \boldsymbol{c}). \tag{9}$$

*If $\text{Var}((\rho_\beta - w(t, \boldsymbol{z}_t))\partial_{\boldsymbol{\theta}_k} \log \pi_{\boldsymbol{\theta}}(\boldsymbol{x} \mid \boldsymbol{z}_t, \boldsymbol{c})) > 0$, then $\text{Var}[g_{\omega,k}]$ is a strictly convex quadratic in $\omega$ and thus admits a unique minimizer $\omega_k^\star$. Moreover,*

$$\text{Var}[g_{\omega_k^\star,k}] \;<\; \min\big\{\,\text{Var}[g_{0,k}],\; \text{Var}[g_{1,k}]\,\big\},$$

A proof for the above proposition is provided in Appendix E.1. A few remarks are in order:

- **Confidence-aware weighting:** The mixture gradient in Equation (9) realizes a *confidence-aware weighting*: uncertain tokens with small $\pi_{\boldsymbol{\theta}}^\beta(\boldsymbol{x}_i \mid \boldsymbol{z}_t, \boldsymbol{c})$, indicating a low recovery chance, have a smaller weight, while confident tokens with large $\pi_{\boldsymbol{\theta}}^\beta(\boldsymbol{x}_i \mid \boldsymbol{z}_t, \boldsymbol{c})$ are upweighted. The sharpness is controlled by parameter $\beta$ and the blend by $\omega$. Furthermore, the convex interpolation of the confidence-aware coefficient of the upper bound with the lower bound ensures clipping tiny gradients to a minimum value and thus prevents vanishing gradients.

- **Lower variance and more stable training:** According to Proposition 1, the gradient of the optimal mixture, i.e., $g_{\omega_k^\star,k}$, has strictly smaller coordinate-wise variance than the gradient of either the lower bound ($g_{0,k}$) or the upper bound ($g_{1,k}$)[1]. In our experiments, we fix $\beta$ and $\omega$ as hyperparameters for simplicity. These values can also be adaptively adjusted during training to better match the evolving training dynamics and data distribution.

Thus, the mixture approach offers theoretical advantages over using either the upper or lower bound alone, as supported by our experimental results in Section 4. Further discussions of the mixture approach and empirical evidence of reduced gradient variance are provided in Appendix E.2 and Figure 7, and Appendix E.3 presents a toy example illustrating the distinct behaviors of the lower and upper bounds. We provide additional discussions on the RL algorithm, including RL stabilization techniques and a reward shifting alternate, in Appendix F.

## 4 EXPERIMENTS

In this section, we present experimental results highlighting the superior performance of SPG across various benchmarks. Further, we provide detailed analysis and ablations of SPG to assess the contribution of each component, examine the influence of key hyperparameters, and evaluate the robustness of our approach under different inference strategies.

---

[1]Proposition 1 extends directly to a single, coordinate-independent optimizer $\omega^\star$ obtained by minimizing the sum of coordinate-wise variances.

Table 1: Model performance on four reasoning benchmarks. The best results are bolded and the second best are underlined. SPG consistently outperforms all other methods. We denote the absolute gain of test accuracy to the previous state-of-the-art in green.

| Model / Seq Len | GSM8K (0-shot) | | | MATH500 (0-shot) | | | Countdown (0-shot) | | | Sudoku (3-shot) | | |
|---|---|---|---|---|---|---|---|---|---|---|---|---|
| | 128 | 256 | 512 | 128 | 256 | 512 | 128 | 256 | 512 | 128 | 256 | 512 |
| LLaDA-8B-Inst. | 69.5 | 77.2 | 79.8 | 28.2 | 32.4 | 34.6 | 18.8 | 16.8 | 16.8 | 5.7 | 27.7 | 26.2 |
| LLaDA-1.5 | 70.4 | 80.5 | 81.9 | 26.8 | 32.2 | 35.8 | 21.9 | 21.1 | 21.5 | 7.4 | 26.9 | 29.0 |
| D1 | 72.2 | 80.6 | 81.3 | 31.4 | 36.0 | _39.4_ | 30.9 | 30.9 | 34.4 | 7.2 | 32.5 | 29.3 |
| WD1 | 74.6 | 81.5 | 83.0 | 31.0 | 37.4 | 39.0 | 48.8 | 52.3 | 50.8 | 33.1 | 32.1 | 22.5 |
| UniGRPO | 74.9 | 82.5 | 82.7 | 32.4 | 37.4 | _39.4_ | 44.5 | 43.0 | 57.0 | 59.0 | 67.0 | 62.9 |
| SPG w/ EUBO | _77.1_ | _83.8_ | _83.9_ | _33.2_ | _37.6_ | 39.4 | _68.4_ | **71.5** | 68.0 | _81.2_ | _87.1_ | _89.9_ |
| SPG w/ Mixture | **78.5**$_{+3.6}$ | **86.1**$_{+3.6}$ | **84.5**$_{+1.5}$ | **33.4**$_{+1.0}$ | **40.0**$_{+2.6}$ | **41.8**$_{+2.4}$ | **68.8**$_{+20}$ | _70.7_$_{+18}$ | **70.3**$_{+13}$ | **82.9**$_{+24}$ | **94.0**$_{+27}$ | **93.1**$_{+30}$ |

## 4.1 EXPERIMENTAL SETUP AND MAIN RESULTS

**Experimental Setup.** We conduct RL fine-tuning with SPG following the experimental settings in D1 (Zhao et al., 2025) and WD1 (Tang et al., 2025). We employ LLaDA-8B-Instruct (Nie et al., 2025), a state-of-the-art open-sourced dLLM without post-training, as the base model, and experiment on six benchmarks: two for mathematical reasoning (GSM8K (Cobbe et al., 2021) and MATH500 (Lightman et al., 2023)), two for logical reasoning (Countdown (Pan et al., 2025) and Sudoku (Arel, 2025)), and two for coding (HumanEval (Chen, 2021) and MBPP (Austin et al., 2021b)). We follow the same train-test splitting, reward functions, and evaluation protocol as D1 and WD1, except for Sudoku. For Sudoku, to avoid train-test leakage, we take the training set from D1 and split the data by Sudoku answers, ensuring that the test set contains entirely new puzzle solutions. This guarantees that the model cannot solve test puzzles merely by memorizing possible answers. All experiments are conducted in the zero-shot setting, except for Sudoku, where 3-shot generation is used for both training and evaluation[2]. For all models, we employ Low-Rank Adaptation (LoRA) with a rank of $r = 128$ and scaling factor $\alpha = 64$. For SPG, we report results using both $\tilde{\mathcal{L}}_{\text{EUBO}}$ (i.e., SPG w/ EUBO) and $\tilde{\mathcal{L}}_{\text{Mix}}$ (i.e., SPG w/ Mixture) for negative advantage traces. We select the value of $\beta$ in the EUBO from $\{1.0, 1.5, 2.0\}$ based on the best average test accuracy across all generation lengths, and fix the mixture coefficient $\omega$ at 0.5. Further experimental details are in Appendix G.1 and Appendix G.2.

**Baselines.** We compare our method with several recent RL algorithms for dLLMs, including D1 (Zhao et al., 2025), WD1 (Tang et al., 2025), and UniGRPO (Yang et al., 2025). For D1 and WD1, we reproduce results using the official codebases and instructions, and for fair comparison, we omit the additional SFT stage in D1 across all models. For UniGRPO, since the code is not publicly available and the original work focuses on vision-language multimodal models, we reimplement the algorithm within our setup. For consistency, we set the number of inner gradient updates $\mu$ to 4 for all models, following GRPO (Shao et al., 2024). We also evaluate LLaDA-1.5 (Zhu et al., 2025) under our settings, which fine-tune LLaDA-8B-Instruct using VRPO, a preference optimization approach on 350K preference pairs. We provide a detailed overview on these RL algorithms for dLLMs in Appendix C and Table 5.

**Generation and Evaluation Setup.** For both RL rollouts and evaluation, we use the semi-autoregressive confidence-based decoding strategy, following LLaDA, D1 and WD1. We apply the same generation setup as D1, with the denoising timestep set to half the total sequence length. The sequence is divided into blocks of 32 tokens, and in diffusion step, we unmask the 2 tokens with the highest confidence (measured by the probability of the sampled token) within the current incomplete block. During RL rollout, to encourage diverse outputs, we use a generation length of 256 and a sampling temperature of 0.9 across all benchmarks, except for sudoku, where the temperature is set to 0.3 as in D1. During evaluation, the sampling temperature is set to 0.0. We evaluate the models every 100 steps, reporting results from the checkpoint that achieves the highest average test accuracy across generation lengths of 128, 256, and 512.

---

[2]We use 3-shot generation for Sudoku because zero-shot is too difficult for this task, resulting in very few meaningful RL rollouts. Few-shot examples used in our experiments are provided in Appendix G.3.

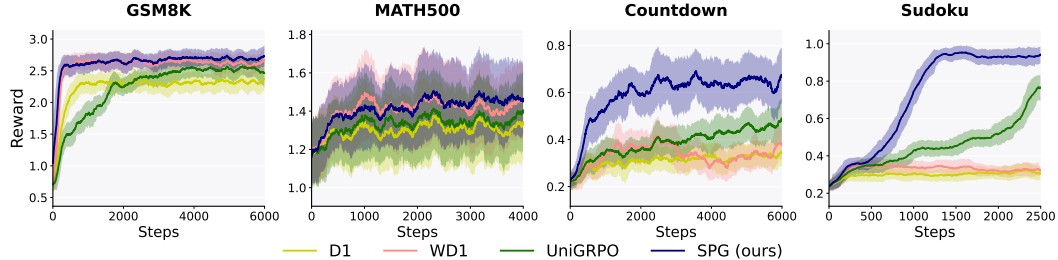

Figure 3: Reward dynamics of SPG w/ Mixture during RL training, compared with D1, WD1, and UniGRPO. SPG consistently leads to faster convergence and higher reward level. We report mean and standard deviation over a rolling window of 50 steps.

**Results.** We provide the performance of SPG on each of the four reasoning benchmarks in comparison to the base model and other baselines in Table 1. Both SPG w/ EUBO and SPG w/ Mixture consistently achieve significant improvements over the baselines across all tasks and generation lengths, with the Mixture approach that combines ELBO and EUBO for negative advantage traces yielding the best performance. In particular, at a generation length of 256, SPG w/ Mixture improves the test accuracy over the previous state-of-the-art by **3.6%** on GSM8K, **2.6%** on MATH500, **18%** on Countdown, and **27%** on Sudoku, showcasing the effectiveness of SPG to conduct RL for dLLMs. Reward dynamics throughout training are illustrated in Figure 3, where SPG shows a rapid and steady increase in reward over the optimization steps, further demonstrating its efficiency and robustness. We provide additional results and comparisons to the baselines in Table 8 and Appendix H.1.

**Extending SPG to coding domains.** We further extend SPG to coding tasks, fine-tuning the LLaDA-8B-Instruct base model on the KodCode-Light-RL-10K dataset (Xu et al., 2025) and evaluating on HumanEval and MBPP benchmarks, following the settings in D1. As shown in Table 2, SPG consistently improves the accuracy on both benchmarks over the baselines across different generation lengths, demonstrating its strong ability in the coding domain.

Table 2: Model performance on HumanEval and MBPP. The best results are bolded and the second best are underlined. Absolute test accuracy gains over the previous state-of-the-art are shown in green. SPG consistently improves accuracies on coding tasks.

| Model / Seq Len | HumanEval (0-shot) | | | MBPP (0-shot) | | |
|---|---|---|---|---|---|---|
| | 128 | 256 | 512 | 128 | 256 | 512 |
| LLaDA-8B-Inst. | 28.1 | 35.4 | 34.8 | 36.2 | 42.0 | 38.1 |
| LLaDA-1.5 | 22.6 | 34.8 | 32.3 | 33.5 | 44.8 | 40.1 |
| D1 | 29.3 | 37.8 | 37.2 | 42.0 | 45.5 | 43.2 |
| WD1 | 25.6 | 39.0 | 38.4 | 38.9 | 45.9 | 44.8 |
| UniGRPO | 28.1 | 39.6 | 38.4 | 39.3 | 45.5 | 45.9 |
| SPG w/ EUBO | 29.3 | **41.5**+1.9 | 33.5 | 43.6 | 46.7 | 46.3 |
| SPG w/ Mixture | **31.1**+1.8 | 39.0 | **39.6**+1.2 | **46.3**+4.3 | **50.6**+4.7 | **49.0**+3.1 |

### 4.2 ABLATIONS AND FURTHER ANALYSIS

We conduct a series of ablation studies to gain deeper insights from the following aspects:

- The contribution of each individual component, including log-likelihood estimation methods for negative advantage traces (Table 3) and the masking strategy in Monte Carlo estimation (Table 4).

- The effect of key hyperparameters, including $\beta$ that controls the tightness of the upper bound and the mixture coefficient $\omega$ (Figure 5).

- The robustness of our approach under various inference strategies (Figure 6).

Due to computational constraints, some ablation experiments are conducted on a representative mathematical reasoning benchmark (MATH500) and a logical reasoning benchmark (Countdown). Unless otherwise noted, we report average test accuracy across generation lengths 128, 256, and 512 for the ablation studies, with detailed results for each generation length provided in Appendix H.2. In Appendix H.2, we also investigate alternative log-likelihood estimation methods for positive advantage traces in place of ELBO, as detailed in Table 15, and study the diversity of model generations by evaluating the pass@K performance of each model in Table 16. Additionally, we conduct ablations on full fine-tuning, prompt masking, and number of Monte Carlo samples, and

Table 3: Ablations on log-likelihood estimation methods for negative advantage traces. The best results are bolded and the second best underlined. We denote the absolute gain of test accuracy to SPG w/ ELBO in green. SPG w/ Mixture consistently outperforms other likelihood estimation methods.

| Model | GSM8K | MATH500 | Countdown | Sudoku |
|---|---|---|---|---|
| SPG wo/ neg | 77.4 | 32.7 | 45.5 | 68.8 |
| SPG w/ ELBO | 80.9 | 37.4 | 67.1 | 82.4 |
| SPG w/ EUBO | 81.6 | 36.7 | 69.3 | 86.1 |
| SPG w/ Mixture | 83.1+2.2 | 38.4+1.0 | 69.9+2.8 | 90.0+7.6 |

Table 4: Ablations on the masking strategies in Monte Carlo estimation. We denote the absolute gain of test accuracy to random masking for each model in green. Our block-wise masking strategy leads to consistent improvement to random masking on both benchmarks.

| Model | Masking | MATH500 | Countdown |
|---|---|---|---|
| SPG w/ EUBO | random | 36.7 | 45.4 |
| | block-wise | 36.7+0.0 | 69.3+23.9 |
| SPG w/ Mixture | random | 36.9 | 62.8 |
| | block-wise | 38.4+1.5 | 69.9+7.1 |

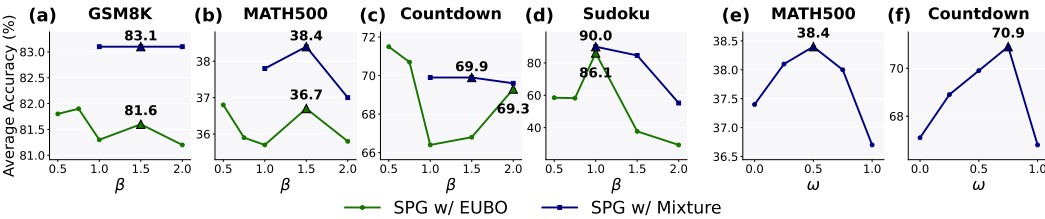

Figure 5: *(a)-(d)*: ablations on the effect of $\beta$ in the upper bound; *(e)-(f)*: ablations on the mixture coefficient $\omega$. The best performed $\beta \geq 1$ and $\omega \in [0, 1]$ are marked by triangle in each setting.

provide the computational cost comparison in Appendix H.2. We also conduct further comparison with the variant of UniGRPO and the validation-based checkpoint selection in Appendix H.3.

**Ablations on Algorithm Components.** We first study the impact of different log-likelihood estimation methods for negative advantage traces in Table 3. Specifically, we compare our approach using $\tilde{\mathcal{L}}_{\text{EUBO}}$ or $\tilde{\mathcal{L}}_{\text{Mix}}$ with those using $\mathcal{L}_{\text{ELBO}}$ (SPG w/ ELBO) or omitting the negative advantage loss entirely (SPG wo/ neg). Removing the negative advantage loss results in a substantial performance drop, highlighting the importance of negative advantage penalties to RL. Additionally, both Mixture and EUBO methods outperform ELBO (except for EUBO in MATH500), showcasing the benefits of evidence upper bound regularization for negative rewards. We provide complete results for each generation length in Table 10.

The effect of log-likelihood estimation methods is further illustrated by the reward dynamics of each model in Figure 4, taking Sudoku as an example. SPG w/ ELBO converges rapidly during training but plateaus early, as minimizing the lower bound does not necessarily minimize the true log-likelihood for negative advantage traces. In contrast, SPG w/ EUBO achieves higher final rewards but converges more slowly and less stably. Combining both, SPG w/ Mixture attains fast, stable convergence and high rewards, leading to an effective balance. This aligns with our discussions in Section 3.3.

We also conduct ablations on the masking strategies in Monte Carlo estimation of $\mathcal{L}_{\text{ELBO}}$, $\tilde{\mathcal{L}}_{\text{EUBO}}$, and $\tilde{\mathcal{L}}_{\text{Mix}}$. As shown in Table 4, the block-wise masking strategy outperforms random masking, demonstrating the importance of aligning input distributions between policy rollout and optimization. We provide complete results for each generation length in Table 11.

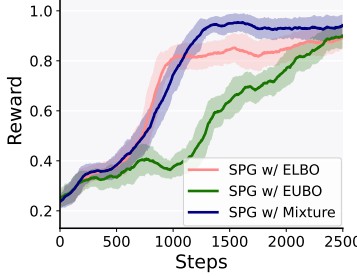

Figure 4: Reward dynamics of different log-likelihood estimation methods for negative advantage traces on Sudoku. SPG w/ Mixture leads to both fast convergence and high rewards.

**Ablations on Key Hyperparameters $\beta$ and $\omega$.** We first examine the effect of $\beta$, a crucial hyperparameter in evidence upper bound estimation, in panels (a)-(d) of Figure 5. In general, a relatively small value of $\beta$ (i.e., close to 1.0) leads to a tighter bound and thus better performance. Nevertheless, SPG consistently

Figure 6: Ablations on inference strategies, including different combinations of decoding orders (i.e., semi-autoregressive (semi-AR) decoding with varying block sizes and full sequence decoding) and unmasking approaches (i.e., confidence-based and random unmasking). We set generation length to 256 and report the average accuracy across four benchmarks. SPG consistently outperforms all baselines by a large margin across different inference strategies.

performs well across a range of $\beta$ values on most tasks, indicating its robustness. For our main results in Table 1, we fix $\omega = 0.5$ and select the optimal $\beta \geq 1$, resulting in $\beta = 1.0$ for Sudoku and $\beta = 1.5$ for the other three benchmarks, except for Countdown with SPG w/ EUBO where $\beta = 2.0$. Besides, since the ELBO corresponds to the case of $\beta = 0$ theoretically and EUBO corresponds to $\beta \geq 1$, we also investigate intermediate values $0 < \beta < 1$, which may serve as an implicit mixture of lower and upper bounds. However, it is unstable in Sudoku and underperform SPG w/ Mixture on most benchmarks.

We also experiment on the effect of the mixture coefficient $\omega$, keeping $\beta$ fixed at its optimal value determined for $\omega = 0.5$ as mentioned before. As illustrated in panels (e)-(f) of Figure 5, combining lower and upper bounds with $\omega \in (0, 1)$ leads to better performance than leveraging either bound solely, resulting in an inverted U-shaped curve. This observation is consistent with our analysis in Proposition 1 and Section 3.3. We provide complete ablation results of $\beta$ and $\omega$ for each generation length in Table 12 and Table 13.

**Ablations on Inference Strategies.** In the above experiments, we adopt a consistent state-of-the-art inference setup during both RL rollout and evaluation, i.e., confidence-based, block-wise semi-autoregressive generation with a block size of 32. The same configuration and block size are also used in our block-wise masking strategy. This raises the question of whether our approach generalizes well to alternative inference strategies. To assess this, we evaluate the base model and all RL fine-tuned models using various inference strategies, as shown in Figure 6. Despite being trained under confidence-based semi-AR decoding, SPG consistently outperforms all baselines by a substantial margin across all inference strategies, demonstrating its robustness and strong generalizability. Complete results for each benchmark individually are provided in Table 17.

## 5 CONCLUSION

We propose SPG, a novel reinforcement learning algorithm for diffusion large language models. SPG addresses the intractable log-likelihood in dLLMs by maximizing a tractable lower bound on positive reward sequences and minimizing an upper bound on negative ones, resulting in a more robust and less biased policy gradient. Additionally, we propose a block-wise masking strategy for Monte Carlo estimation to enhance optimization stability and efficiency. Extensive experiments on four mathematical and logical reasoning benchmarks demonstrate the superior performance of SPG, achieving significant improvement over baselines and the state-of-the-art performance.

ACKNOWLEDGEMENT

CW and TJ acknowledge support from the NSF Expeditions grant (award 1918839) Understanding the World Through Code. CW is supported in part by the GQS PhD Fellowship.

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

## A  RELATED WORK

**Diffusion Language Models.**  Building on the remarkable success of diffusion models for image generation in continuous domains (Song et al., 2020; Ho et al., 2020), researchers have explored their extension to discrete data such as text. Initial attempts focused on training continuous diffusion models in the text embedding space (Li et al., 2022; Gong et al., 2022; Han et al., 2022; Sahoo et al., 2025a), while they face challenges in optimization and generalization due to the discrete nature of text data. Masked diffusion models (Lou et al., 2023; Zheng et al., 2023; Campbell et al., 2024; Sahoo et al., 2024; Shi et al., 2024) address this by defining the diffusion process directly in the discrete token space, using random masking as the forward process, and have achieved strong empirical results. Block Diffusion (Arriola et al., 2025) further advances this direction by combining the strengths of autoregressive models, such as the capability to generate variable-length outputs and using KV cache to accelerate inference, with the benefits of diffusion language models like parallel decoding and flexible, any-order generation within blocks. Recently, large-scale diffusion language models trained with masked diffusion objectives have demonstrated performance competitive with similarly sized autoregressive models (Nie et al., 2025; Gong et al., 2024). More recent works (Wu et al., 2025; Ma et al., 2025; Liu et al., 2025a; Sahoo et al., 2025a;b) have introduced caching and parallel decoding algorithms that greatly enhance the inference efficiency of dLLMs.

**Reinforcement Learning for LLMs and Reasoning.**  The seminal works apply reinforcement learning to large language models (LLMs) to align them with human preferences via reinforcement learning from human feedback (RLHF) (Christiano et al., 2017; Ouyang et al., 2022). More recently, reinforcement learning has proven highly effective at enhancing the reasoning abilities of LLMs during the post-training stage, where rewards can be provided by a process reward model (Lightman et al., 2023) or verifiable reward signals. Algorithms such as Proximal Policy Optimization (PPO) and Trust Region Policy Optimization (TRPO) constrain policy updates to a trust region, reducing variance and promoting stable learning by preventing excessive shifts from the reference policy (Schulman et al., 2015; 2017). Group Relative Policy Optimization (GRPO) (Shao et al., 2024) introduces group-relative rewards, enabling efficient training without the need for an additional value (critic) model. GRPO and its variants have demonstrated strong empirical performance in state-of-the-art models such as DeepSeek-R1 (Guo et al., 2025), particularly on mathematical reasoning tasks, where incorporating long reasoning traces with self-reflection and verification steps yields significant improvements. Recent works (Liu et al., 2025b; Zheng et al., 2025; Team et al., 2025; Cohen et al., 2025) further improve RL algorithms for LLMs by reducing the bias introduced by the GRPO objective, enhancing sample efficiency, and introducing additional regularization.

**Reinforcement Learning for Diffusion Language Models.**  Numerous studies have explored RL-based fine-tuning algorithms for diffusion models with continuous objectives (Fan et al., 2023; Black et al., 2023; Clark et al., 2023). While RL algorithms have achieved notable success to LLMs and continuous diffusion models, their applications to diffusion language models in the discrete space remain underexplored. DRAKES (Wang et al., 2024) leverages reward backpropagation along the denoising trajectory, but is computationally intensive for large scale models as the gradients are propagated through each denoising step. Alternatively, methods like D1 (Zhao et al., 2025) and UniGRPO Yang et al. (2025) utilize the GRPO framework, approximating the log-likelihood through either a one-step unmasking (as in D1) or Monte Carlo estimation using the ELBO (as in UniGRPO). VRPO (Zhu et al., 2025) adapts DPO (Rafailov et al., 2023) to fine-tune dLLMs by applying MC estimation of the ELBO. WD1 (Tang et al., 2025) starts from the GRPO formulation and the same log-likelihood estimation as in D1, while avoiding direct estimation of the old and reference policy log-likelihoods by integrating them into a weighted policy optimization objective. LLaDOU (Huang et al., 2025) decomposes the likelihood along the diffusion steps and utilizes a trajectory-level policy update algorithm for fine-tuning, while such stepwise approach is computationally costly. It also employs a decoding order prediction module and jointly optimizes both model predictions and the order of unmasking during inference. Despite these advances, a principled analysis of RL algorithms for dLLMs, especially the challenging log-likelihood estimation, is missing. This results in substantial bias in the optimization objective and suboptimal performance.

## B  BASICS OF DLLMs

In this section, we provide a more self-contained overview of masked dLLMs. Please also refer to Sahoo et al. (2024) for more details.

**Notation.**  We denote scalars by lowercase letters ($x$), vectors by bold lowercase ($\boldsymbol{x}$), and sequences by $\boldsymbol{x}_{1:n}$. A superscript (e.g., $\boldsymbol{x}^j$) denotes an item's index within a group. We define the set of the first $k$ integers as $[k] := \{1, \ldots, k\}$ and the $k$-dimensional probability simplex as $\Delta^{k-1}$. Distributions include the categorical $\text{Cat}(\cdot \mid \boldsymbol{p})$ and the uniform $\mathcal{U}[a, b]$. Throughout the paper, we use the following primary indices: $i \in [n]$ for position, $j \in [g]$ for a sequence in a group, and $t \in [0, 1]$ for the continuous diffusion timestep.

We start from a discrete time version of the diffusion models with finite $t \in [T]$. Assume a one-hot categorical variable $\boldsymbol{x} \in \{e_1, \ldots, e_k\} \subset \Delta^{k-1}$. Further assume we gradually corrupt $\boldsymbol{x}$ into an absorbing state $\boldsymbol{m}$ (i.e., $e_{[\texttt{mask}]}$ ) with transition matrix $\boldsymbol{Q}_t$ at time $t$. Then:

$$q(\boldsymbol{z}_t \mid \boldsymbol{x}) = \text{Cat}(\boldsymbol{z}_t \mid \overline{\boldsymbol{Q}_t}\boldsymbol{x}) = \text{Cat}(\boldsymbol{z}_t \mid \prod_{\tau=1}^{t} \boldsymbol{Q}_\tau \boldsymbol{x}).$$

Here, $\boldsymbol{z}_t$ is also a one-hot categorical random variable in $\Delta^{k-1}$. In practice, one could choose $\boldsymbol{Q}_t$ such that:

$$q(\boldsymbol{z}_t \mid \boldsymbol{x}) = \text{Cat}(\boldsymbol{z}_t \mid \alpha_t \boldsymbol{x} + (1 - \alpha_t)\boldsymbol{m}).$$

Here, $\alpha_1 = 1, \alpha_T = 0, \alpha_t' < 0$.

Normally, the goal is to construct the lower bound of the evidence (ELBO) and maximize it. For this particular case, consider the discretized Markov chain with $T$ latent variables $\boldsymbol{z}_1, \boldsymbol{z}_2, \ldots, \boldsymbol{z}_T$, where $\boldsymbol{z}_T = \boldsymbol{m}$ and $\boldsymbol{z}_1 = \boldsymbol{x}$. We use the shorthand $\boldsymbol{z} = \boldsymbol{z}_{1:T}$ and write

$$
\begin{aligned}
\mathcal{L}_{\text{ELBO}}(\boldsymbol{x}; \boldsymbol{\theta}) &= \mathbb{E}_{\boldsymbol{z} \sim q(\cdot \mid \boldsymbol{x})} \left[ \log \frac{p_{\boldsymbol{\theta}}(\boldsymbol{x}, \boldsymbol{z})}{q(\boldsymbol{z} \mid \boldsymbol{x})} \right] \\
&= \mathbb{E}_{\boldsymbol{z} \sim q(\cdot \mid \boldsymbol{x})} \left[ \underbrace{\log p_{\boldsymbol{\theta}}(\boldsymbol{x} \mid \boldsymbol{z}_1)}_{=0} + \sum_{t=1}^{T-1} \log \frac{p_{\boldsymbol{\theta}}(\boldsymbol{z}_t \mid \boldsymbol{z}_{t+1})}{q(\boldsymbol{z}_t \mid \boldsymbol{z}_{t+1}, \boldsymbol{x})} + \underbrace{\log \frac{p_{\boldsymbol{\theta}}(\boldsymbol{z}_T)}{q(\boldsymbol{z}_T \mid \boldsymbol{x})}}_{=0} \right] \\
&= \sum_{t=1}^{T-1} \mathbb{E}_{\boldsymbol{z}_t, \boldsymbol{z}_{t+1} \sim q} \left[ \log \frac{p_{\boldsymbol{\theta}}(\boldsymbol{z}_t \mid \boldsymbol{z}_{t+1})}{q(\boldsymbol{z}_t \mid \boldsymbol{z}_{t+1}, \boldsymbol{x})} \right] \\
&= \sum_{t=1}^{T-1} \mathbb{E}_{\boldsymbol{z}_{t+1} \sim q(\cdot \mid \boldsymbol{x})} \mathbb{E}_{\boldsymbol{z}_t \sim q(\cdot \mid \boldsymbol{z}_{t+1}, \boldsymbol{x})} \left[ \log \frac{p_{\boldsymbol{\theta}}(\boldsymbol{z}_t \mid \boldsymbol{z}_{t+1})}{q(\boldsymbol{z}_t \mid \boldsymbol{z}_{t+1}, \boldsymbol{x})} \right].
\end{aligned}
\tag{10}
$$

Here, $\log p_{\boldsymbol{\theta}}(\boldsymbol{x}, \boldsymbol{z}_1) = 0$ because we assume $\boldsymbol{z}_1 = \boldsymbol{x}$, and $p_{\boldsymbol{\theta}}(\boldsymbol{z}_T) = q(\boldsymbol{z}_T \mid \boldsymbol{x})$ because we assume $\boldsymbol{z}_T = \boldsymbol{m}$. A common method to parameterize $p_{\boldsymbol{\theta}}$ is via predicting $\boldsymbol{x}$ with model $\pi_{\boldsymbol{\theta}}$ in $q$:

$$p_{\boldsymbol{\theta}}(\boldsymbol{z}_t \mid \boldsymbol{z}_{t+1}) = q\left(\boldsymbol{z}_t \mid \boldsymbol{z}_{t+1}, \boldsymbol{x} = \pi_{\boldsymbol{\theta}}(\cdot \mid \boldsymbol{z}_{t+1})\right).$$

Now, given that $\boldsymbol{z}_{t+1}$ is either $\boldsymbol{m}$ or $\boldsymbol{x}$ (assuming $\boldsymbol{m} \neq \boldsymbol{x}$). Then the KL term in equation 10 decomposes into the following.

$$
\log \frac{p_{\boldsymbol{\theta}}(\boldsymbol{z}_t \mid \boldsymbol{z}_{t+1})}{q(\boldsymbol{z}_t \mid \boldsymbol{z}_{t+1}, \boldsymbol{x})} = \begin{cases} 0 & \boldsymbol{z}_t = \boldsymbol{z}_{t+1} = \boldsymbol{x}, \\ 0 & \boldsymbol{z}_t = \boldsymbol{m}, \boldsymbol{z}_{t+1} = \boldsymbol{x}, \quad \text{(Impossible)} \\ \log \pi_{\boldsymbol{\theta}}(\boldsymbol{x} \mid \boldsymbol{z}_{t+1}) & \boldsymbol{z}_t = \boldsymbol{x}, \boldsymbol{z}_{t+1} = \boldsymbol{m}, \\ 0 & \boldsymbol{z}_t = \boldsymbol{z}_{t+1} = \boldsymbol{m}. \end{cases}
\tag{11}
$$

Table 5: Comparisons of different RL algorithms for dLLMs.

| Model | Policy Update Algorithm | Log-Likelihood Estimation | Masking Strategy | Others |
|---|---|---|---|---|
| LLaDA-1.5 | DPO | ELBO | Random masking | Variance-reduced sampling budget allocation |
| D1 | GRPO | One-step estimation | – | Prompt masking |
| WD1 | Weighted policy optimization | One-step estimation | – | Prompt masking |
| UniGRPO | GRPO | ELBO | Random masking | Structured noising strategy |
| LLaDOU | Trajectory-level policy update | Stepwise probability | – | Decoding order prediction |
| SPG | Policy gradient | ELBO for positive traces; EUBO/Mixture for negative traces | Block-wise masking | Prompt masking |

Moreover, $q(\boldsymbol{z}_t = \boldsymbol{x} \mid \boldsymbol{z}_{t+1} = \boldsymbol{m}, \boldsymbol{x}) = \frac{\alpha_t - \alpha_{t+1}}{1 - \alpha_{t+1}}$, and note that $\pi_{\boldsymbol{\theta}}(\boldsymbol{x} \mid \boldsymbol{z}_t) = 1$ when $\boldsymbol{z}_t = \boldsymbol{x}$, so we have:

$$
\begin{aligned}
\mathcal{L}_{\mathrm{ELBO}}(\boldsymbol{x}; \boldsymbol{\theta}) &= \sum_{t=1}^{T-1} \mathbb{E}_{\boldsymbol{z}_{t+1} \sim q(\cdot|\boldsymbol{x})} \left[ \frac{\alpha_t - \alpha_{t+1}}{1 - \alpha_{t+1}} \log \pi_{\boldsymbol{\theta}}(\boldsymbol{x} \mid \boldsymbol{z}_{t+1}) \mathbb{1}(\boldsymbol{z}_{t+1} = \boldsymbol{m}) \right] \\
&= \sum_{t=1}^{T-1} \mathbb{E}_{\boldsymbol{z}_{t+1} \sim q(\cdot|\boldsymbol{x})} \left[ \frac{\alpha_t - \alpha_{t+1}}{1 - \alpha_{t+1}} \log \pi_{\boldsymbol{\theta}}(\boldsymbol{x} \mid \boldsymbol{z}_{t+1}) \right]. \quad \text{(If } \boldsymbol{z}_{t+1} = \boldsymbol{x}, \text{ then } \log \pi_{\boldsymbol{\theta}}(\boldsymbol{x} \mid \boldsymbol{z}_{t+1}) = 0\text{)}
\end{aligned}
\tag{12}
$$

Taking the above limit as $T \to \infty$, we have:

$$
\mathcal{L}_{\mathrm{ELBO}}(\boldsymbol{x}; \boldsymbol{\theta}) = \int_{t=0}^{1} \mathbb{E}_{\boldsymbol{z}_t \sim q(\cdot|\boldsymbol{x})} \left[ \frac{\alpha'_t}{\alpha_t - 1} \log \pi_{\boldsymbol{\theta}}(\boldsymbol{x} \mid \boldsymbol{z}_t) \right].
\tag{13}
$$

**Generalization to Sequence** The above is for a single categorical variable $\boldsymbol{x}$. In practice as in language modeling, it becomes a sequence of categorical variables $\boldsymbol{x}_{1:n}$. Then we write

$$
\begin{aligned}
\mathcal{L}_{\mathrm{ELBO}}(\boldsymbol{x}_{1:n}; \boldsymbol{\theta}) &= \mathbb{E}_{\boldsymbol{z}_{1:n} \sim q(\cdot|\boldsymbol{x}_{1:n})} \left[ \log \frac{p_{\boldsymbol{\theta}}(\boldsymbol{x}_{1:n}, \boldsymbol{z}_{1:n})}{q(\boldsymbol{z}_{1:n} \mid \boldsymbol{x}_{1:n})} \right] \\
&= \mathbb{E}_{\{\boldsymbol{z}_i \sim q(\cdot|\boldsymbol{x}_i)\}_{i=1}^n} \left[ \sum_{i=1}^{n} \log \frac{p_{\boldsymbol{\theta}}(\boldsymbol{x}_i, \boldsymbol{z}_{1:n})}{q(\boldsymbol{z}_i \mid \boldsymbol{x}_i)} \right] \quad \text{(Independence of } q(\cdot \mid \boldsymbol{x}_i)\text{)} \\
&= \sum_{i=1}^{n} \mathbb{E}_{\{\boldsymbol{z}_{i'} \sim q(\cdot|\boldsymbol{x}_{i'})\}_{i'=1}^n} \left[ \log \frac{p_{\boldsymbol{\theta}}(\boldsymbol{x}_i, \boldsymbol{z}_{1:n})}{q(\boldsymbol{z}_i \mid \boldsymbol{x}_i)} \right] \\
&= \sum_{i=1}^{n} \mathcal{L}_{\mathrm{ELBO}}(\boldsymbol{x}_i; \boldsymbol{\theta}).
\end{aligned}
\tag{14}
$$

The key distinction from the single-token formulation (mentioned beforehand) is that the reverse process $p_{\boldsymbol{\theta}}$ is conditioned on all $\boldsymbol{z}_{1:n}$ instead of a single token's $\boldsymbol{z}_i$.

## C  DETAILED OVERVIEW ON RL ALGORITHMS FOR DLLMS

In this section, we provide a comprehensive overview of the existing reinforcement learning algorithms for dLLMs that serve as baselines in Section 4. The main distinctions among these methods are highlighted in Table 5, covering aspects such as the policy update algorithm, log-likelihood estimation technique, masking strategy used in likelihood estimation, and other unique components. Below, we elaborate on each of these aspects.

LLaDA-1.5 (Zhu et al., 2025) utilizes the variance-reduced preference optimization (VRPO) algorithm to align the masked diffusion model with preference pairs data. It takes the DPO (Rafailov et al., 2023) algorithm and uses ELBO with random masking to estimate the likelihood terms in the DPO objective. Specifically, they analyzes the variance of the ELBO estimators and propose an optimal Monte Carlo budge allocation in the ELBO estimation.

D1 (Zhao et al., 2025) and UniGRPO (Yang et al., 2025) employs the GRPO (Shao et al., 2024) objective. D1uses the one-step prediction to estimate each likelihood term in the GRPO objective,

which predicts the the clean response sequence based on the partially masked prompt and the fully masked response. UniGRPO alternatively utilizes ELBO with random masking to estimate each likelihood terms. Moreover, UniGRPO takes a structured noising strategy which uniformly increases the noise level during the gradient update iterations.

WD1 (Tang et al., 2025) reformulates the GRPO objective while integrates the old and reference policy likelihoods into the policy rollout distribution. Then it leverages a single weighted policy optimization objective with the weights defined with the group relative advantages. It applies the same log-likelihood estimation method as D1, using the one-step prediction from partially masked prompt and the fully masked response.

Our SPG algorithm utilizes a simple the policy gradient objective with group-relative advantage, and estimates the log-likelihood term with ELBO for positive traces while using EUBO or the mixture of ELBO and EUBO for negative ones. We employ a block-wise masking strategy to generate the perturbed samples in likelihood estimation, and apply the same prompt masking trick as in D1.

In addition to optimizing the diffusion policy $\pi_{\boldsymbol{\theta}}(\boldsymbol{x}|\boldsymbol{c})$ itself, LLaDOU (Huang et al., 2025) incorporates a decoding order prediction module that learns the optimal unmasking order during inference. It decomposes the likelihood of the dLLMs along the diffusion steps and utilizes a trajectory-level policy update algorithm which accumulates the gradient from each diffusion step and jointly optimizes both model predictions and the inference-time decoding order. While learning an optimal decoding order is a promising avenue for enhancing inference strategies, which is complementary to advancements in RL algorithms, and can be integrated with SPG, we leave such an exploration to future work.

For the RL algorithm component, we reimplement the trajectory-level RL method from Huang et al. (2025) in our framework and present the results in Table 6 (denoted as "StepWise"). SPG consistently outperforms StepWise on both the MATH500 and Countdown benchmarks. Additionally, trajectory-level updates as used in Huang et al. (2025) require backpropagation through every diffusion timestep, leading to high computational costs. For instance, on MATH500 with $8\times$A100 GPUs, StepWise takes 7.5 minutes per gradient update, while SPG only takes 0.6 minutes.

Table 6: Comparison with the trajectory-level policy update algorithm (Huang et al., 2025), i.e., StepWise, on MATH500 and Countdown.

| | MATH500 (0-shot) | | | | Countdown (0-shot) | | | |
|---|---|---|---|---|---|---|---|---|
| Model / Seq Len | 128 | 256 | 512 | *Avg.* | 128 | 256 | 512 | *Avg.* |
| StepWise | 33.2 | 35.2 | 38.0 | 35.5 | 42.8 | 42.4 | 53.2 | 46.1 |
| SPG w/ EUBO | 33.2 | 37.6 | 39.4 | 36.7 | 68.4 | 71.5 | 68.0 | 69.3 |
| SPG w/ Mixture | 33.4 | 40.0 | 41.8 | 38.4 | 68.8 | 70.7 | 70.3 | 69.9 |

## D    EVIDENCE UPPER BOUND FOR DLLMS

In this section, we provide the derivation of the evidence upper bound. Following the above section, we start from the discrete time version of the diffusion models.

**Lemma 1** (Rényi Variational Bound; Rényi (1961); Van Erven & Harremos (2014))**.** *Fix an observation x. Let $q(\cdot \mid x)$ be any distribution on $\mathcal{Z}$ such that $p(\cdot \mid x) \ll q(\cdot \mid x)$, denoting that $p(\cdot \mid x)$ is absolutely continuous with respect to $q(\cdot \mid z)$. Then, the following holds for any $\beta \geq 1$:*

$$\mathbb{E}_{z\sim q(\cdot|x)}\left[\log \frac{p(x,z)}{q(z \mid x)}\right] \leq \log p(x) \leq \frac{1}{\beta} \log \mathbb{E}_{z\sim q(\cdot|x)}\left[\left(\frac{p(x,z)}{q(z \mid x)}\right)^{\beta}\right]. \qquad (15)$$

In view of the above lemma, we derive an evidence upper bound for masked diffusion models in the following theorem.

**Theorem 1** (Evidence Upper Bound for Masked Diffusion)**.** *Assume the forward denoising process has $T$ steps with a monotonic schedule $\alpha_t$. For any $\beta \geq 1$ and a sequence of categorical variables $\boldsymbol{x}_{1:n}$, we have:*

$$\log \pi_{\boldsymbol{\theta}}(\boldsymbol{x}_{1:n}) \leq \mathcal{L}_{EUBO}(\boldsymbol{x}_{1:n}; \boldsymbol{\theta}), \qquad (16)$$

*where*

$$\mathcal{L}_{EUBO}(\boldsymbol{x}_{1:n}; \boldsymbol{\theta}) := \frac{1}{\beta} \sum_{i=1}^{n} \log \sum_{t=1}^{T-1} \mathbb{E}_{\boldsymbol{z}_{t+1}} \left[ \frac{\alpha_t - \alpha_{t+1}}{1 - \alpha_{t+1}} \cdot \mathbb{1}(\boldsymbol{z}_{t+1,i} = \boldsymbol{m}) \cdot \pi_{\boldsymbol{\theta}}^{\beta}(\boldsymbol{x}_i \mid \boldsymbol{z}_{t+1}) \right] + C(T), \tag{17}$$

*and* $C(T) = \begin{cases} \frac{1}{\beta} \log \mathbb{E}_{\boldsymbol{z}_{1:T} \sim q(\cdot|\boldsymbol{x})} \left[ q(\boldsymbol{z}_{1:T} \mid \boldsymbol{x})^{-n} \right], & 1 \le \beta < n \\ 0, & \beta \ge n \end{cases}$ *is a constant independent of* $\boldsymbol{\theta}$.

*Proof.* We first consider the case with a single categorical variable $\boldsymbol{x}$. On the account of Lemma 1 and following a similar argument as in equation 10, for any $\beta \ge 1$, we can write

$$\log \pi_{\boldsymbol{\theta}}(\boldsymbol{x}) \le \frac{1}{\beta} \log \mathbb{E}_{\boldsymbol{z} \sim q(\cdot|\boldsymbol{x})} \left[ \left( \frac{p_{\boldsymbol{\theta}}(\boldsymbol{x}, \boldsymbol{z})}{q(\boldsymbol{z} \mid \boldsymbol{x})} \right)^{\beta} \right]$$

$$= \frac{1}{\beta} \log \mathbb{E}_{\boldsymbol{z}_{1:T} \sim q(\cdot|\boldsymbol{x})} \left[ \prod_{t=1}^{T-1} \left( \frac{p_{\boldsymbol{\theta}}(\boldsymbol{z}_t \mid \boldsymbol{z}_{t+1})}{q(\boldsymbol{z}_t \mid \boldsymbol{z}_{t+1}, \boldsymbol{x})} \right)^{\beta} \right] \tag{18}$$

Note that the sequence $\boldsymbol{z}_{1:T}$ has a form $\{\boldsymbol{x}, \dots, \boldsymbol{x}, \boldsymbol{m}, \dots, \boldsymbol{m}\}$. Define the transition event:

$$\mathcal{A}_t := \{\boldsymbol{z}_t = \boldsymbol{x}, \boldsymbol{z}_{t+1} = \boldsymbol{m}\} \tag{19}$$

Then, by the law of total expectations, equation 18 can be expressed as:

$$\frac{1}{\beta} \log \mathbb{E}_{\boldsymbol{z}_{1:T} \sim q(\cdot|\boldsymbol{x})} \left[ \prod_{t=1}^{T-1} \left( \frac{p_{\boldsymbol{\theta}}(\boldsymbol{z}_t \mid \boldsymbol{z}_{t+1})}{q(\boldsymbol{z}_t \mid \boldsymbol{z}_{t+1}, \boldsymbol{x})} \right)^{\beta} \right]$$

$$= \frac{1}{\beta} \log \sum_{t=1}^{T-1} \mathbb{P}(\mathcal{A}_t) \mathbb{E}_{\boldsymbol{z} \sim q(\cdot|\boldsymbol{x})} \left[ \prod_{s=1}^{T-1} \left( \frac{p_{\boldsymbol{\theta}}(\boldsymbol{z}_s \mid \boldsymbol{z}_{s+1})}{q(\boldsymbol{z}_s \mid \boldsymbol{z}_{s+1}, \boldsymbol{x})} \right)^{\beta} \Bigg| \mathcal{A}_t \right]$$

$$= \frac{1}{\beta} \log \sum_{t=1}^{T-1} \mathbb{E}_{\boldsymbol{z}_{t+1} \sim q(\cdot|\boldsymbol{x})} \left[ \mathbb{1}(\boldsymbol{z}_{t+1} = \boldsymbol{m}) q(\boldsymbol{z}_t = \boldsymbol{x} \mid \boldsymbol{z}_{t+1} = \boldsymbol{m}, \boldsymbol{x}) \left( \frac{p_{\boldsymbol{\theta}}(\boldsymbol{z}_t = \boldsymbol{x} \mid \boldsymbol{z}_{t+1} = \boldsymbol{m})}{q(\boldsymbol{z}_t = \boldsymbol{x} \mid \boldsymbol{z}_{t+1} = \boldsymbol{m}, \boldsymbol{x})} \right)^{\beta} \right]$$

$$= \frac{1}{\beta} \log \sum_{t=1}^{T-1} \mathbb{E}_{\boldsymbol{z}_{t+1} \sim q(\cdot|\boldsymbol{x})} \left[ \mathbb{1}(\boldsymbol{z}_{t+1} = \boldsymbol{m}) \frac{\alpha_t - \alpha_{t+1}}{1 - \alpha_{t+1}} \pi_{\boldsymbol{\theta}}^{\beta}(\boldsymbol{x} \mid \boldsymbol{z}_{t+1}) \right] \tag{20}$$

The penultimate line is due to the fact that conditioned on the event $\mathcal{A}_t$, the ratio $\frac{p_{\boldsymbol{\theta}}(\boldsymbol{z}_s|\boldsymbol{z}_{s+1})}{q(\boldsymbol{z}_s|\boldsymbol{z}_{s+1}, \boldsymbol{x})}$ is equal to one for any $s \ne t$. The last line uses the formula for $q$. The indicator $\mathbb{1}(\boldsymbol{z}_t = \boldsymbol{m})$ appears in the final expression because the terms in the bound are only non-trivial when the model must make a prediction from a corrupted state.

Now we generalize the above to a sequence of categorical variables $\boldsymbol{x} = \boldsymbol{x}_{1:n}$. Similar as Equation (18), we have

$$\log \pi_{\boldsymbol{\theta}}(\boldsymbol{x}_{1:n}) \le \frac{1}{\beta} \log \mathbb{E}_{\boldsymbol{z}_{1:T} \sim q(\cdot|\boldsymbol{x})} \left[ \prod_{t=1}^{T-1} \prod_{i=1}^{n} \left( \frac{p_{\boldsymbol{\theta}}(\boldsymbol{z}_{t,i} \mid \boldsymbol{z}_{t+1})}{q(\boldsymbol{z}_{t,i} \mid \boldsymbol{z}_{t+1}, \boldsymbol{x})} \right)^{\beta} \right]$$

Denote $\tilde{C}(T) := \frac{1}{\beta} \log \mathbb{E}_{\boldsymbol{z}_{1:T} \sim q(\cdot | \boldsymbol{x})} \left[ q(\boldsymbol{z}_{1:T} \mid \boldsymbol{x})^{-n} \right]$. The upper bound in the RHS can be further derived as

$$\frac{1}{\beta} \log \mathbb{E}_{\boldsymbol{z}_{1:T} \sim q(\cdot | \boldsymbol{x})} \left[ \prod_{t=1}^{T-1} \prod_{i=1}^{n} \left( \frac{p_{\boldsymbol{\theta}}(\boldsymbol{z}_{t,i} \mid \boldsymbol{z}_{t+1})}{q(\boldsymbol{z}_{t,i} \mid \boldsymbol{z}_{t+1}, \boldsymbol{x})} \right)^{\beta} \right]$$

$$= \frac{1}{\beta} \log \mathbb{E}_{\boldsymbol{z}_{1:T} \sim q(\cdot | \boldsymbol{x})} \left[ q(\boldsymbol{z}_{1:T} \mid \boldsymbol{x})^{-n} \prod_{i=1}^{n} \sum_{\boldsymbol{y}_{1:T}^i} q(\boldsymbol{y}_{1:T}^i \mid \boldsymbol{x}) \mathbb{1}(\boldsymbol{y}_{1:T}^i = \boldsymbol{z}_{1:T}) \prod_{t=1}^{T-1} \left( \frac{p_{\boldsymbol{\theta}}(\boldsymbol{y}_{t,i}^i \mid \boldsymbol{y}_{t+1}^i)}{q(\boldsymbol{y}_{t,i}^i \mid \boldsymbol{y}_{t+1}^i, \boldsymbol{x})} \right)^{\beta} \right]$$

$$\leq \frac{1}{\beta} \log \mathbb{E}_{\boldsymbol{z}_{1:T} \sim q(\cdot | \boldsymbol{x})} \left[ q(\boldsymbol{z}_{1:T} \mid \boldsymbol{x})^{-n} \prod_{i=1}^{n} \sum_{\boldsymbol{y}_{1:T}^i} q(\boldsymbol{y}_{1:T}^i \mid \boldsymbol{x}) \prod_{t=1}^{T-1} \left( \frac{p_{\boldsymbol{\theta}}(\boldsymbol{y}_{t,i}^i \mid \boldsymbol{y}_{t+1}^i)}{q(\boldsymbol{y}_{t,i}^i \mid \boldsymbol{y}_{t+1}^i, \boldsymbol{x})} \right)^{\beta} \right]$$

$$= \frac{1}{\beta} \log \left( \mathbb{E}_{\boldsymbol{z}_{1:T} \sim q(\cdot | \boldsymbol{x})} \left[ q(\boldsymbol{z}_{1:T} \mid \boldsymbol{x})^{-n} \right] \cdot \left( \prod_{i=1}^{n} \sum_{\boldsymbol{y}_{1:T}^i} q(\boldsymbol{y}_{1:T}^i \mid \boldsymbol{x}) \prod_{t=1}^{T-1} \left( \frac{p_{\boldsymbol{\theta}}(\boldsymbol{y}_{t,i}^i \mid \boldsymbol{y}_{t+1}^i)}{q(\boldsymbol{y}_{t,i}^i \mid \boldsymbol{y}_{t+1}^i, \boldsymbol{x})} \right)^{\beta} \right) \right)$$

$$= \frac{1}{\beta} \log \prod_{i=1}^{n} \mathbb{E}_{\boldsymbol{z}_{1:T} \sim q(\cdot | \boldsymbol{x})} \left[ \prod_{t=1}^{T-1} \left( \frac{p_{\boldsymbol{\theta}}(\boldsymbol{z}_{t,i} \mid \boldsymbol{z}_{t+1})}{q(\boldsymbol{z}_{t,i} \mid \boldsymbol{z}_{t+1}, \boldsymbol{x})} \right)^{\beta} \right] + \frac{1}{\beta} \log \mathbb{E}_{\boldsymbol{z}_{1:T} \sim q(\cdot | \boldsymbol{x})} \left[ q(\boldsymbol{z}_{1:T} \mid \boldsymbol{x})^{-n} \right]$$

$$= \frac{1}{\beta} \sum_{i=1}^{n} \log \mathbb{E}_{\boldsymbol{z}_{1:T} \sim q(\cdot | \boldsymbol{x})} \left[ \prod_{t=1}^{T-1} \left( \frac{p_{\boldsymbol{\theta}}(\boldsymbol{z}_{t,i} \mid \boldsymbol{z}_{t+1})}{q(\boldsymbol{z}_{t,i} \mid \boldsymbol{z}_{t+1}, \boldsymbol{x})} \right)^{\beta} \right] + \tilde{C}(T) \tag{21}$$

Here, $\boldsymbol{y}_{1:T}^i$ are copies of $\boldsymbol{z}_{1:T}$ enforced to agree with $\boldsymbol{z}_{1:T}$ using the indicator $\mathbb{1}(\boldsymbol{y}_{1:T}^i = \boldsymbol{z}_{1:T})$. $C(T)$ is a constant independent of $\boldsymbol{\theta}$, and the first term in Equation (21) can be derived similar to the single variable case in Equation (20):

$$\frac{1}{\beta} \sum_{i=1}^{n} \log \mathbb{E}_{\boldsymbol{z}_{1:T} \sim q(\cdot | \boldsymbol{x})} \left[ \prod_{t=1}^{T-1} \left( \frac{p_{\boldsymbol{\theta}}(\boldsymbol{z}_{t,i} \mid \boldsymbol{z}_{t+1})}{q(\boldsymbol{z}_{t,i} \mid \boldsymbol{z}_{t+1}, \boldsymbol{x})} \right)^{\beta} \right]$$

$$= \frac{1}{\beta} \sum_{i=1}^{n} \log \sum_{t=1}^{T-1} \mathbb{E}_{\boldsymbol{z}_{t+1} \sim q(\cdot | \boldsymbol{x})} \left[ \frac{\alpha_t - \alpha_{t+1}}{1 - \alpha_{t+1}} \cdot \mathbb{1}(\boldsymbol{z}_{t+1,i} = \boldsymbol{m}) \cdot \pi_{\boldsymbol{\theta}}^{\beta}(\boldsymbol{x}_i \mid \boldsymbol{z}_{t+1}) \right]$$

Additionally, for $\beta \geq n$, we can obtain a tighter bound using Hölder's inequality:

$$\frac{1}{\beta} \log \mathbb{E}_{\boldsymbol{z}_{1:T} \sim q(\cdot | \boldsymbol{x})} \left[ \prod_{t=1}^{T-1} \prod_{i=1}^{n} \left( \frac{p_{\boldsymbol{\theta}}(\boldsymbol{z}_{t,i} \mid \boldsymbol{z}_{t+1})}{q(\boldsymbol{z}_{t,i} \mid \boldsymbol{z}_{t+1}, \boldsymbol{x})} \right)^{\beta} \right]$$

$$= \frac{1}{n\beta} \sum_{i=1}^{n} \log \mathbb{E}_{\boldsymbol{z}_{1:T} \sim q(\cdot | \boldsymbol{x})} \left[ \prod_{t=1}^{T-1} \left( \frac{p_{\boldsymbol{\theta}}(\boldsymbol{z}_{t,i} \mid \boldsymbol{z}_{t+1})}{q(\boldsymbol{z}_{t,i} \mid \boldsymbol{z}_{t+1}, \boldsymbol{x})} \right)^{n\beta} \right]$$

$$= \frac{1}{n\beta} \sum_{i=1}^{n} \log \sum_{t=1}^{T-1} \mathbb{E}_{\boldsymbol{z}_{t+1} \sim q(\cdot | \boldsymbol{x})} \left[ \frac{\alpha_t - \alpha_{t+1}}{1 - \alpha_{t+1}} \cdot \mathbb{1}(\boldsymbol{z}_{t+1,i} = \boldsymbol{m}) \cdot \pi_{\boldsymbol{\theta}}^{n\beta}(\boldsymbol{x}_i \mid \boldsymbol{z}_{t+1}) \right] \tag{22}$$

Denote $\tilde{\beta} := n\beta$. Then the upper bound $\log \pi_{\boldsymbol{\theta}}(\boldsymbol{x}_{1:n}) \leq \mathcal{L}_{\text{EUBO}}(\boldsymbol{x}_{1:n}; \boldsymbol{\theta})$ holds with the constant term $C(T)$ being zero and the hyperparameter $\tilde{\beta} \geq n$. Combining these two cases, we have

$$C(T) = \begin{cases} \tilde{C}(T), & 1 \leq \beta < n \\ 0, & \beta \geq n \end{cases}.$$

$\square$

Furthermore, we can derive the continuous time version by omitting the constant term that does not affect the gradient with respect to $\boldsymbol{\theta}$, and taking the limit of $T \to \infty$ similar as the derivations for $\mathcal{L}_{\text{ELBO}}$, as shown in Corollary 1:

**Corollary 1.** *Taking the limit of $T \to \infty$, we have:*

$$\nabla_{\boldsymbol{\theta}} \mathcal{L}_{EUBO}(\boldsymbol{x}_{1:n}; \boldsymbol{\theta}) = \nabla_{\boldsymbol{\theta}} \left( \tilde{\mathcal{L}}_{EUBO}(\boldsymbol{x}_{1:n}; \boldsymbol{\theta}) + C(T) \right) = \nabla_{\boldsymbol{\theta}} \tilde{\mathcal{L}}_{EUBO}(\boldsymbol{x}_{1:n}; \boldsymbol{\theta}), \quad where$$

$$\tilde{\mathcal{L}}_{EUBO}(\boldsymbol{x}_{1:n}; \boldsymbol{\theta}) = \frac{1}{\beta} \sum_{i=1}^{n} \log \mathbb{E}_{t, \boldsymbol{z}_t} \left[ w(t) \cdot \mathbb{1}(\boldsymbol{z}_{t,i} = \boldsymbol{m}) \cdot \pi_{\boldsymbol{\theta}}^{\beta}(\boldsymbol{x}_i \mid \boldsymbol{z}_t) \right]. \tag{23}$$

One caveat of the above $\tilde{\mathcal{L}}_{\text{EUBO}}$ is that the $\log$ is outside of the expectation, which in general makes Monte Carlo sample estimates biased. One could certainly further loosen the bound using the inequality $\log x \leq x - 1$:

$$\mathcal{L}_{\text{EUBO}}(\boldsymbol{x}) \leq \frac{1}{\beta} \sum_{i=1}^{n} \mathbb{E}_{t \sim \mathcal{U}[0,1], \boldsymbol{z}_t \sim q} \left[ w(t) \cdot \mathbb{1}(\boldsymbol{z}_{t,i} = \boldsymbol{m}) \cdot \pi_{\boldsymbol{\theta}}^{\beta}(\boldsymbol{x}_i \mid \boldsymbol{z}_t) \right] - \frac{n}{\beta} \tag{24}$$

But in practice we found this results in much worse performance, as demonstrated in Table 14, potentially due to the much larger gap between EUBO and likelihood.

# E   ADDITIONAL ANALYSIS ON UPPER AND LOWER BOUNDS

## E.1   PROOF OF PROPOSITION 1

**Proposition 1** (Optimal Mixture Strictly Reduces Variance). *Fix a coordinate $k$ and let $\rho_\beta := w(t, \boldsymbol{z}_t)\pi_{\boldsymbol{\theta}}^{\beta}(\boldsymbol{x}_i \mid \boldsymbol{z}_t, \boldsymbol{c})/\mathbb{E}\left[w(t, \boldsymbol{z}_t)\pi_{\boldsymbol{\theta}}^{\beta}(\boldsymbol{x}_i \mid \boldsymbol{z}_t, \boldsymbol{c})\right]$, where $w(t, \boldsymbol{z}_t) := w(t)\mathbb{1}(z_t = \boldsymbol{m})$. Then, the gradient of mixture objective (8) is given by*

$$g_{\omega,k} = ((1 - \omega)w(t, \boldsymbol{z}_t) + \omega \rho_\beta)\, \partial_{\boldsymbol{\theta}_k} \log \pi_{\boldsymbol{\theta}}(\boldsymbol{x} \mid \boldsymbol{z}_t, \boldsymbol{c}). \tag{25}$$

*If $\mathrm{Var}((\rho_\beta - w(t, \boldsymbol{z}_t))\partial_{\boldsymbol{\theta}_k} \log \pi_{\boldsymbol{\theta}}(\boldsymbol{x} \mid \boldsymbol{z}_t, \boldsymbol{c})) > 0$, then $\mathrm{Var}[g_{\omega,k}]$ is a strictly convex quadratic in $\omega$ and thus admits a unique minimizer $\omega_k^\star$. Moreover,*

$$\mathrm{Var}[g_{\omega_k^\star, k}] \;<\; \min\{\, \mathrm{Var}[g_{0,k}],\; \mathrm{Var}[g_{1,k}]\},$$

*Proof.* We first derive the formulas for the gradient of each objective. Consider a specific example $x_i$. The gradient of the $\mathcal{L}_{\text{ELBO}}$ and $\tilde{\mathcal{L}}_{\text{ELBO}}$ are given by:

$$\nabla_{\boldsymbol{\theta}} \mathcal{L}_{\text{ELBO}} = \mathbb{E}\left[w(t, \boldsymbol{z}_t)\nabla \log \pi_{\boldsymbol{\theta}}(\boldsymbol{x}_i \mid \boldsymbol{z}_t, \boldsymbol{c})\right] \tag{26}$$

$$\nabla_{\boldsymbol{\theta}} \tilde{\mathcal{L}}_{\text{EUBO}} = \frac{\mathbb{E}\left[w(t, \boldsymbol{z}_t)\pi_{\boldsymbol{\theta}}^{\beta}(\boldsymbol{x}_i \mid \boldsymbol{z}_t, \boldsymbol{c})\nabla \log \pi_{\boldsymbol{\theta}}(\boldsymbol{x}_i \mid \boldsymbol{z}_t, \boldsymbol{c})\right]}{\mathbb{E}\left[w(t, \boldsymbol{z}_t)\pi_{\boldsymbol{\theta}}^{\beta}(\boldsymbol{x}_i \mid \boldsymbol{z}_t, \boldsymbol{c})\right]} \tag{27}$$

Then the gradient of the mixture objective $\tilde{\mathcal{L}}_{\text{Mix}}$ is given by:

$$\nabla_{\boldsymbol{\theta}} \tilde{\mathcal{L}}_{\text{Mix}} = \mathbb{E}\left[\left((1 - \omega)w(t, \boldsymbol{z}_t) + \omega \rho_\beta\right)\nabla_{\boldsymbol{\theta}} \log \pi_{\boldsymbol{\theta}}(\boldsymbol{x}_i \mid \boldsymbol{z}_t, \boldsymbol{c})\right] \tag{28}$$

We further compute the per-parameter (per-dimension) variance of the gradient of $\tilde{\mathcal{L}}_{\text{Mix}}$ and consider the optimal mixture coefficient $\omega$ to minimize the variance. For simplicity, we use the following short-hand notation:

$$s_k := \partial_{\boldsymbol{\theta}_k} \log \pi_{\boldsymbol{\theta}}(\boldsymbol{x}_i \mid \boldsymbol{z}_t, \boldsymbol{c})$$

We denote the $k$-th coordinate of the gradient $\nabla_{\boldsymbol{\theta}} \tilde{\mathcal{L}}_{\text{Mix}}$ by $g_{\omega,k}$. Then, the coordinate-wise variance of the gradient is given by

$$\mathrm{Var}\left[g_{\omega,k}\right] = \mathbb{E}\left[\left((1 - \omega)\, w + \omega\, \rho_\beta\right)^2 s_k^2\right] - \left(\mathbb{E}\left[\left((1 - \omega)\, w + \omega\, \rho_\beta\right) s_k\right]\right)^2$$

$$= \mathrm{Var}(ws_k) + 2\omega\, \mathrm{Cov}(ws_k, (\rho_\beta - w)s_k) + \omega^2\, \mathrm{Var}((\rho_\beta - w)s_k)$$

where we used the shorthand $w \equiv w(t, z_t)$. The above expression is quadratic in $\omega$ and we find the optimal $\omega$ by setting the derivative of variance to zero:

$$\frac{\partial}{\partial \omega} \operatorname{Var}\big[g_{\omega,k}\big] = 2 \operatorname{Cov}\big(w\, s_k, \ (\rho_\beta - w)\, s_k\big) + 2\omega \operatorname{Var}\big((\rho_\beta - w)\, s_k\big) = 0$$

$$\Rightarrow \omega_k^\star = -\frac{\operatorname{Cov}\big(w\, s_k, \ (\rho_\beta - w)\, s_k\big)}{\operatorname{Var}\big((\rho_\beta - w)\, s_k\big)}.$$

The above yields a per-coordinate optimal $\omega_k^\star$. Equivalently, we can write $\omega_k^\star$ as follows:

$$\omega_k^\star = \frac{\operatorname{Var}\big(w\, s_k\big) - \operatorname{Cov}\big(w\, s_k, \rho_\beta\, s_k\big)}{\operatorname{Var}\big(w\, s_k\big) + \operatorname{Var}\big(\rho_\beta\, s_k\big) - 2\, \operatorname{Cov}\big(w\, s_k, \rho_\beta\, s_k\big)}$$

Furthermore, $\omega_k^\star$ is a minimizer of coordinate-wise variance in the non-degenerative case with $\operatorname{Var}\big((\rho_\beta - w)\, s_k\big) > 0$, as the variance is strongly convex in $\omega$.

The coordinate-wise variance of gradients in $\mathcal{L}_{\mathrm{ELBO}}$ ($\omega = 0$) and $\tilde{\mathcal{L}}_{\mathrm{ELBO}}$ ($\omega = 1$), and the optimal mixture coefficient $\omega^\star$ are then given by

$$\mathcal{L}_{\mathrm{ELBO}}: \quad \operatorname{Var}\big[g_{0,k}\big] = \operatorname{Var}\big[w\, s_k\big],$$

$$\tilde{\mathcal{L}}_{\mathrm{ELBO}}: \quad \operatorname{Var}\big[g_{1,k}\big] = \operatorname{Var}\big[w\, s_k\big] + 2\, \operatorname{Cov}\big(w\, s_k, \ (\rho_\beta - w)\, s_k\big) + \operatorname{Var}\big((\rho_\beta - w)\, s_k\big),$$

$$\text{Optimal:} \quad \operatorname{Var}\big[g_{\omega_k^\star,k}\big] = \operatorname{Var}\big[w\, s_k\big] - \frac{\Big(\operatorname{Cov}\big(w\, s_k, \ (\rho_\beta - w)\, s_k\big)\Big)^2}{\operatorname{Var}\big((\rho_\beta - w)\, s_k\big)},$$

The difference between the variance of $\mathcal{L}_{\mathrm{ELBO}}$ and $\tilde{\mathcal{L}}_{\mathrm{ELBO}}$ with the optimal mixture coefficient can then be derived as follows:

$$\operatorname{Var}\big[w\, s_k\big] - \operatorname{Var}\big[g_{\omega_k^\star,k}\big] = \frac{\Big(\operatorname{Cov}\big(w\, s_k, \ (\rho_\beta - w)\, s_k\big)\Big)^2}{\operatorname{Var}\big((\rho_\beta - w)\, s_k\big)} \geq 0$$

$$\operatorname{Var}\big[\rho_\beta\, s_k\big] - \operatorname{Var}\big[g_{\omega_k^\star,k}\big] = \frac{\Big(\operatorname{Cov}\big(w\, s_k, \ (\rho_\beta - w)\, s_k\big) + \operatorname{Var}\big((\rho_\beta - w)\, s_k\big)\Big)^2}{\operatorname{Var}\big((\rho_\beta - w)\, s_k\big)} \geq 0$$

$\square$

### E.2  ADDITIONAL COMPARISON BETWEEN THE MIXTURE LOSS AND THE LOWER AND UPPER BOUNDS

**Comparing Mixture with the Lower Bound.** Consider the ratio of the coefficient of score function $\nabla_\theta \log \pi_\theta(x_i \mid z_t, c)$ in the gradient in the case of the mixture objective (i.e., $\nabla_\theta \tilde{\mathcal{L}}_{\mathrm{Mix}}$ in Equation (28)) over using only the lower bound (i.e., $\nabla_\theta \mathcal{L}_{\mathrm{ELBO}}$ in Equation (26)):

$$\frac{w_{\mathrm{Mix}}}{w_{\mathrm{ELBO}}} = \frac{(1-\omega)w(t, z_t) + \omega\rho_\beta}{w(t, z_t)} = (1-\omega) + \omega\, \frac{\pi_\theta^\beta(x_i \mid z_t, c)}{\mathbb{E}\Big[w(t, z_t)\pi_\theta^\beta(x_i \mid z_t, c)\Big]}$$

Treating the expectation over all samples $\mathbb{E}\Big[w(t, z_t)\pi_\theta^\beta(x_i \mid z_t, c)\Big]$ as a constant (since it is averaged), the second term in the above ratio is strictly increasing in $\pi_\theta^\beta(x_i \mid z_t, c)$. This realizes a *confidence-aware weighting*: uncertain tokens with small $\pi_\theta^\beta(x_i \mid z_t, c)$, i.e., those with a low recovery chance, have a smaller weight, while confident tokens with large $\pi_\theta^\beta(x_i \mid z_t, c)$ are upweighted, with sharpness being controlled by parameter $\beta$ and the blend by $\omega$.

**Comparing Mixture with the Upper Bound.** We compute the ratio of coefficient of score function in the gradient of upper bound (i.e., $\nabla_\theta \tilde{\mathcal{L}}_{\mathrm{EUBO}}$ in Equation (27)) over the mixture gradient:

$$\frac{w_{\mathrm{EUBO}}}{w_{\mathrm{Mix}}} = \frac{\omega\rho_\beta}{(1-\omega)w(t, z_t) + \omega\rho_\beta}$$

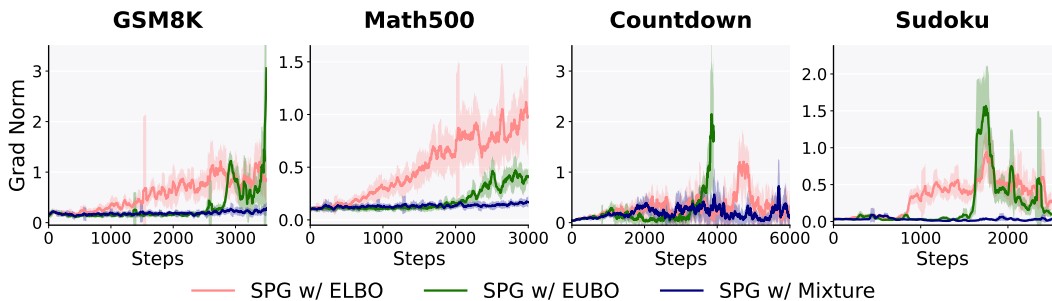

Figure 7: Dynamics of the gradient norm of models trained with different log-likelihood estimation methods. SPG w/ Mixture achieves lower gradient norm and more stable optimization. We report mean and standard deviation over a rolling window of 50 steps.

Considering the above ratio, when $\pi_{\boldsymbol{\theta}}^{\beta}(\boldsymbol{x}_i \mid \boldsymbol{z}_t, \boldsymbol{c})$ is very small, the coefficient of score function in $\nabla_{\boldsymbol{\theta}} \tilde{\mathcal{L}}_{\text{EUBO}}$, $w_{\text{EUBO}}$, becomes very small, preventing updates to the parameters. However, the mixing approach maintains per-sample weights by preventing that from collapsing to (near) zero. In other words, for each sample, the mixture coefficient computes a convex interpolation that simultaneously floors very small EUBO weights to a minimum value and applies an uncertainty-aware capping to large EUBO weights.

**Empirical Evidence of Reduced Gradient Variance.** As a practical indicator of gradient variance, we plot the gradient norm of each model trained with different log-likelihood estimation methods for negative advantage traces in Figure 7. When using the mixture objective, the model has consistently smaller and more stable gradient norm throughout training, aligning well with our theoretical analysis.

### E.3 TOY EXAMPLE FOR UPPER AND LOWER BOUNDS.

In this section, we provide a toy example highlighting the contrasting behaviors and landscapes of the upper and lower bounds, further demonstrating the necessity to select the appropriate bound for optimization based on the optimization direction.

Consider a simple case where the sequence length is 2 and the vocabulary size is 2, i.e., $\boldsymbol{x} = [\boldsymbol{x}_1, \boldsymbol{x}_2]$ and $\mathcal{V} = \{A, B\}$. Then, We can calculate $\mathcal{L}_{\text{ELBO}}$ and $\tilde{\mathcal{L}}_{\text{EUBO}}$ in closed form:

$$\mathcal{L}_{\text{ELBO}}(\boldsymbol{x} = \text{AA}) = \frac{1}{2}\Big[ \log \pi_{\boldsymbol{\theta}}(\boldsymbol{x}_1 = \text{A} \mid \text{MA}) + \log \pi_{\boldsymbol{\theta}}(\boldsymbol{x}_1 = \text{A} \mid \text{MM}) \tag{29}$$

$$+ \log \pi_{\boldsymbol{\theta}}(\boldsymbol{x}_2 = \text{A} \mid \text{AM}) + \log \pi_{\boldsymbol{\theta}}(\boldsymbol{x}_2 = \text{A} \mid \text{MM})\Big] \tag{30}$$

$$\tilde{\mathcal{L}}_{\text{EUBO}}(\boldsymbol{x} = \text{AA}) = \frac{1}{\beta} \log \left(\frac{\pi_{\boldsymbol{\theta}}^{\beta}(\boldsymbol{x}_1 = \text{A} \mid \text{MA}) + \pi_{\boldsymbol{\theta}}^{\beta}(\boldsymbol{x}_1 = \text{A} \mid \text{MM})}{2}\right) \tag{31}$$

$$+ \frac{1}{\beta} \log \left(\frac{\pi_{\boldsymbol{\theta}}^{\beta}(\boldsymbol{x}_2 = \text{A} \mid \text{AM}) + \pi_{\boldsymbol{\theta}}^{\beta}(\boldsymbol{x}_2 = \text{A} \mid \text{MM})}{2}\right) \tag{32}$$

For simplicity, denote $a := \pi_{\boldsymbol{\theta}}(\boldsymbol{x}_1 = \text{A} \mid \text{MA})$ and $b := \pi_{\boldsymbol{\theta}}(\boldsymbol{x}_1 = \text{A} \mid \text{MM})$, and consider the of the likelihood of the first token $\boldsymbol{x}_1$. We have

$$\mathcal{L}_{\text{ELBO}}(\boldsymbol{x}_1) = \frac{1}{2}(\log a + \log b) \tag{33}$$

$$\tilde{\mathcal{L}}_{\text{EUBO}}(\boldsymbol{x}_1) = \frac{1}{\beta} \log \left(\frac{a^{\beta} + b^{\beta}}{2}\right) \tag{34}$$

Take the partial gradient with respect to $a$ and $b$ respectively,

$$\frac{\partial \mathcal{L}_{\text{ELBO}}(\boldsymbol{x}_1)}{\partial a} = \frac{1}{2a}; \quad \frac{\partial \mathcal{L}_{\text{ELBO}}(\boldsymbol{x}_1)}{\partial b} = \frac{1}{2b} \tag{35}$$

$$\frac{\partial \tilde{\mathcal{L}}_{\text{EUBO}}(\boldsymbol{x}_1)}{\partial a} = \frac{a^{\beta-1}}{a^{\beta} + b^{\beta}}; \quad \frac{\partial \tilde{\mathcal{L}}_{\text{EUBO}}(\boldsymbol{x}_1)}{\partial b} = \frac{b^{\beta-1}}{a^{\beta} + b^{\beta}} \tag{36}$$

Therefore, for $\tilde{\mathcal{L}}_{\text{EUBO}}$, the gradient direction is dominated by the larger one between $a$ and $b$, while for $\mathcal{L}_{\text{ELBO}}$, the gradient direction is dominated by the smaller one. Such property is illustrated in the landscapes of $-\mathcal{L}_{\text{ELBO}}$ and $-\tilde{\mathcal{L}}_{\text{EUBO}}$ for $a, b \in (0, 1)$ in Figure 8.

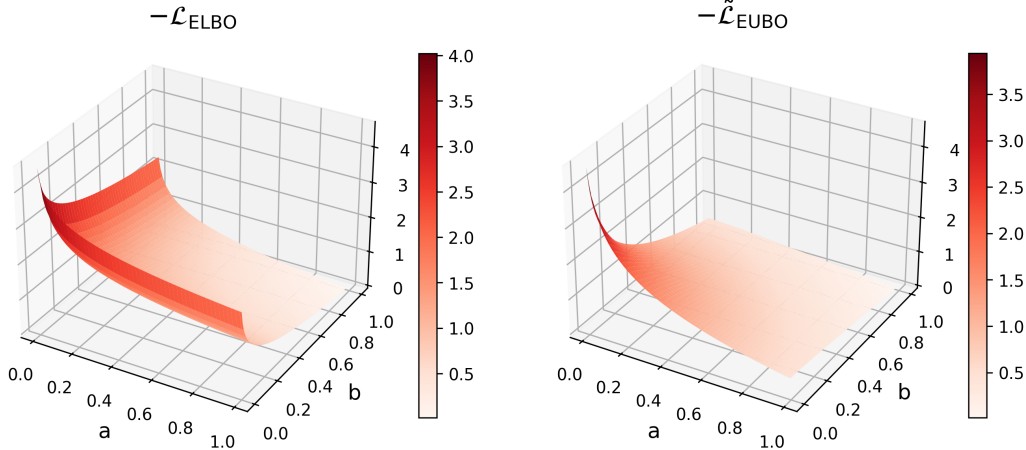

Figure 8: Landscapes of $-\mathcal{L}_{\text{ELBO}}$ and $-\tilde{\mathcal{L}}_{\text{EUBO}}$ for $0 < a, b < 1$. $-\tilde{\mathcal{L}}_{\text{EUBO}}$ is flatter among low value regions while sharper among high value regions, making it more suitable for log-likelihood minimization; vice versa for $-\mathcal{L}_{\text{ELBO}}$.

When $\boldsymbol{x} = $ AA has negative advantage, the corresponding $\mathcal{L}_{\text{ELBO}}$ and $\tilde{\mathcal{L}}_{\text{EUBO}}$ are minimized. For $\mathcal{L}_{\text{ELBO}}$, the model benefits more from further decreasing the smaller one between probabilities $a$ and $b$. In the extreme case, $\mathcal{L}_{\text{ELBO}} = -\infty$ when either $a$ or $b$ equals to zero, leaving the other term not sufficiently decreased. Instead, when using $\tilde{\mathcal{L}}_{\text{EUBO}}$ for negative advantage traces, the larger one between $a$ and $b$ is preferentially minimized, leading to a more balanced optimization that stably decreases the log-likelihood.

Similarly, when $\boldsymbol{x} = $ AA has positive advantage, the corresponding $\mathcal{L}_{\text{ELBO}}$ and $\tilde{\mathcal{L}}_{\text{EUBO}}$ are maximized. Using $\mathcal{L}_{\text{ELBO}}$ enables effectively increasing the smaller likelihood, while $\tilde{\mathcal{L}}_{\text{EUBO}}$ focuses on the larger one, leading to a less efficient optimization.

## F  Additional Discussions and Future Directions on the RL Algorithm

**RL Stabilization.**  In SPG, we employ a simple policy gradient objective. However, as proposed in PPO (Schulman et al., 2017) and GRPO (Shao et al., 2024), various components including the importance sampling, clipping, and KL regularization are applied to resolve issues like off-policyness and stabilize the training. In principle, we should be able to adopt these techniques in SPG. For example, the same heuristic trick as in D1or UniGRPO can be applied, using a one-step estimation or the ELBO as a proxy for log-likelihood in the importance sampling ratio and KL divergence terms. However, it remains unclear what is a principally correct and elegant way to incorporate these techniques to dLLMs, especially with the intractable log-likelihood challenge.

Empirically, we found our current SPG algorithms already performant. Therefore, we chose to keep the current SPG in its simplest form. We leave correctly adapting RL stabilization to dLLMs as an interesting future direction, such as how to use KL and trust-region optimization correctly for dLLMs.

Additionally, calculating group-relative advantages can introduce biases. These biases can be addressed by employing the leave-one-out empirical expectation of the reward, known as RLOO (Ahmadian et al., 2024; Kool et al., 2019). Notably, RLOO remains unbiased even in off-policy settings (Flet-Berliac et al., 2024), eliminating the need for importance sampling. Further reducing biases in the policy update objective for dLLMs using RLOO or other methods is an interesting direction for future work.

**The Reward Shifting Alternate.** We note that in episodic RL with guaranteed termination and end-point reward only, or when KL is applied (the maximal entropy RL framework), optimal policy is invariant to reward shifting. This suggests an alternative approach to the log-likelihood estimation problem in dLLMs: simply shift reward values by a constant so they are always positive, and then apply the ELBO across all traces. However, this method breaks down when partial rewards are present (such as with long CoT traces), or in infinite horizon settings (e.g., in online learning), where reward shifting is no longer applicable. More importantly, even if reward shifting were possible, it could alter the learning dynamics. Prior work in RL for language models shows that RL can reduce forgetting in pretrained models and enhance exploration compared to SFT-only approaches (while SFT could be viewed as assigning 0 reward to negative traces and 1 to positive ones). In fact, decreasing the log-likelihood of negative traces increases entropy, thereby improving exploration and reducing overfitting. Such benefit is missing when we conduct the reward shifting. Therefore, we believe our SPG method remains advantageous.

Empirically, we experiment with a reward-shifted objective by normalizing each trace's advantage using the minimum value within the batch, applying both REINFORCE and GRPO and estimating the log-likelihood with the ELBO. The results are shown in Table 7. Under both settings, the reward shifting method underperforms SPG and most other baselines.

Table 7: Performance of the reward shifting method on MATH500 and Countdown. SPG outperforms the reward shifting with both REINFORCE and GRPO objectives.

| | MATH500 (0-shot) | | | | Countdown (0-shot) | | | |
|---|---|---|---|---|---|---|---|---|
| Model / Seq Len | 128 | 256 | 512 | *Avg.* | 128 | 256 | 512 | *Avg.* |
| REINFORCE + Reward Shift | 28.0 | 34.4 | 39.0 | 33.8 | 13.3 | 25.0 | 37.9 | 25.4 |
| GRPO + Reward Shift | 24.4 | 33.4 | 36.2 | 31.3 | 23.1 | 23.8 | 48.1 | 31.6 |
| SPG w/ EUBO | 33.2 | 37.6 | 39.4 | 36.7 | 68.4 | 71.5 | 68.0 | 69.3 |
| SPG w/ Mixture | 33.4 | 40.0 | 41.8 | 38.4 | 68.8 | 70.7 | 70.3 | 69.9 |

# G    ADDITIONAL EXPERIMENTAL DETAILS

## G.1    DATASETS AND REWARD FUNCTIONS

We follow the setting in D1 (Zhao et al., 2025) and WD1 (Tang et al., 2025), using the same reward functions and train-test splitting, except for Sudoku. The rewards are designed to encourage both correctness and proper formatting, with varying levels of granularity tailored for each task. For completeness, we provide details as follows.

**GSM8K.** We utilize the train split of the GSM8K dataset[3] for RL training, and evaluate model performance on the test split. We follow the Unsloth reward setup[4], utilizing five equally-weighted additive components:

- XML Structure Reward: +0.125 per correct formatting tag; small penalties for extra contents after the closing tag.
- Soft Format Reward: +0.5 for outputs matching the pattern:
  `<reasoning>...</reasoning><answer>...</answer>`
- Strict Format Reward: +0.5 for exact formatting with correct line breaks.

---

[3] `https://huggingface.co/datasets/openai/gsm8k`
[4] `https://unsloth.ai/blog/r1-reasoning`

- Integer Answer Reward: +0.5 if the answer is a valid integer.
- Correctness Reward: +2.0 of the answer matches the ground truth.

**MATH500.** We utilize the train split of the MATH dataset[5] for RL training, and evaluate model performance on the test split. We use a format reward and a correctness reward:

- Format Reward: We award 1.00 if <answer></answer> tags are present with \boxed inside them; 0.75 if answer tags are present without \boxed; 0.50 if answer tags are not present but \boxed is present; 0.25 if neither the answer tags nor \boxed is present.
- Correctness Reward: We award 2.00 if the answer in `\boxed{}` matches the ground truth.

**Countdown.** We utilize the train split of the Countdown dataset[6] for RL training, restricting to instances that use only three numbers. We evaluate on the same set of 256 synthetically generated countdown questions with 3 numbers as in D1 (Zhao et al., 2025). The reward covers three cases: +1.0 if the expression reaches the target using the exact numbers; +0.1 if the numbers are correct but does not reach the target; +0.0 otherwise.

**Sudoku.** We experiment on the $4\times4$ Sudoku dataset[7] generated by Arel (2025). The original training split contains 1M unique Sudoku puzzles covering all 288 $4\times4$ Soduku solutions. To avoid train-test leakage and potential cheating by memorizing all the solutions, we randomly select 200 solutions and include all puzzles corresponding to these solutions into the new training set, resulting in 694,006 training puzzles. We then randomly select 2 or 3 puzzles corresponding to the left 88 solutions to construct the test set, which has 256 Soduku puzzles in total.

We observe that the zero-shot setting is too difficult for the base LLaDA-8B-Instruct model, which has test accuracy below 7% with a generation length of 256 and struggles to correctly interpret the questions, leading to very few meaningful RL rollouts. Therefore, we instead use 3-shot for all the Sudoku experiments. We ensure that the solutions presented in the 3-shot samples do not appear in test set solutions, and the puzzles do not appear in both train and test set. The detailed few-shot samples are provided in Appendix G.3.

**Coding.** We fine-tune the base model on the KodCode-Light-RL-10k[8] dataset. We use a structure reward and a correctness reward, together with a safe code guarantee:

- XML Structure Reward: the same function for GSM8K is used, with an additional 0.5 points if the program is within answer tags. If the code block is not wrapped in ` ```python``` `, 0 points are given.
- Correctness Reward: we use unit tests to verify the correctness of the code, and use the fraction of unit tests passed as the reward.
- Safe Code: we assign 0 reward if any blocked modules are used, including `os`, `sys`, `shutil`, `subprocess`, `socket`, `psutil`, `ctypes`, `pathlib`, `builtins`, and `__import__`.

### G.2 Hyperparameter Settings and Implementation Details

We follow D1 (Zhao et al., 2025) for most hyperparameter settings. We employ Low-Rank Adaptation (LoRA) with a rank of $r = 128$ and scaling factor $\alpha = 64$. The training was conducted on 8 NVIDIA A100-80G or NVIDIA H100-80G GPU, with the following hyperparameters: batch size of 6 per GPU, and gradient accumulation steps of 2. We set the number of inner gradient update $\mu$ as 4 for all models. We use the AdamW optimizer (Loshchilov & Hutter, 2017), with $\beta_1 = 0.9$, $\beta_2 = 0.99$, weight decay of 0.1, learning rate of $3 \times 10^{-6}$, and gradient clipping at 0.2. We utilize Flash Attention 2 (Dao, 2023)and 4-bit quantization.

---

[5] `https://huggingface.co/datasets/ankner/math-500`
[6] `https://huggingface.co/datasets/Jiayi-Pan/Countdown-Tasks-3to4`
[7] `https://github.com/Black-Phoenix/4x4-Sudoku-Dataset`
[8] `https://huggingface.co/datasets/KodCode/KodCode-Light-RL-10K`

For RL rollout, we use sequence length of 256 tokens, and 128 diffusion steps. We employ confidence-based semi-autoregressive generation with block size 32, and set the temperature as 0.9 (except for Sudoku where temperature is set as 0.3 following D1). We set number of completions per prompt $g$ as 6, and number of Monte Carlo estimation samples $m$ as 2 due to computational constraint. Since the rollout stage dominates the training time, the average time per gradient update step for SPG is similar to that of the other baselines.

We train 6000 steps (i.e., number of gradient updates) for GSM8K, Countdown, and coding, 4000 steps for MATH500, and 2500 steps for Sudoku. For all RL models, we run evaluation every 100 steps with generation sequence length 128, 256, 512, and report the result of the checkpoint with the highest average accuracy over the three generation lengths, except for the ablations on inference strategies in Table 17, where we only evaluate on generation length 256.

### G.3 Few-Shot Samples for Sudoku

**Sample 1:**

Question:
Solve the following Sudoku puzzle: 3014002020004130
Answer:
<reasoning>
Interpret puzzle as 4 rows of 4:
R1: 3 0 1 4
R2: 0 0 2 0
R3: 2 0 0 0
R4: 4 1 3 0

Fill easy singles:
R1 missing 2 → R1C2=2.
R4 missing 2 → R4C4=2.
Box D (R3-4,C3-4) then needs 1,4; column4 can only accept 1 → R3C4=1, R3C3=4.
R3 now missing 3 → R3C2=3.
Column1 missing 1 → R2C1=1.
Column2 missing 4 → R2C2=4.
Last cell R2C4=3.

Final grid:
R1: 3 2 1 4
R2: 1 4 2 3
R3: 2 3 4 1
R4: 4 1 3 2
</reasoning>
<answer>
3214142323414132
</answer>

**Sample 2:**

Question:
Solve the following Sudoku puzzle: 0000100420013142
Answer:
<reasoning>
Interpret puzzle as 4 rows of 4:
R1: 0 0 0 0
R2: 1 0 0 4
R3: 2 0 0 1
R4: 3 1 4 2

Fill easy singles:
Col1 missing 4 → R1C1=4.

Col4 missing 3 → R1C4=3.
Box A (R1-2,C1-2) missing 2,3 and R1 now needs 1,2 → R1C2=2, R2C2=3.
R1C3=1.
R2 now missing 2 → R2C3=2.
Col2 missing 4 → R3C2=4, then R3C3=3.

Final grid:
R1: 4 2 1 3
R2: 1 3 2 4
R3: 2 4 3 1
R4: 3 1 4 2
</reasoning>
<answer>
4213132424313142
</answer>

**Sample 3:**

Question:
Solve the following Sudoku puzzle: 2001403002001420
Answer:
<reasoning>
Interpret puzzle as 4 rows of 4:
R1: 2 0 0 1
R2: 4 0 3 0
R3: 0 2 0 0
R4: 1 4 2 0

Fill easy singles:
R1 missing 3,4; Col2 can't be 1 so R1C2=3 → R1C3=4.
R4 missing 3 → R4C4=3.
Col4 missing 2,4; R2 must take 2 → R2C4=2 → R2C2=1.
Col1 missing 3 → R3C1=3.
Col3 missing 1 → R3C3=1 → R3C4=4.

Final grid:
R1: 2 3 4 1
R2: 4 1 3 2
R3: 3 2 1 4
R4: 1 4 2 3
</reasoning>
<answer>
2341413232141423
</answer>

# H   ADDITIONAL RESULTS

## H.1   ADDITIONAL EVALUATIONS TO THE MAIN RESULTS

**Complete evaluation results.** We provide the complete evaluation results, along with those reported in D1 (Zhao et al., 2025) and WD1 (Tang et al., 2025), in Table 8. Our reproduced numbers closely match the reported results. d1-LLaDA (Zhao et al., 2025) denotes the model that conducts first SFT and then RL (using D1). All other models are trained solely with RL. In D1 and d1-LLaDA, the best result for each generation length is reported separately, whereas we select a single checkpoint with the highest average accuracy across all three generation lengths, leading to slightly worse results than the reported numbers. The reported results in WD1 are based on evaluations on fewer checkpoints, so they are generally a bit lower than our reproduced values.

Table 8: Complete model performance on four reasoning benchmarks compared with baselines. We provide both the reported and the reproduced results for D1 and WD1. The best results are bolded and the second best are underlined. SPG consistently outperforms all other models.

| Model / Seq Len | GSM8K (0-shot) | | | MATH500 (0-shot) | | | Countdown (0-shot) | | | Sudoku (3-shot) | | |
|---|---|---|---|---|---|---|---|---|---|---|---|---|
| | 128 | 256 | 512 | 128 | 256 | 512 | 128 | 256 | 512 | 128 | 256 | 512 |
| LLaDA-8B-Instruct | 69.5 | 77.2 | 79.8 | 28.2 | 32.4 | 34.6 | 18.8 | 16.8 | 16.8 | 5.7 | 27.7 | 26.2 |
| LLaDA-1.5 | 70.4 | 80.5 | 81.9 | 26.8 | 32.2 | 35.8 | 21.9 | 21.1 | 21.5 | 7.4 | 26.9 | 29.0 |
| D1 (reported) | 72.6 | 79.8 | 81.9 | 33.2 | 37.2 | 39.2 | 33.2 | 31.3 | 37.1 | - | - | - |
| D1 (reproduced) | 72.2 | 80.6 | 81.3 | 31.4 | 36.0 | 39.4 | 30.9 | 30.9 | 34.4 | 7.2 | 32.5 | 29.3 |
| d1-LLaDA (reported) | 73.2 | 81.1 | 82.1 | **33.8** | 38.6 | 40.2 | 34.8 | 32.0 | 42.2 | - | - | - |
| WD1 (reported) | - | 80.8 | 82.3 | - | 34.4 | 39.0 | - | 51.2 | 46.1 | - | - | - |
| WD1 (reproduced) | 74.6 | 81.5 | 83.0 | 31.0 | 37.4 | 39.0 | 48.8 | 52.3 | 50.8 | 33.1 | 32.1 | 22.5 |
| UniGRPO | 74.9 | 82.5 | 82.7 | 32.4 | 37.4 | 39.4 | 44.5 | 43.0 | 57.0 | 59.0 | 67.0 | 62.9 |
| SPG w/ EUBO (ours) | 77.1 | 83.8 | 83.9 | 33.2 | 37.6 | 39.4 | 68.4 | **71.5** | 68.0 | 81.2 | 87.1 | 89.9 |
| SPG w/ mixture (ours) | **78.5** | **86.1** | **84.5** | 33.4 | **40.0** | **41.8** | **68.8** | 70.7 | 70.3 | **82.9** | **94.0** | **93.1** |

**Dynamics of Completion Length.** We provide the dynamics of the effective sequence length of SPG during RL training in Figure 9. We also report the effective length of the best checkpoint in Table 9. SPG leads to effective usage of the total given length and good adaptation to task difficulties.

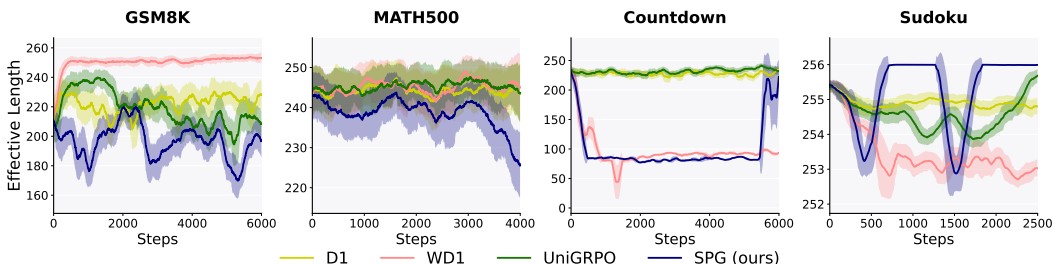

Figure 9: Dynamics of the effective generation length of SPG during RL training, compared with D1, WD1, and UniGRPO. SPG leads to concise solutions with better token efficiency. We report mean and standard deviation over a rolling window of 50 steps.

Table 9: Effective sequence length of each model at the best checkpoint corresponding to Table 1 on four reasoning benchmarks.

| Model / Seq Len | GSM8K (0-shot) | | | MATH500 (0-shot) | | | Countdown (0-shot) | | | Sudoku (3-shot) | | |
|---|---|---|---|---|---|---|---|---|---|---|---|---|
| | 128 | 256 | 512 | 128 | 256 | 512 | 128 | 256 | 512 | 128 | 256 | 512 |
| LLaDA-8B-Instruct | 114 | 212 | 257 | 123 | 235 | 402 | 111 | 213 | 407 | 111 | 232 | 448 |
| LLaDA-1.5 | 115 | 214 | 265 | 123 | 237 | 407 | 114 | 215 | 411 | 112 | 232 | 419 |
| D1 | 115 | 209 | 261 | 123 | 234 | 399 | 107 | 211 | 397 | 111 | 231 | 449 |
| WD1 | 115 | 225 | 312 | 123 | 231 | 378 | 83 | 84 | 90 | 105 | 227 | 473 |
| UniGRPO | 114 | 211 | 257 | 123 | 235 | 400 | 100 | 207 | 374 | 113 | 230 | 472 |
| SPG w/ EUBO | 110 | 196 | 227 | 120 | 228 | 382 | 68 | 70 | 78 | 89 | 137 | 249 |
| SPG w/ mixture | 108 | 176 | 195 | 121 | 229 | 384 | 75 | 78 | 79 | 115 | 239 | 491 |

## H.2 ADDITIONAL ABLATION RESULTS

In this section, we provide the complete results for each generation length and task in supplement to Section 4.2. We also include additional ablation studies on the looser upper bound, different log-likelihood estimation methods for positive advantage traces, and Pass@K performance.

**Ablations on Algorithm Components.** We provide the complete results for ablations on log-likelihood estimation methods in Table 10 and for ablations on masking strategies in Table 11.

Table 10: Ablations on log-likelihood estimation methods for negative advantage traces. The best results are bolded and the second best are underlined. SPG w/ Mixture consistently outperforms other likelihood estimation methods.

| Model | GSM8K (0-shot) | | | | MATH500 (0-shot) | | | | Countdown (0-shot) | | | | Sudoku (3-shot) | | | |
|---|---|---|---|---|---|---|---|---|---|---|---|---|---|---|---|---|
| | 128 | 256 | 512 | *Avg.* | 128 | 256 | 512 | *Avg.* | 128 | 256 | 512 | *Avg.* | 128 | 256 | 512 | *Avg.* |
| SPG wo/ neg | 72.0 | 79.0 | 81.3 | 77.4 | 28.2 | 32.2 | 37.8 | 32.7 | 43.8 | 48.1 | 44.5 | 45.5 | 55.0 | 82.9 | 68.4 | 68.8 |
| SPG w/ ELBO | 75.6 | 82.8 | 84.4 | 80.9 | 35.8 | 37.6 | 38.8 | 37.4 | 66.8 | 66.0 | 68.4 | 67.1 | 73.8 | 89.4 | 84.1 | 82.4 |
| SPG w/ EUBO | 77.1 | 83.8 | 83.9 | 81.6 | 33.2 | 37.6 | 39.4 | 36.7 | 68.4 | 71.5 | 68.0 | 69.3 | 81.2 | 87.1 | 89.9 | 86.1 |
| SPG w/ Mixture | 78.5 | 86.1 | 84.5 | 83.0 | 33.4 | 40.0 | 41.8 | 38.4 | 68.8 | 70.7 | 70.3 | 69.9 | 82.9 | 94.0 | 93.1 | 90.0 |

Table 11: Ablations on the masking strategies in Monte Carlo estimation. Our block-wise masking strategy leads to consistent improvement to random masking on both benchmarks.

| Model | Masking | MATH500 (0-shot) | | | | Countdown (0-shot) | | | |
|---|---|---|---|---|---|---|---|---|---|
| | | 128 | 256 | 512 | *Avg.* | 128 | 256 | 512 | *Avg.* |
| SPG w/ EUBO | random | 33.4 | 35.4 | 41.4 | 36.7 | 42.6 | 41.0 | 52.7 | 45.4 |
| | block-wise | 33.2 | 37.6 | 39.4 | 36.7 | 68.4 | 71.5 | 68.0 | 69.3 |
| SPG w/ Mixture | random | 33.8 | 38.2 | 38.8 | 36.9 | 52.3 | 64.5 | 71.5 | 62.8 |
| | block-wise | 33.4 | 40.0 | 41.8 | 38.4 | 68.8 | 70.7 | 70.3 | 69.9 |

**Ablations on Key Hyperparameters $\beta$ and $\omega$.** We provide the complete results for ablations on $\beta$ in Table 12 and for ablations on $\omega$ in Table 13.

Table 12: Ablations on the value of $\beta$ in the upper bound.

| Model | $\beta$ | GSM8K (0-shot) | | | | MATH500 (0-shot) | | | | Countdown (0-shot) | | | | Sudoku (3-shot) | | | |
|---|---|---|---|---|---|---|---|---|---|---|---|---|---|---|---|---|---|
| | | 128 | 256 | 512 | *Avg.* | 128 | 256 | 512 | *Avg.* | 128 | 256 | 512 | *Avg.* | 128 | 256 | 512 | *Avg.* |
| | 0.50 | 77.7 | 83.2 | 84.5 | 81.8 | 32.8 | 36.4 | 41.2 | 36.8 | 71.1 | 68.8 | 74.6 | 71.5 | 64.7 | 53.4 | 57.4 | 58.5 |
| | 0.75 | 77.2 | 83.9 | 84.5 | 81.9 | 31.0 | 36.6 | 40.0 | 35.9 | 70.7 | 70.7 | 70.7 | 70.7 | 63.4 | 65.7 | 45.4 | 58.2 |
| SPG w/ EUBO | 1.00 | 76.5 | 83.9 | 83.6 | 81.3 | 31.0 | 37.4 | 38.8 | 35.7 | 66.0 | 66.8 | 66.4 | 66.4 | 81.2 | 87.1 | 89.9 | 86.1 |
| | 1.50 | 77.1 | 83.8 | 83.9 | 81.6 | 33.2 | 37.6 | 39.4 | 36.7 | 69.5 | 64.5 | 66.4 | 66.8 | 32.7 | 40.5 | 39.9 | 37.7 |
| | 2.00 | 76.5 | 83.9 | 83.2 | 81.2 | 32.4 | 36.8 | 38.2 | 35.8 | 68.4 | 71.5 | 68.0 | 69.3 | 28.1 | 31.9 | 28.0 | 29.3 |
| | 1.00 | 78.8 | 85.6 | 84.9 | 83.1 | 34.0 | 40.2 | 39.2 | 37.8 | 69.9 | 69.5 | 70.3 | 69.9 | 82.9 | 94.0 | 93.1 | 90.0 |
| SPG w/ Mixture | 1.50 | 78.5 | 86.1 | 84.5 | 83.1 | 33.4 | 40.0 | 41.8 | 38.4 | 68.8 | 70.7 | 70.3 | 69.9 | 83.2 | 86.0 | 84.6 | 84.6 |
| | 2.00 | 78.8 | 85.7 | 84.7 | 83.1 | 32.4 | 38.8 | 39.8 | 37.0 | 70.3 | 69.1 | 69.5 | 69.6 | 44.3 | 60.5 | 60.7 | 55.2 |

Table 13: Ablations on the mixture coefficient $\omega$ on MATH500 and Countdown.

| SPG w/ Mixture | MATH500 (0-shot) | | | | Countdown (0-shot) | | | |
|---|---|---|---|---|---|---|---|---|
| $\omega$ | 128 | 256 | 512 | *Avg.* | 128 | 256 | 512 | *Avg.* |
| 0.00 | 35.8 | 37.6 | 38.8 | 37.4 | 66.8 | 66.0 | 68.4 | 67.1 |
| 0.25 | 34.6 | 37.6 | 42.2 | 38.1 | 71.5 | 68.0 | 67.2 | 68.9 |
| 0.50 | 33.4 | 40.0 | 41.8 | 38.4 | 68.8 | 70.7 | 70.3 | 69.9 |
| 0.75 | 34.2 | 38.6 | 41.2 | 38.0 | 69.5 | 69.1 | 74.2 | 70.9 |
| 1.00 | 33.2 | 37.6 | 39.4 | 36.7 | 69.5 | 64.5 | 66.4 | 66.8 |

**Ablations on Inference Strategies.** We provide complete results for ablations on different inference strategies in Table 17. Note that the reported numbers of each method for "Semi-AR, Confidence, Block=32" is in general slightly higher than the results in Table 1 under the same inference setting. This is because in Table 17, we select the checkpoint with the highest accuracy specifically for generation length 256 to maintain consistency with other inference settings, while in Table 1, we choose the checkpoint with the highest average accuracy across generation lengths 128, 256, and 512.

**Ablations on the Looser Upper Bound.** As mentioned in Section 3.2 and Appendix D, a looser but unbiased bound can be derived using inequalities like $\log(x) \leq x - 1$, i.e., $\tilde{\mathcal{L}}_{\text{Loose}}$ (Equation (24)). However, as shown in Table 14, this looser bound performs worse empirically than the tighter upper bound $\tilde{\mathcal{L}}_{\text{EUBO}}$ we used, possibly due to a larger discrepancy from the true log-likelihood.

Table 14: Ablations on the looser upper bound. The loose bound performs worse than the tighter upper bound we used, indicating inferior performance due to a larger discrepancy from the true log-likelihood.

| SPG w/ EUBO | | MATH500 (0-shot) | | | | Countdown (0-shot) | | | |
|---|---|---|---|---|---|---|---|---|---|
| $\beta$ | Upper Bound | 128 | 256 | 512 | *Avg.* | 128 | 256 | 512 | *Avg.* |
| 1.0 | $\tilde{\mathcal{L}}_{\text{Loose}}$ | 29.4 | 35.4 | 39.4 | 34.7 | 43.8 | 65.2 | 64.8 | 57.9 |
| | $\tilde{\mathcal{L}}_{\text{EUBO}}$ | 31.0 | 37.4 | 38.8 | **35.7** | 66.0 | 66.8 | 66.4 | **66.4** |
| 1.5 | $\tilde{\mathcal{L}}_{\text{Loose}}$ | 29.8 | 31.8 | 38.8 | 33.5 | 46.9 | 54.7 | 57.0 | 52.9 |
| | $\tilde{\mathcal{L}}_{\text{EUBO}}$ | 33.2 | 37.6 | 39.4 | **36.7** | 69.5 | 64.5 | 66.4 | **66.8** |

Table 15: Ablations on log-likelihood estimation for positive advantage traces. Using the upper bound for log-likelihood estimation of positive advantage traces perform worse than using the lower bound.

| | Positive traces | MATH500 (0-shot) | | | | Countdown (0-shot) | | | |
|---|---|---|---|---|---|---|---|---|---|
| Model | likelihood estimation | 128 | 256 | 512 | *Avg.* | 128 | 256 | 512 | *Avg.* |
| SPG w/ EUBO | $\tilde{\mathcal{L}}_{\text{EUBO}}$ ($\beta = 1.0$) | 34.4 | 36.2 | 39.2 | 36.6 | 48.1 | 46.7 | 50.8 | 48.5 |
| | $\mathcal{L}_{\text{ELBO}}$ | 33.2 | 37.6 | 39.4 | **36.7** | 68.4 | 71.5 | 68.0 | **69.3** |
| SPG w/ Mixture | $\tilde{\mathcal{L}}_{\text{Mix}}$ ($\beta = 1.0, \omega = 0.5$) | 35.4 | 38.4 | 39.0 | 37.6 | 69.1 | 68.4 | 70.3 | 69.3 |
| | $\mathcal{L}_{\text{ELBO}}$ | 33.4 | 40.0 | 41.8 | **38.4** | 68.8 | 70.7 | 70.3 | **69.9** |

**Ablations on Log-Likelihood Estimations for Positive Advantage Traces.** Instead of always using $\mathcal{L}_{\text{ELBO}}$ for positive advantage traces, we experiment on MATH500 and Countdown benchmarks using both $\tilde{\mathcal{L}}_{\text{EUBO}}$ and $\tilde{\mathcal{L}}_{\text{Mix}}$ for positive advantage traces. Correspondingly, we use $\omega = 0.5$ and the best performed $\beta$ as previously discussed for negative advantage traces. For the positive advantage traces, we always use the tightest $\beta = 1.0$ for both $\tilde{\mathcal{L}}_{\text{EUBO}}$ and $\tilde{\mathcal{L}}_{\text{Mix}}$. The results are shown in Table 15, indicating that using the upper bound for likelihood estimation of positive advantage traces performs worse than using $\mathcal{L}_{\text{ELBO}}$. This aligns well with our theoretical insights that the lower bound is a better objective for log-likelihood maximization.

**Ablations on Pass@K Performance.** In all previous experiments, we apply greedy sampling by setting temperature as 0.0 following D1 and LLaDA. However, beyond accuracy, it is essential for models to generate a diverse set of outputs that can cover the correct solution and allow for explorations. In this section, we investigate the models' ability to generate diverse outputs using a higher temperature, and evaluate their Pass@K performance on MATH500 and Countdown, as shown in Table 16. Specifically, we set temperature to 0.9 and generation length to 256, conduct evaluations every 100 steps, and report results from the checkpoint with the highest accuracy. For comparison, we also include results from greedy sampling, denoted as Pass@1$_{\text{Greedy}}$. As expected, increasing the temperature leads to a decrease in Pass@1 performance across all models, aligning with observations from previous work. For K>1, the Pass@K scores improve for all models as K increases from 1 to 4. SPG achieves the best performance across all settings, with SPG w/ Mixture reaching 55.6% Pass@4 accuracy on MATH500 and 76.6% on Countdown, demonstrating the ability of SPG to generate diverse outputs that can recover the correct solution.

**Ablations on Full Fine-Tuning.** In our main experiments, we follow the setting of D1 and WD1, using LoRA for SPG and all the baselines. This is a typical setting for RL-based fine-tuning of language models. As noted by Schulman & Lab (2025), LoRA performs equivalently to full fine-tuning for reinforcement learning even with small ranks since RL requires very low capacity.

To further validate this in our context, we provide additional results for SPG w/ Mixture using full fine-tuning, alongside those obtained with LoRA, in Table 18. Full fine-tuning yields comparable or even better performance relative to LoRA, demonstrating the robustness and consistent improvements of SPG. Note that full fine-tuning generally requires a smaller learning rate than LoRA. Accordingly, we use a learning rate of $2 \times 10^{-7}$ for MATH500 and $3 \times 10^{-7}$ for Countdown.

Table 16: Pass@K performance of each model on MATH500 and Countdown. We set temperature as 0.9 and report results of the best checkpoint of each case at a generation length of 256. For comparison, we also include the greedy sampling performance, i.e., Pass@1$_{\text{Greedy}}$. The best results are bolded and the second best are underlined.

| Model | MATH500 (0-shot) | | | | | Countdown (0-shot) | | | | |
|---|---|---|---|---|---|---|---|---|---|---|
| | Pass@1$_{\text{Greedy}}$ | Pass@1 | Pass@2 | Pass@3 | Pass@4 | Pass@1$_{\text{Greedy}}$ | Pass@1 | Pass@2 | Pass@3 | Pass@4 |
| LLaDA-8B-Instruct | 32.4 | 31.5 | 40.9 | 45.7 | 48.8 | 16.8 | 15.8 | 28.1 | 37.7 | 45.3 |
| LLaDA-1.5 | 32.2 | 32.6 | 42.2 | 47.4 | 50.4 | 21.1 | 18.2 | 32.1 | 42.5 | 50.0 |
| D1 | 37.8 | 34.3 | 43.1 | 48.0 | 52.0 | 32.4 | 24.5 | 40.4 | 51.4 | 60.6 |
| WD1 | 38.6 | 36.0 | 44.9 | 49.9 | 53.6 | 54.7 | 44.3 | 60.6 | 68.0 | 73.1 |
| UniGRPO | 38.4 | 34.7 | 43.9 | 49.5 | 53.2 | 44.9 | 36.8 | 55.2 | 65.0 | 72.3 |
| SPG w/ EUBO | 38.0 | 34.4 | 44.3 | 49.9 | 54.0 | **71.5** | **68.2** | 71.9 | 73.9 | **76.6** |
| SPG w/ mixture | **40.0** | **36.5** | **46.0** | **51.2** | **55.6** | 71.1 | 67.5 | **72.5** | **75.1** | **76.6** |

**Ablations on Prompt Masking.** We adopt the same prompt masking technique as D1, randomly masking some prompt tokens in the perturbed sample $z_t$ with a small probability during log-likelihood estimation. As discussed in Zhao et al. (2025), this stochastic masking creates perturbed versions of the same (prompt, completion) pairs, which acts as a regularizer for policy optimization. It can also be interpreted as data augmentation, enabling the extraction of more supervision signals from the same dataset.

Additionally, we report results for SPG w/ Mixture without applying the prompt masking trick in Table 19. In practice, prompt masking has minimal impact on SPG, yielding nearly identical average accuracy on both MATH and Countdown. This highlights the robustness of SPG's log-likelihood estimation method, in contrast to the one-step estimation used in D1, which already provides strong supervision signals even without prompt masking.

**Ablations on the Number of Monte Carlo Samples.** Using larger values of $m$ resulted in out-of-memory issues in our experiments, so we set $m = 2$ for efficiency. We also conducted experiments with $m = 4$, as shown in Table 20. Increasing $m$ leads to similar performance on MATH and better performance on Countdown. We also observe that the gradient norm for models trained with $m = 4$ is lower than that for $m = 2$ throughout the RL training steps.

**Computational Cost Comparisons.** We compare the computational cost of SPG with those of the baseline methods. The block-wise masking strategy in SPG does not introduce extra computational overhead relative to random masking, since the only difference lies in the formula for the distribution $q(\cdot|x^j)$ used in likelihood estimation. While Monte Carlo sampling does add some computation compared to the one-step estimation in D1, the primary computational cost for these RL algorithms comes from the rollout step. As a result, the impact of MC sampling on token throughput is minimal. We report the average time cost per gradient update on 8×A100 GPUs for GSM8K. As shown in Table 21, SPG has a computational time cost comparable to D1 and UniGRPO.

### H.3 Additional Evaluations

In this section, we provide additional evaluations on SPG's performance, including a detailed clarification of the differences between SPG and the UniGRPO baseline, as well as results from validation-based checkpoint selection.

**Difference with UniGRPO.** As shown in Table 5, even when both methods use the ELBO for log-likelihood estimation, SPG w/ ELBO and UniGRPO differ in several key aspects, including the policy update algorithm, masking strategy and other modifications.

We have demonstrated significant contributions of each of our algorithm components, particularly a better log-likelihood estimation method and the block-wise masking strategy, through the ablation studies in Table 3 and Table 4, where all other settings are held constant except for the component under investigation. We additionally experiment with UniGRPO using the block-wise masking strategy. As shown in Table 22, block-wise masking enhances UniGRPO's performance, but it still significantly underperforms SPG. This further underscores the effectiveness of SPG's novel log-likelihood estimation approach.

Table 17: Ablations on the inference strategy. SPG leads to consistently superior performance to baselines with different inference strategies. The best results are bolded and the second best are underlined for each setting. We report results for generation length 256.

| Inference Strategy | Model | GSM8K | MATH500 | Countdown | Sudoku | *Avg.* |
|---|---|---|---|---|---|---|
| Semi-AR, Confidence, Block=16 | LLaDA-8B-Instruct | 78.7 | 31.4 | 13.7 | 26.2 | 37.5 |
| | LLaDA-1.5 | 78.8 | 33.4 | 16.0 | 23.0 | 37.8 |
| | D1 | 79.7 | 37.2 | 27.0 | 31.4 | 43.8 |
| | WD1 | 82.3 | 37.4 | 53.9 | 36.8 | 52.6 |
| | UniGRPO | 82.5 | 36.8 | 46.5 | 63.4 | 57.3 |
| | SPG w/ EUBO | 84.7 | 37.4 | 70.3 | 82.2 | 68.7 |
| | SPG w/ Mixture | **86.4** | **40.8** | **70.7** | **96.2** | **73.5** |
| Semi-AR, Confidence, Block=32 | LLaDA-8B-Instruct | 77.2 | 32.4 | 16.8 | 27.7 | 38.5 |
| | LLaDA-1.5 | 80.5 | 32.2 | 21.1 | 26.9 | 40.2 |
| | D1 | 80.6 | 37.8 | 32.4 | 32.8 | 45.9 |
| | WD1 | 81.7 | 38.6 | 54.7 | 35.7 | 58.1 |
| | UniGRPO | 82.6 | 38.4 | 44.9 | 67.0 | 58.2 |
| | SPG w/ EUBO | 84.8 | 38.0 | **71.5** | 88.5 | 70.7 |
| | SPG w/ Mixture | **86.2** | **40.0** | 71.1 | **95.6** | **73.2** |
| Semi-AR, Confidence, Block=64 | LLaDA-8B-Instruct | 78.6 | 33.2 | 27.3 | 32.6 | 42.9 |
| | LLaDA-1.5 | 81.0 | 35.4 | 20.3 | 36.4 | 43.3 |
| | D1 | 80.9 | 37.6 | 38.3 | 39.8 | 49.2 |
| | WD1 | 82.5 | 37.4 | 52.3 | 41.8 | 53.5 |
| | UniGRPO | 82.3 | 37.4 | 53.5 | 82.9 | 64.0 |
| | SPG w/ EUBO | 84.3 | 37.4 | 69.5 | 88.8 | 70.0 |
| | SPG w/ Mixture | **85.5** | **41.4** | **69.9** | **93.8** | **72.7** |
| Semi-AR, Random, Block=32 | LLaDA-8B-Instruct | 63.5 | 21.0 | 6.3 | 24.4 | 28.8 |
| | LLaDA-1.5 | 67.1 | 24.8 | 10.9 | 27.5 | 32.6 |
| | D1 | 69.7 | 27.4 | 18.4 | 29.9 | 36.4 |
| | WD1 | 74.1 | 30.8 | 37.5 | 29.9 | 43.1 |
| | UniGRPO | 72.8 | 29.8 | 41.4 | 60.1 | 51.0 |
| | SPG w/ EUBO | 74.1 | **31.4** | 42.6 | 72.6 | 55.2 |
| | SPG w/ Mixture | **78.4** | 31.0 | **66.0** | **86.9** | **65.6** |
| Full Sequence, Random | LLaDA-8B-Instruct | 52.7 | 17.8 | 13.3 | 27.1 | 27.7 |
| | LLaDA-1.5 | 53.5 | 19.8 | 13.7 | 28.0 | 28.8 |
| | D1 | 53.0 | 22.6 | 28.5 | 31.8 | 34.0 |
| | WD1 | 54.6 | 23.4 | 43.4 | 30.0 | 37.9 |
| | UniGRPO | 57.6 | 22.6 | 47.7 | 57.3 | 46.3 |
| | SPG w/ EUBO | 58.5 | **25.2** | 56.6 | 71.3 | 52.9 |
| | SPG w/ Mixture | **61.2** | **25.2** | **64.1** | **72.2** | **55.7** |
| Full Sequence, Confidence | LLaDA-8B-Instruct | 23.9 | 17.8 | 0.0 | 68.3 | 27.5 |
| | LLaDA-1.5 | 41.4 | 20.4 | 0.0 | 67.9 | 32.4 |
| | D1 | 57.5 | 22.6 | 0.0 | 72.3 | 38.1 |
| | WD1 | 56.7 | **25.0** | 10.2 | 68.9 | 40.2 |
| | UniGRPO | 50.0 | 24.2 | 8.2 | 95.6 | 44.5 |
| | SPG w/ EUBO | 54.3 | 23.4 | 63.3 | 88.9 | 57.5 |
| | SPG w/ Mixture | **58.8** | 24.0 | **65.6** | **96.5** | **61.2** |

Table 18: Ablations on full fine-tuning on MATH500 and Countdown.

| SPG w/ Mixture | MATH500 (0-shot) | | | | Countdown (0-shot) | | | |
|---|---|---|---|---|---|---|---|---|
| **FT Setting / Seq Len** | **128** | **256** | **512** | *Avg.* | **128** | **256** | **512** | *Avg.* |
| LoRA | 33.4 | 40.0 | 41.8 | **38.4** | 68.8 | 70.7 | 70.3 | 69.9 |
| Full-FT | 34.0 | 39.8 | 40.6 | 38.1 | 78.5 | 76.6 | 76.2 | **77.1** |

Table 19: Ablations on prompt masking on MATH500 and Countdown.

| SPG w/ Mixture | MATH500 (0-shot) | | | | Countdown (0-shot) | | | |
|---|---|---|---|---|---|---|---|---|
| **Model / Seq Len** | **128** | **256** | **512** | *Avg.* | **128** | **256** | **512** | *Avg.* |
| SPG | 33.4 | 40.0 | 41.8 | **38.4** | 68.8 | 70.7 | 70.3 | **69.9** |
| SPG w/o prompt masking | 34.0 | 39.0 | 40.2 | 37.7 | 69.5 | 69.9 | 70.3 | **69.9** |

Table 20: Ablations on the number of Monte Carlo samples on MATH500 and Countdown.

| SPG w/ Mixture | MATH500 (0-shot) | | | | Countdown (0-shot) | | | |
|---|---|---|---|---|---|---|---|---|
| Model / Seq Len | 128 | 256 | 512 | *Avg.* | 128 | 256 | 512 | *Avg.* |
| $m = 2$ | 33.4 | 40.0 | 41.8 | **38.4** | 68.8 | 70.7 | 70.3 | 69.9 |
| $m = 4$ | 33.2 | 39.2 | 40.4 | 37.6 | 80.5 | 82.0 | 82.8 | **81.8** |

Table 21: Computational time cost of each method on GSM8K with $8 \times$A100 GPUs. We report the average time (in minutes) per gradient update.

| Model | D1 | UniGRPO | SPG |
|---|---|---|---|
| Time (min) | 0.49 | 0.51 | 0.51 |

Table 22: Comparison with UniGRPO with the block-wise masking strategy on MATH500 and Countdown. The best average accuracies across generation lengths are bolded and the second best are underlined.

| | MATH500 (0-shot) | | | | Countdown (0-shot) | | | |
|---|---|---|---|---|---|---|---|---|
| Model / Seq Len | 128 | 256 | 512 | *Avg.* | 128 | 256 | 512 | *Avg.* |
| UniGRPO | 32.4 | 37.4 | 39.4 | 36.4 | 44.5 | 43.0 | 57.0 | 48.2 |
| UniGRPO + block-wise masking | 33.2 | 35.6 | 41.0 | 36.6 | 46.1 | 54.3 | 53.1 | 51.2 |
| SPG w/ EUBO | 33.2 | 37.6 | 39.4 | 36.7 | 68.4 | 71.5 | 68.0 | 69.3 |
| SPG w/ Mixture | 33.4 | 40.0 | 41.8 | **38.4** | 68.8 | 70.7 | 70.3 | **69.9** |

**Validation-Based Checkpoint Selection.** For our main results, we select the checkpoint that achieves the best performance for each model. This is a widely used evaluation protocol in RL for language models, as demonstrated in prior work on dLLMs (Zhao et al., 2025; Tang et al., 2025) and LLMs (Wang et al., 2025c). Importantly, we apply this protocol consistently across all RL runs for every model and baseline, ensuring a fair comparison given the inherent variability at any specific evaluation point.

To further address potential test set overfitting issue, we also apply validation-based checkpoint selection for all models. Specifically, we randomly divide the original test set into two equal halves, designating one as the validation set and the other as the new test set. We then select checkpoints based on validation accuracy and report the average accuracy on the new test set across generation lengths 128, 256, and 512. The results, presented in Table 23, demonstrate that SPG maintains consistently superior performance compared to the baselines.

Table 23: Model performance using validation-based checkpoint selection. We split the original test set into two equal parts, using one half for validation to choose the best checkpoint, and then report the average accuracy across different generation lengths on the remaining test set. The best results are bolded and second best underlined. Across all tasks, SPG consistently surpasses all baselines.

| Model | GSM8K | MATH500 | Countdown | Sudoku |
|---|---|---|---|---|
| LLaDA-8B-Inst. | 75.8 | 31.7 | 18.2 | 19.3 |
| LLaDA-1.5 | 77.0 | 30.4 | 18.8 | 20.1 |
| D1 | 77.6 | 34.4 | 26.0 | 22.7 |
| WD1 | 79.7 | 34.9 | 48.4 | 28.9 |
| UniGRPO | 78.6 | 34.9 | 45.6 | 61.9 |
| SPG w/ EUBO | 80.4 | **36.9** | 64.6 | 83.5 |
| SPG w/ Mixture | **82.3** | 36.5 | **68.0** | **90.0** |

## H.4 Qualitative Examples on GSM8K

We present qualitative examples for all models on GSM8K, using a generation length of 256 and 128 diffusion steps. Incorrect reasoning steps or answers are highlighted in red and the correct answers are highlighted in green. SPG leads to more concise and accurate reasoning compared to other

models. Note that the models may sometimes achieve the correct answer through flawed reasoning, while such artifacts are expected to diminish with a stronger base model.

### Model: LLaDA-8B-Instruct

**Question:** Kylar went to the store to buy glasses for his new apartment. One glass costs $5, but every second glass costs only 60% of the price. Kylar wants to buy 16 glasses. How much does he need to pay for them?
**Model:** LLaDA-8B-Instruct
**Answer:**
To determine how much Kylar needs to pay for 16 glasses, we need to consider the cost pattern of the glass. The first glass costs $5, and every second glass costs 60% of the price. Therefore, the cost of every second glass is $0.6 \times 5 = 3$ dollars.
We can break down the cost as follows:

- The first glass costs $5.
- The second glass costs $3.
- The third glass costs $5.
- The fourth glass costs $3.
- And so on...

We can see that the cost alternates between $5 and $3. Since Kylar wants to buy 16 glasses, we need pair them up.
The first of each pair costs $5, and the second of each pair costs $3. Therefore, the cost of each pair is $5 + 3 = 8$ dollars.
Since there are 8 pairs of glasses (16 glasses divided into pairs), the total cost is $8 \times 8 = 64$ dollars.
</reasoning>
<answer>
64
</answer>

**Question:** Amy had two eyeshadow palettes with four colors each and three makeup sets that came with six eyeshadow colors each. Her sister steals one of the palettes. Amy uses up half of the colors from one makeup set. How many eyeshadow colors does she have left?
**Model:** LLaDA-8B-Instruct
**Answer:**
First, let's determine the total number of eyeshadow colors Amy initially has. She has two eyeshadow palettes with four colors each, $2 \times 4 = 8$. She also has three makeup sets with six colors each, $3 \times 6 = 18$. So, the total number of eyeshadow colors Amy initially has is: $8 + 18 = 26$. Next, her sister steals one of the palettes, which she loses completely. Therefore, the number of colors left from the remaining palettes is: $8 - 4 = 4$. Amy uses up half of the colors from one makeup set. Each makeup set has six colors, so half of that is: $\frac{6}{2} = 3$. Thus, the number of colors left from the one makeup set is: $6 - 3 = 3$. Adding, the total number of eyes left is: $4 + 3 = 11$.
</reasoning>
<answer>
11
</answer>

### Model: D1

**Question:** Kylar went to the store to buy glasses for his new apartment. One glass costs $5, but every second glass costs only 60% of the price. Kylar wants to buy 16 glasses. How much does he need to pay for them?
**Model:** D1
**Answer:**
To determine how much Kylar needs to pay for 16 glasses, we need to consider the cost pattern of the glass. The first glass costs $5, and every second glass costs 60% of the price. Therefore, the cost of every second glass is $5 \times 0.6 = 3$ dollars. We can break down the cost into two parts:

1. The cost of the first 8 glasses, each costing $5.

2. The cost of the next 8 glasses, each costing $3.

First, let's calculate the cost of the first 8 glasses: $8 \times 5 = 40$ dollars.
Next, let's calculate the cost of the next 8 glasses: $8 \times 3 = 24$ dollars.
Now, we add these two amounts together: $40 + 24 = 64$ dollars.
Therefore, Kylar needs to pay $64 for them.
</reasoning>
<answer>
64
</answer>

**Question:** Amy had two eyeshadow palettes with four colors each and three makeup sets that came with six eyeshadow colors each. Her sister steals one of the palettes. Amy uses up half of the colors from one makeup set. How many eyeshadow colors does she have left?
**Model:** D1
**Answer:**
Amy initially has two eyeshadow palettes with four colors each, so she has $2 \times 4 = 8$ colors from the palettes. She also has three makeup sets with six colors each, so she has $3 \times 6 = 18$ colors from the makeup sets. Her's sister steals one of the palettes, so she now has $8 - 1 = 7$ colors from the palettes. Amy uses up half of the colors from one makeup set, so she uses $\frac{6}{2} = 3$ colors from that set. Therefore, she has $18 - 3 = 15$ colors left from the makeup sets. Adding these to the remaining palettes palettes, she has a total of $7 + 15 = 22$ eyeshadow colors left.
</reasoning>
<answer>
22
</answer>

**Model: WD1**

**Question:** Kylar went to the store to buy glasses for his new apartment. One glass costs $5, but every second glass costs only 60% of the price. Kylar wants to buy 16 glasses. How much does he need to pay for them?
**Model:** WD1
**Answer:**
To determine how much Kylar needs to pay for 16 glasses, we need to consider the cost of both the full price and the discounted price glasses.
First, let's calculate the cost of the first 8 glasses, which are sold at the full price of $5 each: $8 \times 5 = 40$ dollars.
Next, let's calculate the cost of the next 8 glasses, which are sold at 60% of the full price.
Since 60% of $5 is: $0.6 \times 5 = 3$ dollars. So, the cost for the next 8 glasses is: $8 \times 3 = 24$ dollars.
Now, we add the cost of the first 8 glasses and the cost of the next 8 glasses: $40 + 24 = 64$ dollars.
Therefore, Kylar needs to pay $\boxed{64}$.
</reasoning>
<answer>
64
</answer>

**Question:** Amy had two eyeshadow palettes with four colors each and three makeup sets that came with six eyeshadow colors each. Her sister steals one of the palettes. Amy uses up half of the colors from one makeup set. How many eyeshadow colors does she have left?
**Model:** WD1
**Answer:**
First, let's determine the total number of eyeshadow colors Amy initially has. She has two eyeshadow palettes with four colors each: $2 \times 4 = 8$. Additionally, she has three makeup sets with six colors each: $3 \times 6 = 18$. So, together, she has: $8 + 18 = 26$. Next, her sister steals one of the palettes, which she loses 4 colors: $26 - 4 = 22$. Then, Amy uses up half of the colors from one makeup set. Each makeup set has six colors, so half of that is: $\frac{6}{2} = 3$. Thus, she uses 3 colors from one makeup set, leaving: $6 - 3 = 3$ colors in that makeup set.
Finally, we add the remaining colors from the makeup set to the remaining colors: $22 + 3 = 25$.
</reasoning>

<answer>
25
</answer>

**Model: UniGRPO**

**Question:** Kylar went to the store to buy glasses for his new apartment. One glass costs $5, but every second glass costs only 60% of the price. Kylar wants to buy 16 glasses. How much does he need to pay for them?
**Model:** UniGRPO
**Answer:**
To determine how much Kylar needs to pay for 16 glasses, we need to consider the cost of both the regular glasses and the discounted glasses. A regular glass costs $5, and every second glass costs 60% of that price, which is $5 \times 0.6 = 3$.
Since Kylar will buy 16 glasses, he will have 8 regular glasses and 8 discounted glasses. The cost of 8 regular glasses is $8 \times 5 = 40$ dollars. The cost of 8 discounted glasses is $8 \times 3 = 24$ dollars. Therefore, the total cost for 16 glasses is $40 + 24 = 64$ dollars.
</reasoning>
<answer>
64
</answer>

**Question:** Amy had two eyeshadow palettes with four colors each and three makeup sets that came with six eyeshadow colors each. Her sister steals one of the palettes. Amy uses up half of the colors from one makeup set. How many eyeshadow colors does she have left?
**Model:** UniGRPO
**Answer:**
First, let's determine the total number of eyeshadow colors Amy initially has. She has two eyeshadow palettes with four colors each, so she has $2 \times 4 = 8$ colors from the palettes. She also has three makeup sets with six colors each, so she has $3 \times 6 = 18$ colors from the makeup sets. Adding these together gives her initially $8 + 18 = 26$ eyeshadow colors. Next, her sister steals one of the palettes, so she now has $8 - 4 = 14$ colors from the palettes. She then uses up half of the colors from one makeup set. Since she used one makeup set, containing six colors, she used $\frac{6}{2} = 3$ colors from that set. Therefore, she has $18 - 3 = 15$ colors left from the makeup sets.
Finally, adding the remaining colors from the palettes and the makeup sets, Amy has $14 + 15 = 29$ eyeshadow colors left.
</reasoning>
<answer>
29
</answer>

**Model: SPG w/ EUBO**

**Question:** Kylar went to the store to buy glasses for his new apartment. One glass costs $5, but every second glass costs only 60% of the price. Kylar wants to buy 16 glasses. How much does he need to pay for them?
**Model:** SPG w/ EUBO
**Answer:**
To determine how much Kylar needs to pay for 16 glasses, we need to consider the cost structure of the glasses. The first glass costs $5, and every second glass costs 60% of the price of the first glass. Therefore, the cost of each second glass is $5.0 \times 0.6 = 3.0$ dollars.
Kylar wants to buy 16 glasses, which means he will have 8 pairs of glasses. Each pair consists of one first glass and one second glass. The cost of one pair is: $5.0 + 3.0 = 8.0$ dollars. Since there are 8 pairs, the total cost is: $8 \times 8.0 = 64.0$ dollars. Thus, Kylar needs to pay $\boxed{64}$ dollars for 16 glasses.
</reasoning>
<answer>
64
</answer>

**Question:** Amy had two eyeshadow palettes with four colors each and three makeup sets that came with six eyeshadow colors each. Her sister steals one of the palettes. Amy uses up half of the colors from one makeup set. How many eyeshadow colors does she have left?
**Model:** SPG w/ EUBO
**Answer:**
First, let's calculate the total number of eyeshadow colors Amy initially has:

- She has two eyeshadow palettes with four colors each: $2 \times 4 = 8$ colors.
- She has three makeup sets with six colors each: $3 \times 6 = 18$ colors.
- Adding these together gives: $8 + 18 = 2$

Next, we account for the changes:

- Her sister steals one palette, so she loses 4 colors: $26 - 4 = 22$
- Amy uses up half of the colors from one makeup set, which is $\frac{6}{2} = 3$ colors: $22 - 3 = 19$

Therefore, Amy has 19 eyeshadow colors left.
</reasoning>
<answer>
19
</answer>

## Model: SPG w/ Mixture

**Question:** Kylar went to the store to buy glasses for his new apartment. One glass costs $5, but every second glass costs only 60% of the price. Kylar wants to buy 16 glasses. How much does he need to pay for them?
**Model:** SPG w/ Mixture
**Answer:**

1. The first glass costs $5.
2. Every second glass costs 60% of $5, which is $0.6 \times 5 = 3$ dollars.
3. Kylar wants to buy 16 glasses, so he will have 8 full-priced glasses and 8 discounted glasses.
4. The cost for 8 full-priced glasses is $8 \times 5 = 40$ dollars.
5. The cost for 8 discounted glasses is $8 \times 3 = 24$ dollars.
6. Adding these the total cost is $40 + 24 = 64$ dollars.

</reasoning>
<answer>
64
</answer>

**Question:** Amy had two eyeshadow palettes with four colors each and three makeup sets that came with six eyeshadow colors each. Her sister steals one of the palettes. Amy uses up half of the colors from one makeup set. How many eyeshadow colors does she have left?
**Model:** SPG w/ Mixture
**Answer:**

1. Amy initially has two eyeshadow palettes with four colors each: $2 \times 4 = 8$ colors
2. She also has three makeup sets with six colors each: $3 \times 6 = 18$ colors
3. Adding these together gives gives: $8 + 18 = 26$ colors
4. Her sister steals one palette, so she loses: $4$ colors
5. Amy uses up half of the colors from one makeup set: $\frac{6}{2} = 3$ colors
6. Subtracting these lost and used colors: $26 - 4 - 3 = 19$ colors

</reasoning>
<answer>
19
</answer>

