# OpenReview forum: "SPG: Sandwiched Policy Gradient for Masked Diffusion Language Models"
_ICLR.cc/2026/Conference — ICLR 2026 Poster_

### Official Review · Reviewer_o6J6 · 2025-10-20

**Soundness:** 2
**Presentation:** 2
**Contribution:** 2
**Rating:** 4
**Confidence:** 4

**Summary:**

This paper introduces a policy gradient approach for RL fine-tuning diffusion LLMs (dLLMs). The authors not that the policy gradient involves the gradient of the log-likelihood of the policy, this log-likelihood being intractable in the case of dLLLMs. Existing approaches rely on an evidence lower bound (ELBO), which holds only when the return is positive. The authors build an evidence upper bound (EUBO) as a corollary of the Rényi variational bound, and use it for negative returns (ELBO being used for positive rewards). The authors also provide additional improvements. One is the mixing of ELBO and EUBO for negative returns, with a theoretical argument about decreasing variance (not discussing the bias though), the other one being a block-wise masking strategy for MC estimation. The authors then present a quite thorough experimental study, comparing to a number of baselines and ablating the different components of the proposed approach.

**Strengths:**

* The paper is clearly written and structured, easy to follow globally and well argued
* The point of the ELBO being no valid for negative rewards is a very good one, and the proposed approach makes a lot of sense, and provides good experimental results
* The empirical study is quite thorough with interesting ablations and relevant baselines

**Weaknesses:**

* From a reinforcement learning perspective, a lot of aspects regarding the proposed approach lack clarity. See questions for more details.
* The experiments may be presented in a too favorable aspect for the proposed approach. This may also be related to not enough discussed baselines (especially what are the key differences). See questions for more details.

**Questions:**

### Clarity on the reinforcement learning aspect

The reinforcement learning part is presented in a quite limited way, raising a number of questions.
* The underlying Markov decision process is not even defined, it could
* The fundamental argument of the paper is that the ELBO doesn’t hold for a negative reward. This is true. However, in RL, the optimal policy is invariant to a reward shifting. So in principle one can assume the reward to be positive without loss of generality. This may not be the case empirically speaking, for various reasons (one being the variance of the policy gradient, for example), but this would call for at least some discussion, ideally some baseline/ablation. The more close ablation is SPG w/ neg, which is something fundamentally different (it’s closer to SFT on positive examples, ignoring negative ones, a smooth version of best-of-n). Such a baseline could be Reinforce, but one could also imagine a GRPO variation with a normalization leading to positive advantages (using the fact that a state-dependent baseline does not bias the gradient). Please discuss this aspect as thoroughly as possible.
* Objective in Eq (4) is biased, because GRPO one is biased (because the baseline does depend on the sampled action). The correct way to do it would be to have a leave-one-out empirical expectation of the return, known as RLOO [A, B]. It’s not a big deal, probably doesn’t change much empirically speaking, but given that the whole point is about better sandwiching the policy objective, it is worth starting from an unbiased one.
* Obj in Eq (4), and then the proxy in Eq (5), do not consider importance sampling at all. However, it seems that most of the baselines are. Is the proposed approach purely on-policy, like Reinforce, or does it insures some off-policyness, like GRPO, but not taken into account in the loss, implying some additional bias in what is sandwiched ? (As a side remark, RLOO without importance sampling is not biased in the off-policy case, as a corollary of [C], but the overall discussion point remains)
* Many approaches in LLMs, but also in dLLMs, consider regularized RL (typically KL-regularization towards the initial model). It is not discussed here. Is it that no regularization is considered (which is perfectly fine, but worth discussing), or that it is skipped to lighten notations (but in this case it should not, could have implications)

### Too favorable aspect of experiments

This may be a wrong feeling, happy to be corrected, but it relies on the following points.

* As an initial side note, it would make sense to consider some baselines/ablations suggested above (related to the optimal policy invariance to shifting rewards).
* The main point is that the considered baselines (D1, WD1, UniGRPO) are not described enough (even considering the related works section in the appendix), and so we do not know what are the key differences with the proposed approach. The current ablation shows well that the different components are important to proposed approach, but what is missing is that the considered baselines are probably a form of ablation somehow. For example, SPG w/ ELBO seems pretty close to some of the baselines (like uniGRPO, but without IS/clipping?), how does it compare? Another example is that, if we look at the results of Table 1 for the baselines, and compare it to the results of SPG with the random masking strategy, the results are much closer (eg uniGRPO vs SPG w/ EUBO and random masking). So one may wonder if just applying the block-wise masking to uniGRPO would not lead to very good results too (weakening the EUBO contribution). Maybe it is not the case thanks to the mixing, but also maybe that the results are more nuanced then how they are showed (or not, but this would then call for a better discussion about baselines, their key differences, notably wrt SPG).
* Another point that presents too favorably the proposed approach is that the SPG specific hyper parameters ($\beta$ and $\omega$) are basically tuned on the test set, which biases too favorably SPG.


[A] Back to Basics: Revisiting REINFORCE Style Optimization for Learning from Human Feedback in LLM, Ahmadian et al., 2024
[B] Buy 4 REINFORCE Samples, Get a Baseline for Free! Kool et al, 2019.
[C] Contrastive Policy Gradient: Aligning LLMs on sequence-level scores in a supervised-friendly fashion, Flet-Berliac et al, 2024.

---

> ### Author Response · Authors · 2025-11-25
> **Response to Reviewer o6J6**
>
> We thank the reviewer for their time and feedback that helped us improve the work. In response to their review, we believe we have addressed their concerns as concretely as possible, including detailed discussions of the RL algorithms, and clarifications and additional results of the experiments. Below, we share our thoughts on the questions asked, and all of the discussions are also added to the revised manuscript in magenta.
>
> **Clarity on the reinforcement learning aspect**
> > The underlying Markov decision process is not even defined, it could.
>
> We thank the reviewer for pointing this out. The underlying Markov decision process can be defined similarly as in [a]. We have included a discussion in Section 2.2 in the updated manuscript.
>
> [a] Training Diffusion Models with Reinforcement Learning. ICLR 2024.
>
>
> > The fundamental argument of the paper is that the ELBO doesn’t hold for a negative reward. This is true. However, in RL, the optimal policy is invariant to a reward shifting. So in principle one can assume the reward to be positive without loss of generality. This may not be the case empirically speaking, for various reasons (one being the variance of the policy gradient, for example), but this would call for at least some discussion, ideally some baseline/ablation. The more close ablation is SPG w/ neg, which is something fundamentally different (it’s closer to SFT on positive examples, ignoring negative ones, a smooth version of best-of-n). Such a baseline could be Reinforce, but one could also imagine a GRPO variation with a normalization leading to positive advantages (using the fact that a state-dependent baseline does not bias the gradient). Please discuss this aspect as thoroughly as possible.
>
> We agree that in episodic RL with guaranteed termination and end-point reward only, or when KL is applied (the maximal entropy RL framework), optimal policy is invariant to reward shifting. However, this indicates that reward shifting no longer works when partial rewards are present (such as with long CoT traces), or in infinite horizon settings (e.g., in online learning). More importantly, even if reward shifting were possible, the learning dynamics might be different. It is well known in the literature of RL for language models that RL helps mitigate forgetting of pretrained models and improves exploration, compared to SFT only methods (while SFT could be viewed as assigning 0 reward to negative traces and 1 to positive traces). In fact, decreasing the log-likelihood of the negative traces is enlarging the entropy, thus improving exploration and avoiding overfitting. Such benefit is missing when we conduct the reward shifting. Therefore, we believe the SPG method is still beneficial.
>
> Empirically, we experiment with a reward-shifted objective, where the advantage of each trace is normalized by the minimum value in the batch, and apply both Reinforce and GRPO, using the ELBO as the log-likelihood estimation. The results are shown in the table below. Under both settings, the models underperform SPG.
>
> | Task | MATH500 | |  |  || Countdown | |  |  |
> |-----|-----|-----|-----|-----|-----|-----|-----|-----|-----|
> | Model / Seq Len | 128 | 256 | 512 | **avg.** || 128 | 256 | 512 | **avg.** |
> |||||||||||
> | LLaDA-8B-Inst. |  28.2 | 32.4 | 34.6 | 31.7 || 18.8 | 16.8 | 16.8 | 17.5  |
> |||||||||||
> | Reinforce + rwd. shift | 28.0 | 34.4 | 39.0 | 33.8 || 13.3 | 25.0 | 37.9 | 25.4 |
> | GRPO + rwd. shift | 24.4 | 33.4 | 36.2 | 31.3 || 23.1 | 23.8 | 48.1 | 31.6 |
> |||||||||||
> | SPG w/ EUBO | 33.2 | 37.6 | 39.4 | 36.7 || 68.4 | 71.5 | 68.0 | 69.3 |
> | SPG w/ Mixture | 33.4 | 40.0 | 41.8 | **38.4** || 68.8 | 70.7 | 70.3 | **69.9** |
>
> > Objective in Eq (4) is biased, because GRPO one is biased. The correct way to do it would be to have a leave-one-out empirical expectation of the return, known as RLOO [A, B]. It’s not a big deal, probably doesn’t change much empirically speaking, but given that the whole point is about better sandwiching the policy objective, it is worth starting from an unbiased one. As a side remark, RLOO without importance sampling is not biased in the off-policy case, as a corollary of [C], but the overall discussion point remains.
> [A] Back to Basics: Revisiting REINFORCE Style Optimization for Learning from Human Feedback in LLM, Ahmadian et al., 2024
> [B] Buy 4 REINFORCE Samples, Get a Baseline for Free! Kool et al, 2019.
> [C] Contrastive Policy Gradient: Aligning LLMs on sequence-level scores in a supervised-friendly fashion, Flet-Berliac et al, 2024.
>
> We thank the reviewer for pointing out the RLOO method to tackle the biases in the GRPO type of advantage calculation, and it is great to see that it can omit the off-policyness issue even without importance sampling. We have added a detailed discussion on this aspect in Appendix F, and leave further reducing the biases in the policy update objective for dLLMs with RLOO or other techniques as future works.

---

> ### Author Response · Authors · 2025-11-25
> **Response to Reviewer o6J6 (continued)**
>
> > Obj in Eq (4), and then the proxy in Eq (5), do not consider importance sampling at all. However, it seems that most of the baselines are. Is the proposed approach purely on-policy, like Reinforce, or does it insures some off-policyness, like GRPO, but not taken into account in the loss, implying some additional bias? Many approaches in LLMs, but also in dLLMs, consider regularized RL (typically KL-regularization towards the initial model). It is not discussed here.
>
> SPG incurs multiple gradient updates after one round of rollout. Thus, we agree with the reviewer that this would incur some off-policyness, and applying the importance sampling correction could be helpful. Regarding the KL regularization, we do not apply the KL regularization to our proposed SPG.
>
> In principle, we should be able to apply these techniques to SPG. For example, the same heuristic trick as in D1 or UniGRPO can be applied, using a one-step estimation or the ELBO as a proxy for log-likelihood in the importance sampling ratio and the KL divergence term. However, it remains unclear what is a principally correct and elegant way to incorporate these techniques to dLLMs, especially with the intractable log-likelihood challenge.
>
> Empirically, we found our current SPG algorithms already performant. Therefore, we chose to keep the current SPG in its simplest form. We leave correctly adapting KL regularizations and trust-region optimization to dLLMs as an interesting future direction. We have included a discussion regarding these RL stabilization techniques in Appendix F.
>
> **Too favorable aspect of experiments**
> > The considered baselines (D1, WD1, UniGRPO) are not described enough, and so we do not know what are the key differences with the proposed approach. The current ablation shows well that the different components are important to proposed approach, but what is missing is that the considered baselines are probably a form of ablation somehow. For example, SPG w/ ELBO seems close to some of the baselines (like uniGRPO, but without IS/clipping?), how does it compare? Another example is that, if we look at the results of Table 1 for the baselines, and compare it to the results of SPG with random masking, the results are much closer. So one may wonder if just applying the block-wise masking to uniGRPO would not lead to very good results too.
>
> We thank the reviewer for the great suggestion. We have added a detailed description along with a table to discuss and compare all the baselines in Appendix C and Table 5. As shown in the table below, the baselines mainly differ in the policy update algorithms, log-likelihood estimation, and masking strategy.
>
> | Model | Policy update alg. | Log-Likelihood Est. | Masking Strategy | Others |
> |-----|-----|-----|-----|-----|
> | LLaDA-1.5 | DPO | ELBO | random masking | Variance-reduced sampling budget allocation |
> | D1 | GRPO | one-step estimation | - | Prompt masking |
> | WD1 | Weighted policy optimization | one-step estimation | - | Prompt masking |
> | UniGRPO | GRPO | ELBO | random masking | Structured Noising Strategy |
> | SPG | Policy gradient | ELBO for positive traces; EUBO/Mixture for negative traces | block-wise masking | Prompt masking |
>
> In particular, SPG w/ ELBO is different from UniGRPO in **all** the perspectives of the table above except for the log-likelihood estimation part, including (1) UniGRPO uses GRPO which contains the importance sampling, clipping, and the KL regularization term while SPG uses a simple policy gradient; (2) UniGRPO uses random masking while SPG uses block-wise masking; (3) UniGRPO additionally applies a structured noising strategy that gradually increases the noise level of the perturbed sample along the gradient updates.
>
> We have demonstrated significant contributions of each of our algorithm components, a better log-likelihood estimation method and the block-wise masking strategy, through the ablation studies in Table 2 and Table 3 in Section 4.2, where all other settings are kept the same and the only difference is the mentioned changes of the specific component. To further address the reviewer’s concern about the contribution of a better log-likelihood estimation, we additionally experiment with UniGRPO with block-wise masking strategy. As shown in the table below, block-wise masking improves the performance of UniGRPO, while still significantly underperforming SPG. This further confirms the contribution of the log-likelihood estimation of SPG.
>
> | Task | MATH500 | |  |  || Countdown | |  |  |
> |-----|-----|-----|-----|-----|-----|-----|-----|-----|-----|
> | Model / Seq Len | 128 | 256 | 512 | **avg.** || 128 | 256 | 512 | **avg.** |
> ||||||||||||
> | UniGRPO | 32.4 | 37.4 | 39.4 | 36.4 || 44.5 | 43.0 | 57.0 | 48.2 |
> | UniGRPO + block-wise | 33.2 | 35.6 | 41.0 | 36.6 || 46.1 | 54.3 | 53.1 | 51.2 |
> ||||||||||||
> | SPG w/ EUBO | 33.2 | 37.6 | 39.4 | 36.7 || 68.4 | 71.5 | 68.0 | 69.3 |
> | SPG w/ Mixture | 33.4 | 40.0 | 41.8 | 38.4 || 68.8 | 70.7 | 70.3 | 69.9 |

---

> ### Author Response · Authors · 2025-11-25
> **Response to Reviewer o6J6 (continued 2)**
>
> > Another point that presents too favorably the proposed approach is that the SPG specific hyper parameters ($\beta$ and $\omega$) are basically tuned on the test set, which biases too favorably SPG.
>
> In all our experiments except for the ablations of $\omega$ in Figure 5 (e)-(f), we never tune the mixture weight $\omega$ and just set it to 0.5 as a heuristic.
>
> We also only tune $\beta$ very lightly, selecting the best value from $\beta \in \\{1.0, 1.5, 2.0\\}$. As shown in Figure 5, for most benchmarks except Sudoku, the model performance turns out to be fairly stable wrt $\beta$, especially for SPG w/ Mixture, outperforming all the baselines across different $\beta$ values. Empirically, $\beta=1.5$ turns out to be a generally good choice, except for Sudoku where we need a smaller $\beta=1.0$.
>
> We note that all other models also have their specific hyperparameters that are tuned, for example the weight of the KL regularization in D1 and UniGRPO and the hyperparameter $\lambda$ in WD1. Also, for most of the general hyperparameter settings, we simply follow the setting of D1 without tuning, which could also be tuned favorably for D1 but not for ours.

---

> > ### Comment · Reviewer_o6J6 · 2025-11-26
> > **Rebuttal acknowledgement**
> >
> > I thank the authors for the revised version and additional experiments, that answers my questions, i'll update my score accordingly.

---

> > > ### Author Response · Authors · 2025-11-26
> > >
> > > Thank you again for your insightful review and the increased score! We are glad that we have addressed your concerns and greatly appreciate your valuable suggestions.

---

### Official Review · Reviewer_bWLw · 2025-10-31

**Soundness:** 2
**Presentation:** 3
**Contribution:** 2
**Rating:** 4
**Confidence:** 5

**Summary:**

This paper proposes a new reinforcement learning (RL) algorithm for diffusion large language models (dLLMs), which have intractable likelihoods that make standard policy gradient methods infeasible. Prior approaches approximate the log-likelihood using the ELBO, but the lower-bound approximation introduces gradient bias and limits learning from negative rewards. To overcome this, the authors introduce the Sandwiched Policy Gradient (SPG) method, which optimizes a “sandwiched” objective combining both a lower bound (ELBO) for positive-reward samples and an upper bound (EUBO) for negative-reward samples, thereby reducing bias in the policy gradient. SPG further employs a block-wise masking strategy to stabilize Monte Carlo estimation and a mixture formulation that adaptively blends upper and lower bounds to reduce gradient variance. Experiments on four reasoning benchmarks—GSM8K, MATH500, Countdown, and Sudoku—show that SPG achieves consistent improvements.

**Strengths:**

- Originality:
A novel Sandwiched Policy Gradient (SPG) framework is proposed to leverage both lower and upper bounds of log-likelihood for diffusion LLM, which is a clear conceptual advance over prior ELBO-only RL approaches.

- Quality:
Theoretical development is coherent and well-motivated, with a solid integration of Rényi-based upper bounds and a mixture formulation that balances bias and variance.

- Clarity:
The paper is well-written and logically structured. Figures and equations clearly illustrate the SPG process.

**Weaknesses:**

1. Non-standard evaluation protocol.
The paper selects checkpoints every 100 steps based on the highest test accuracy, which risks test set overfitting. While this follows d1 for consistency, it is methodologically questionable. The model should instead be selected via a validation set or by reporting the final checkpoint performance, as adopted in [1]. Revising the evaluation protocol would improve the experimental rigor.

2. Absence of standard RL stabilization techniques.
SPG appears to use a naive policy gradient update without employing the clipping or KL-regularization mechanisms used in PPO or GRPO. Without importance sampling ratio corrections, it is unclear whether SPG remains stable for off-policy updates $\mu > 1$. This raises concerns about potential instability or divergence during training.

 3. Lack of comparison with key related work.
The paper omits comparison with [2], which is the first to successfully apply trajectory-level RL to diffusion LLMs and reports state-of-the-art results on similar reasoning benchmarks. Including this baseline is essential for contextualizing SPG’s improvements and validating its claimed advantage.


[1] DiffuCoder: Understanding and Improving Masked Diffusion Models for Code Generation, arXiv:2506.20639

[2] Reinforcing the Diffusion Chain of Lateral Thought with Diffusion Language Models, NeurIPS 2025.

**Questions:**

- Regarding Weakness 1: Would the authors consider adopting a more standard evaluation setup (e.g., validation-based checkpoint selection or reporting final-step results) to reduce overfitting concerns?

- Regarding Weakness 2: Could the authors clarify why SPG does not adopt clipping or KL regularization? Is this omission due to the intractability of computing importance ratios caused by the constant $C(T)$ term in EUBO, or was it omitted for simplicity?

- Regarding the tightness of the EUBO: As discussed in [3] and [4], the ELBO of dLLMs equals the AO-ARM loss, and becomes tight when the joint distribution $p(x|\sigma)$ is consistent across different orders of $\sigma$ (i.e., the inequality in Eq. (2) of [3] holds with equality). However, as shown in Appendix C.3 of this paper, the EUBO does not appear to be tight even under this condition. Could the authors explain why this phenomenon occurs, and whether it implies a fundamental looseness in the Rényi-based upper bound?

[3] AUTOREGRESSIVE DIFFUSION MODELS, ICLR 2022

[4] YOUR ABSORBING DISCRETE DIFFUSION SECRETLY  MODELS THE CONDITIONAL DISTRIBUTIONS OF  CLEAN DATA, ICLR 2025

---

> ### Author Response · Authors · 2025-11-25
> **Response to Reviewer bWLw**
>
> We thank the reviewer for their time and feedback that helped us improve the work. In response to their review, we believe we have addressed their concerns as concretely as possible, including clarification and additional results of the evaluation protocol, comparison to additional baselines, and detailed discussions of the RL stabilization techniques and the tightness of EUBO. Below, we share our thoughts on the questions asked, and all of the discussions are also added to the revised manuscript in magenta.
>
> > Non-standard evaluation protocol. The paper selects checkpoints every 100 steps based on the highest test accuracy, which risks test set overfitting. While this follows d1 for consistency, it is methodologically questionable. The model should instead be selected via a validation set or by reporting the final checkpoint performance, as adopted in [1]. Revising the evaluation protocol would improve the experimental rigor.
> Would the authors consider adopting a more standard evaluation setup (e.g., validation-based checkpoint selection or reporting final-step results) to reduce overfitting concerns?
> [1] DiffuCoder: Understanding and Improving Masked Diffusion Models for Code Generation, arXiv:2506.20639
>
> The practice in our paper is one of the standard evaluation protocols for RL of language models, which has been applied in both diffusion language models [a,b], and LLMs [c]. Besides, since we apply the same protocol consistently to all RL runs of all models and baselines, this is still a fair comparison across different algorithms given the inherent variance at any fixed evaluation point.
>
> We understand the reviewer’s concern on the test set overfitting issue. Thus, we additionally adopt the validation-based checkpoint selection for all the models. Specifically, we randomly split the original test data into two equally sized subsets, one as the validation set and the other as the new test set. We select the best checkpoint based on validation accuracy and report the average accuracy on the new test set across generation lengths 128, 256, and 512. The results are as below, which indicate the consistently superior performance of SPG over the baselines.
>
> [a] d1: Scaling Reasoning in Diffusion Large Language Models via Reinforcement Learning. NeurIPS 2025.
>
> [b] wd1: Weighted Policy Optimization for Reasoning in Diffusion Language Models. arXiv 2025.
>
> [c] Reinforcement Learning for Reasoning in Large Language Models with One Training Example. NeurIPS 2025.
>
> | Model | GSM8K | MATH500 | Countdown | Sudoku |
> |-----|-----|-----|-----|-----|
> | LLaDA-8B-Inst. | 75.8 | 31.7 | 18.2 | 19.3 |
> | LLaDA-1.5 | 77.0 | 30.4 | 18.8 | 20.1 |
> | D1 | 77.6 | 34.4 | 26.0 | 22.7 |
> | WD1 | 79.7 | 34.9 | 48.4 | 28.9 |
> | UniGRPO | 78.6 | 34.9 | 45.6 | 61.9 |
> ||||||
> | SPG w/ EUBO | 80.4 | **36.9** | 64.6 | 83.5 |
> | SPG w/ Mixture | **82.3** | 36.5 | **68.0** | **90.0** |
>
>
> > Absence of standard RL stabilization techniques. SPG appears to use a naive policy gradient update without employing the clipping or KL-regularization mechanisms used in PPO or GRPO. Without importance sampling ratio corrections, it is unclear whether SPG remains stable for off-policy updates $\mu>1$. This raises concerns about potential instability or divergence during training.
> Could the authors clarify why SPG does not adopt clipping or KL regularization? Is this omission due to the intractability of computing importance ratios caused by the constant C(T) term in EUBO, or was it omitted for simplicity?
>
> In principle, we should be able to adopt these standard stabilization techniques like clipping and KL regularization. For example, the same heuristic trick as in D1 or UniGRPO can be applied, using a one-step estimation or the ELBO as a proxy for log-likelihood in the importance sampling ratio and KL divergence terms. However, it remains unclear what is a principally correct and elegant way to incorporate these techniques to dLLMs, especially with the intractable log-likelihood challenge.
>
> Empirically, we found our current SPG algorithms already performant. Therefore, we chose to keep the current SPG in its simplest form. We leave correctly adapting RL stabilization to dLLMs as an interesting future direction, including how to use KL and trust-region optimization correctly for dLLMs.
>
> We have included a discussion regarding these RL stabilization techniques in Appendix F in the updated manuscript.

---

> ### Author Response · Authors · 2025-11-25
> **Response to Reviewer bWLw (continued)**
>
> > Lack of comparison with key related work. The paper omits comparison with [2], which is the first to successfully apply trajectory-level RL to diffusion LLMs and reports state-of-the-art results on similar reasoning benchmarks. Including this baseline is essential for contextualizing SPG’s improvements and validating its claimed advantage.
> [2] Reinforcing the Diffusion Chain of Lateral Thought with Diffusion Language Models, NeurIPS 2025.
>
> We thank the reviewer for pointing this out. We have added detailed discussions to [2] in the updated manuscript (Introduction, Appendix A, Appendix C, and Table 5). Notably, [2] is trained with both a trajectory-level policy update and an optimal decoding order prediction module. The second one to learn the best token decoding order is orthogonal to our approach focusing on optimizing the diffusion policy $\pi_{\theta}(\textbf{x}|\textbf{c})$ itself that generally applies to all different inference-time decoding strategies. While learning optimal decoding order is an interesting and promising direction for inference-time strategies, and SPG can be combined with learning optimal decoding order, this is out of the scope of this paper.
>
> For the RL algorithm component, we reimplement the trajectory-level RL method from [2] in our framework and report the results in the table below (namely “StepWise”). SPG consistently outperforms StepWise in both MATH500 and Countdown. Besides, in terms of computational cost, the trajectory-level update in [2] requires back propagation through each diffusion timestep, making it computationally expensive. For example, for MATH500, on 8xA100, StepWise requires 7.5min per gradient update, while SPG only requires 0.6min.
>
> | Task | MATH500 | |  |  || Countdown | |  |  |
> |-----|-----|-----|-----|-----|-----|-----|-----|-----|-----|
> | Model / Seq Len | 128 | 256 | 512 | **avg.** || 128 | 256 | 512 | **avg.** |
> |||||||||||
> | LLaDA-8B-Inst. |  28.2 | 32.4 | 34.6 | 31.7 || 18.8 | 16.8 | 16.8 | 17.5  |
> | LLaDA-1.5 |  26.8 | 32.2 | 35.8 | 31.6 || 21.9 | 21.1 | 21.5 | 21.5 |
> | D1 | 31.4 | 36.0 | 39.4 | 35.6 || 30.9 | 30.9 | 34.4 | 32.1 |
> | WD1 | 31.0 | 37.4 | 39.0 | 35.8 || 48.8 | 52.3 | 50.8 | 50.6 |
> | UniGRPO | 32.4 | 37.4 | 39.4 | 36.4 || 44.5 | 43.0 | 57.0 | 48.2 |
> | *StepWise*| 33.2 | 35.2 | 38.0 | 35.5 || 42.8 | 42.4 | 53.2 | 46.1 |
> |||||||||||
> | SPG w/ EUBO | 33.2 | 37.6 | 39.4 | 36.7 || 68.4 | 71.5 | 68.0 | 69.3 |
> | SPG w/ Mixture | 33.4 | 40.0 | 41.8 | **38.4** || 68.8 | 70.7 | 70.3 | **69.9** |
>
> We would also like to point out some differences in the experimental setups and implementations between ours (following the setting of D1) and those in [2]. For example, in the MATH task, [2] uses the whole MATH dataset as the training and eval data, while we use the subset MATH500. Also, we use LoRA for parameter efficient fine-tuning while [2] applies full fine-tuning, and the reward functions are defined differently. The generation setup is also different: we use 128 reverse diffusion steps for sequence length 256 while [2] use 256. Such difference is also reflected in the significant difference in the performance of the LLaDA-8B-Instruct base model: the accuracies of LLaDA-8B-Instruct on GSM8K and MATH reported in our paper is 77.2% and 32.4% respectively, very similar to those reported in the D1 paper (76.7% and 32.4%), while much lower than those reported in [2] (78.3% and 38.9%). Due to the time constraint of the rebuttal period, we could not present a complete comprehensive comparison. However, as noted, we have included a thorough discussion of [2] in our updated manuscript.

---

> ### Author Response · Authors · 2025-11-25
> **Response to Reviewer bWLw (continued 2)**
>
> > Regarding the tightness of the EUBO: As discussed in [3] and [4], the ELBO of dLLMs equals the AO-ARM loss, and becomes tight when the joint distribution $p(x|\sigma)$ is consistent across different orders of $\sigma$ (i.e., the inequality in Eq. (2) of [3] holds with equality). However, as shown in Appendix C.3 of this paper, the EUBO does not appear to be tight even under this condition. Could the authors explain why this phenomenon occurs, and whether it implies a fundamental looseness in the Rényi-based upper bound?
> [3] AUTOREGRESSIVE DIFFUSION MODELS, ICLR 2022
> [4] YOUR ABSORBING DISCRETE DIFFUSION SECRETLY MODELS THE CONDITIONAL DISTRIBUTIONS OF CLEAN DATA, ICLR 2025
>
> The ELBO in AO-ARM is over the latent $\sigma$ (ordering), hence the jenson gap is removed, whenever it is order-consistent ($p(x | \sigma)$ is consistent across all $\sigma$). In the Rényi bound, which is also a variational bound, the latent is $q$, which is the entire masking/noising path. The equality only holds when the likelihood ratio $p/q$ is almost surely constant under $q$. Hence even when it is order invariant, it can still be loose. In short, order-consistency makes the ELBO tight because it removes the Jensen gap over $\sigma$, but our Rényi EUBO is tight only when the forward path distribution $q$ matches the Rényi-tilted posterior over diffusion paths; since $q$ is fixed by the masking process and not equal to that tilted posterior in general, the EUBO remains strictly above log likelihood even under order consistency.
>
> However, we note that the consistency condition across all different orders is a very strong condition and rarely attainable in practice. It implies that the model outputs the same likelihood under all $L!$ orders ($L$ is the sequence length) for any data point, and would require a huge amount of data, maximal model capacity, as well as perfect optimization. Actually, in practice, certain orders are easier to model than others.

---

> > ### Comment · Reviewer_bWLw · 2025-11-26
> >
> > Thank the authors for their detailed revision and additional experiments, which answered most of my questions. I decide to update my score.

---

> > > ### Author Response · Authors · 2025-11-26
> > >
> > > Thank you again for your insightful review and the increased score! We are glad that we have addressed most of your concerns and greatly appreciate your valuable suggestions.

---

### Official Review · Reviewer_JppV · 2025-11-01

**Soundness:** 3
**Presentation:** 3
**Contribution:** 3
**Rating:** 6
**Confidence:** 3

**Summary:**

This paper addresses policy gradients for Diffusion language models where $\log \pi_{\theta} (x |c)$ is intractable by optimizing a lower bound for positive-advantage traces and an upper bound for negative advantage traces with a practical block-wise masking estimator. Because dLLMs do not expose a tractable log-likelihood, prior works optimize ELBO or one-step proxies, which bias gradients (ELBO is only a lower bound). Sandwiched Policy Gradient proposes to "sandwich" the objective: maximize a tractable lower bound $L_{ELBO}$ on positive-advantage samples while minimizing a tractable upper bound $U_{EUBO}$ on negative advantage samples (For instance, GRPO).  The problem is well-defined with clear mathematical analysis and extensive experiments on RL baselines.

**Strengths:**

1. Very clear problem: ELBO-only training biases gradients when rewards can be negative; upper bounds make the algorithm penalize low-reward traces without relying on true $\log \pi$.
2. Strong experimental results: 4 reasoning tasks achieve very high improvement/performance gains.
3. The theory is reasonable. EUBO derived from a Renyi variational inequality gives a tractable surrogate. Mixing lower/upper bounds reduces variance both intuitively and theoretically (Prop. 1) and avoids gradient clipping/vanishing.

**Weaknesses:**

1. Baselines. Comparisons focus on GRPO-like/diffusion-RL. Missing: preference optimization tailored to diffusion (DPO-style for dLLMs), score-function or pathwise estimators based on learned surrogates or likelihood-ratio via learned decoder controls.
2. The datasets are limited to reasoning tasks. Though the performance gains are significant, the datasets are limited to math/logic. No tool-use or multi-turn agents settings. No natural-language preferences.
3. Training cost vs baselines and wall-clock time are not clearly examined. Blockwise MC might add computational overhead. Can the authors report that the computational overhead for each method?

**Questions:**

What’s the wall-clock and token-throughput overhead of block-wise MC vs random masking and vs GRPO/D1/W1, at equal budgets?
Could the authors add (even small-scale) comparisons to (a) DPO-style for diffusion, (b) score-matching/pathwise surrogates with a learned decoder, (c) GRPO + stronger variance reduction?
Can the authors add more datasets? (non-reasoning, multi-turn, tool-use benchmarks)

---

> ### Author Response · Authors · 2025-11-25
> **Response to Reviewer JppV**
>
> We would like to thank the reviewer for their insightful review that helped us improve our work and their positive feedback regarding our proposed method, experiments, and theoretical analysis. In response to their review, we believe we have addressed their questions and concerns as concretely as possible. The discussions are also added to the revised manuscript in magenta.
>
>
> > Baselines. Comparisons focus on GRPO-like/diffusion-RL. Missing: preference optimization tailored to diffusion (DPO-style for dLLMs), score-function or pathwise estimators based on learned surrogates or likelihood-ratio via learned decoder controls.
> Could the authors add (even small-scale) comparisons to (a) DPO-style for diffusion, (b) score-matching/pathwise surrogates with a learned decoder, (c) GRPO + stronger variance reduction?
>
> LLaDA-1.5 applies the VRPO method to conduct fine-tuning, which is actually a preference optimization method applying the DPO objective and using the ELBO as log-likelihood estimation. We have added a detailed description along with a table to discuss and compare all the baselines in Appendix C and Table 5.
>
> Regarding the other baselines mentioned, for clarity, we would appreciate it if the reviewer could provide more details or references on what is meant by “score-matching/pathwise surrogates with a learned decoder” and “GRPO + stronger variance reduction”.
>
>
> > The datasets are limited to reasoning tasks. Though the performance gains are significant, the datasets are limited to math/logic. No tool-use or multi-turn agents settings. No natural-language preferences.
> Can the authors add more datasets? (non-reasoning, multi-turn, tool-use benchmarks)
>
> One key advantage and motivation of applying RL algorithms to language models during post-training is its great capability to improve reasoning ability of the model, as emphasized by Deepseek-R1 and D1. Therefore, we focus on reasoning tasks in our experiments following these papers.
>
> Since dLLMs are still at their early stage of development compared to established autoregressive LLMs, frameworks for advanced multi-turn agentic workflows or tool use remain largely underdeveloped. To the best of our knowledge, the only precedent is the recent update of Inception Labs’ Mercury model, which is a fully closed-source proprietary model. Therefore, successfully adapting dLLMs to multi-turn agentic settings itself represents a distinct and significant research challenge and falls outside the scope of this work.
>
> In addition to the math and logic tasks presented in our paper, we additionally experiment in the coding task, fine-tuning the LLaDA-8B-Instruct model on the KodCode-Light-RL-10K dataset and evaluate the performance on HumanEval and MBPP following the setting in D1. SPG consistently outperforms all the baselines, demonstrating its effectiveness in a non-reasoning setting.
>
> | Task | HumanEval | |  |  || MBPP | |  |  |
> |-----|-----|-----|-----|-----|-----|-----|-----|-----|-----|
> | Model / Seq Len | 128 | 256 | 512 | **avg.** || 128 | 256 | 512 | **avg.** |
> |||||||||||
> | LLaDA-8B-Inst. |  28.1 | 35.4 | 34.8 | 32.7 || 36.2 | 42.0 | 38.1 | 38.8  |
> | LLaDA-1.5 |  22.6 | 34.8 | 32.3 | 29.9 || 33.5 | 44.8 | 40.1 | 39.4 |
> | D1 | 29.3 | 37.8 | 37.2 | 34.8 || 42.0 | 45.5 | 43.2 | 43.6 |
> | WD1 | 25.6 | 39.0 | 38.4 | 34.4 || 38.9 | 45.9 | 44.8 | 43.2 |
> | UniGRPO | 28.1 | 39.6 | 38.4 | 35.4 || 39.3 | 45.5 | 45.9 | 43.6 |
> |||||||||||
> | SPG w/ EUBO | 29.3 | 41.5 | 33.5 | 34.8 || 43.6 | 46.7 | 46.3 | 45.5 |
> | SPG w/ Mixture | 31.1 | 39.0 | 39.6 | **36.6** || 46.3 | 50.6 | 49.0 | **48.6** |
>
>
> > Training cost vs baselines and wall-clock time are not clearly examined. Blockwise MC might add computational overhead. Can the authors report that the computational overhead for each method?
> What’s the wall-clock and token-throughput overhead of block-wise MC vs random masking and vs GRPO/D1/W1, at equal budgets?
>
> The block-wise masking does not incur additionally computational overhead compared to random masking, as the only difference is the formula of the distribution $q(\cdot|\mathbf{x}^j)$ in the likelihood estimations.
>
> Monte Carlo sampling incurs additional computation compared to the one-step estimation used by D1. However, since the main computational cost of these RL algorithms lies in the rollout step, the MC only leads to very marginal differences in the token throughput. Below, we list the average time cost per gradient update on 8xA100 for GSM8K. SPG has similar computational time cost as D1 and UniGRPO.
>
> | Model | D1 | UniGRPO | SPG |
> |-----|-----|-----|-----|
> | Time (min) | 0.49 | 0.51 | 0.51 |

---

### Official Review · Reviewer_Vqzb · 2025-11-01

**Soundness:** 3
**Presentation:** 3
**Contribution:** 3
**Rating:** 6
**Confidence:** 3

**Summary:**

The paper introduces SPG for dLLMs. Because dLLMs have intractable log-likelihoods, standard policy gradients cannot be applied directly. SPG sandwiches the likelihood by maximizing an ELBO for positive advantage samples and minimizing an EUBO for negative advantage samples, paired with blockwise masking for Monte Carlo estimation. SPG achieves SOTA accuracy on GSM8K, MATH500, Countdown, and Sudoku among RL for dLLMs.

**Strengths:**

1. The paper articulates the RL bottleneck for dLLMs and motivates a natural sandwiched approach.

2. Experimental results show consistent gains across benchmarks, sequence lengths, and inference strategies; ablations on some choices further support robustness.

3. The paper is easy to follow and uses a compact structure that makes the method clear.

**Weaknesses:**

1. LoRA-only evidence without full fine-tuning controls: all experiments rely on LoRA, so findings may hinge on LoRA’s inductive biases; it remains unclear whether the gains and stability persist under full fine-tuning.

2. The authors mention that masking 15% of the prompt can improve performance. Although they claim this choice follows d1, they do not seem to explain here the motivation or why it leads to better results.

**Questions:**

1. In all main experiments, the paper set the number of Monte Carlo samples to $m=2$. Ignoring computational constraints, how does increasing $m$ affect optimization variance and end-task performance?

2. Figure 9 depicts the training dynamics of the effective generation length. What do these trajectories imply about how SPG allocates its reasoning budget over training? For tasks such as Countdown and Sudoku, the effective length shows large-amplitude fluctuations; what factors drive these oscillations?

---

> ### Author Response · Authors · 2025-11-25
> **Response to Reviewer Vqzb**
>
> We would like to thank the reviewer for their insightful review that helped us improve our work and their positive feedback regarding our proposed method and experiments. In response to their review, we believe we have addressed their questions and concerns as concretely as possible. The discussions are also added to the revised manuscript in magenta.
>
>
> > LoRA-only evidence without full fine-tuning controls: all experiments rely on LoRA, so findings may hinge on LoRA’s inductive biases; it remains unclear whether the gains and stability persist under full fine-tuning.
>
> We follow the setting of d1 and wd1, using LoRA for SPG and all the baselines. Actually, this is a typical setting for RL-based fine-tuning of language models. As mentioned in [a], LoRA performs equivalently to full fine-tuning for reinforcement learning even with small ranks since RL requires very low capacity.
>
> To further validate this for our method, we provide additional results of SPG w/ Mixture using full fine-tuning compared with those using LoRA in the table below. Using full fine-tuning leads to similar or even better performance compared to LoRA, verifying the consistent gains and stability of SPG.
>
> [a] LoRA Without Regret. John Schulman and Thinking Machines Lab. https://thinkingmachines.ai/blog/lora/
>
>
> | Task | MATH500 | |  |  || Countdown | |  |  |
> |-----|-----|-----|-----|-----|-----|-----|-----|-----|-----|
> | Model / Seq Len | 128 | 256 | 512 | **avg.** || 128 | 256 | 512 | **avg.** |
> |||||||||||
> | LoRA | 33.4 | 40.0 | 41.8 | 38.4 || 68.8 | 70.7 | 70.3 | 69.9 |
> | Full-FT | 34.0 | 39.8 | 40.6 | 38.1 | |78.5 | 76.6 | 76.2 | 77.1 |
>
>
>
>
> > The authors mention that masking 15% of the prompt can improve performance. Although they claim this choice follows d1, they do not seem to explain here the motivation or why it leads to better results.
>
> We follow d1 to apply the same trick. As explained in Section 3.2, paragraph 2 of the d1 paper, “Intuitively, this stochastic masking introduces perturbed views of the same (prompt, completion) pairs, serving as a form of regularization for policy optimization. It can also be viewed as a form of data augmentation, extracting more supervision signals from the same data”. We have added this discussion to our updated manuscript. Experimental results in section 4.3 and Figure 5 of the d1 paper demonstrate the empirical effectiveness of this choice.
>
> We further provide additional results of SPG w/ Mixture without the prompt masking trick in the table below. Actually, the prompt masking has very little effect on our SPG, leading to almost the same average accuracy on MATH and Countdown. This reflects the robustness of the log-likelihood estimation method of SPG compared to the one-step estimation in d1, which already provided rich supervision signals even without this additional prompt masking trick.
>
> | Task | MATH500 | |  |  || Countdown | |  |  |
> |-----|-----|-----|-----|-----|-----|-----|-----|-----|-----|
> | Model / Seq Len | 128 | 256 | 512 | **avg.** || 128 | 256 | 512 | **avg.** |
> |||||||||||
> | SPG | 33.4 | 40.0 | 41.8 | 38.4 || 68.8 | 70.7 | 70.3 | 69.9 |
> | SPG w/o prompt masking | 34.0 | 39.0 | 40.2 | 37.7 || 69.5 | 69.9 | 70.3 | 69.9 |
>
>
> > In all main experiments, the paper set the number of Monte Carlo samples to m=2. Ignoring computational constraints, how does increasing m affect optimization variance and end-task performance?
>
> Larger m leads to out of memory issues in our experiments. Thus, we set m=2 for efficiency. We additionally experiment with m=4 as the table below. Increasing m leads to similar performance in MATH and better performance in Countdown. We also notice that the gradient norm for the model trained with m=4 is smaller than that of m=2 across the RL steps.
>
> | Task | MATH500 | |  |  || Countdown | |  |  |
> |-----|-----|-----|-----|-----|-----|-----|-----|-----|-----|
> | Model / Seq Len | 128 | 256 | 512 | **avg.** || 128 | 256 | 512 | **avg.** |
> |||||||||||
> | m=2 | 33.4 | 40.0 | 41.8 | 38.4 || 68.8 | 70.7 | 70.3 | 69.9 |
> | m=4 | 33.2 | 39.2 | 40.4 | 37.6 || 80.5 | 82.0 | 82.8 | 81.8 |
>
> > Figure 9 depicts the training dynamics of the effective generation length. What do these trajectories imply about how SPG allocates its reasoning budget over training? For tasks such as Countdown and Sudoku, the effective length shows large-amplitude fluctuations; what factors drive these oscillations?
>
> In general, SPG tends to generate more precise reasoning paths, using smaller effective length while still achieving superior performance compared with the baselines. Regarding the oscillations, for Sudoku, since the y-axis scale of the effective length is from 252 to 256, thus, SPG’s effective length varying from 253 to 256 is not a significant fluctuation. For Countdown, we do not fully understand why rollout lengths fluctuate. However, from the length changes, we could say that D1, WD1, and SPG lead to different training dynamics.

---

> > ### Comment · Reviewer_Vqzb · 2025-11-26
> >
> > I thank the authors for their response. I have no further questions and will maintain my score.

---

> > > ### Author Response · Authors · 2025-11-26
> > >
> > > Thank you again for your insightful review and for maintaining a positive score on our paper! We greatly appreciate your valuable suggestions and are glad that we have addressed your concerns.

---

### Author Response · Authors · 2025-11-25
**Consolidated response to all reviewers and the AC**

We thank all the reviewers for their constructive feedback that helped us improve our paper. Based on these comments, we have revised our manuscript, with the changes highlighted in magenta. Below we provide a brief summary of the key revisions and clarifications. Please refer to the responses to individual reviewers for more details.

* Additional experiments: we conduct experiments on the coding task, further validating the superior performance of SPG (reviewer JppV). We also conduct additional ablations to better understand the behavior of SPG (reviewer Vqzb), analyze its computational cost (reviewer JppV), and include an additional trajectory-level RL baseline (reviewer bWLw).
* Clarifications of experimental results: we have included a detailed description on each baseline to better demonstrate the difference between SPG and the baselines (reviewer o6J6). We also conduct a validation-based checkpoint selection to demonstrate the robust superior performance of SPG under such evaluation protocol (reviewer bWLw).
* Discussions on RL: we add additional explanations on our algorithm details, including discussions on RL stabilization techniques (reviewer bWLw and o6J6), comparison with a reward-shifting variant (reviewer o6J6), and discussions on techniques to reduce GRPO biases (reviewer o6J6).

---

### Meta-Review · Area_Chair_B8wX · 2026-01-07

**Summary:**

The reviewers were split about this paper and did not come to a consensus. On one hand they appreciated the clarity of the paper and the theoretical development. On the other they had issues with (a) the evaluation procedure and (b) missing baselines. The authors responded convincingly to (a) and (b). Both of the reviewers recommending rejection have acknowledged the rebuttal and stated that it answers their questions. They have increased their score. Given that all of their concerns are addressed, there is no further reason to recommend rejection. I vote to accept.

**Reviewer Concerns:**

Please see above.

**Reviewer Scores:**

I believe the reviewers would have kept or increased their scores.

---

### Decision · Program_Chairs · 2026-01-26

Accept (Poster)